# Heterochromatin formation and remodeling by IRTKS condensates counteract cellular senescence

Jia Xie [1,2], Zhao-Ning Lu [1,2], Shi-Hao Bai [1,2], Xiao-Fang Cui[1], He-Yuan Lian [1], Chen-Yi Xie [1], Na Wang [1], Lan Wang[1] & Ze-Guang Han [1]✉

## Abstract

Heterochromatin, a key component of the eukaryotic nucleus, is fundamental to the regulation of genome stability, gene expression and cellular functions. However, the factors and mechanisms involved in heterochromatin formation and maintenance still remain largely unknown. Here, we show that insulin receptor tyrosine kinase substrate (IRTKS), an I-BAR domain protein, is indispensable for constitutive heterochromatin formation via liquid–liquid phase separation (LLPS). In particular, IRTKS droplets can infiltrate heterochromatin condensates composed of HP1α and diverse DNA-bound nucleosomes. IRTKS can stabilize HP1α by recruiting the E2 ligase Ubc9 to SUMOylate HP1α, which enables it to form larger phase-separated droplets than unmodified HP1α. Furthermore, IRTKS deficiency leads to loss of heterochromatin, resulting in genome-wide changes in chromatin accessibility and aberrant transcription of repetitive DNA elements. This leads to activation of cGAS-STING pathway and type-I interferon (IFN-I) signaling, as well as to the induction of cellular senescence and senescence-associated secretory phenotype (SASP) responses. Collectively, our findings establish a mechanism by which IRTKS condensates consolidate constitutive heterochromatin, revealing an unexpected role of IRTKS as an epigenetic mediator of cellular senescence.

**Keywords** IRTKS; Heterochromatin Formation; Liquid-Liquid Phase Separation; SUMOylation; Cellular Senescence
**Subject Category** Chromatin, Transcription & Genomics

## Introduction

Insulin receptor tyrosine kinase substrate (IRTKS), also known as BAI1-associated protein 2-like 1 (BAIAP2L1), is a member of the IRSp53 (BAR/IMD domain containing adapter protein 2, BAIAP2)/MIM (missing-in-metastasis) homology domain family, which has been known to exert an important function in the formation of plasma membrane protrusions (Ahmed et al, 2010; Hu et al, 2000; Millard

et al, 2007). Similar to IRSp53, the founder of the inverse-BAR (I-BAR) domain family, IRTKS can mediate the clustering of actin bundles, extracellular vesicles and microvillus biogenesis and trigger pathogen-driven actin base formation (Aitio et al, 2010; Crepin et al, 2010; de Poret et al, 2022; Gaeta et al, 2021; Vingadassalom et al, 2009). However, unlike IRSp53, IRTKS displays diverse functions in different subcellular locations due to the lack of a CRIB domain that can bind to activated Cdc42 tethered to the plasma membrane, and it has also been associated with some diseases.

Recently, we and others have revealed that IRTKS plays crucial roles in insulin signal transduction, antiviral immunity, and tumorigenesis (Huang et al, 2018; Huang et al, 2013; Wang et al, 2013; Wu et al, 2019; Xia et al, 2015). In the presence of insulin stimulation, IRTKS can bind to insulin receptors on the cell membrane, resulting in enhancement of insulin signaling and its downstream pathway, whereas IRTKS deficiency decreases insulin sensitivity, leading to insulin resistance, including hyperglycemia, hyperinsulinemia, and glucose intolerance (Huang et al, 2013). In addition, IRTKS may suppress SHIP2 phosphoinositide phosphatase activity, which can convert phosphatidylinositol 3,4,5-triphosphate (PIP3) to phosphatidylinositol (3,4) bisphosphate (PI(3,4)P2), causing PIP3 accumulation on the cell membrane and subsequent AKT–mTOR signaling activation and cell proliferation (Wu et al, 2019). Interestingly, IRTKS is involved in the negative regulation of innate antiviral immunity (Xia et al, 2015), suggesting that IRTKS functions as a negative modulator of excessive inflammation.

More work has focused on the role of IRTKS/BAIAP2L1 in tumorigenesis. Previous reports showed that aberrant chromosomal rearrangement generated the constitutively activated fusion gene FGFR3-BAIAP2L1 in bladder cancer, lung cancer, and other cancers (Nakanishi et al, 2015; Williams et al, 2013). We and others reported that IRTKS was overexpressed in liver cancer (Wang et al, 2013), gastric cancer (Huang et al, 2018), ovarian cancer (Chao et al, 2015), and colorectal cancer (Wang et al, 2021). Most studies suggested that elevated IRTKS expression or the activated fusion gene FGFR3-BAIAP2L1 promoted tumor growth and progression by activating FGFR3 (Nakanishi et al, 2015), epidermal growth factor receptor (EGFR) signaling (Wang et al, 2013), and known membrane protrusions.

However, we noticed that, in addition to the above functions related to the cell membrane and cytoplasm, IRTKS may also play an

[1]Key Laboratory of Systems Biomedicine (Ministry of Education) and State Key Laboratory of Medical Genomics, National Research Center for Translational Medicine at Shanghai, Shanghai Center for Systems Biomedicine, Shanghai Jiao Tong University, Shanghai 200240, China. [2]These authors contributed equally: Jia Xie, Zhao-Ning Lu, Shi-Hao Bai. ✉E-mail: hanzg@sjtu.edu.cn

important role within the cell nucleus. Our studies showed that IRTKS overexpression promoted tumor suppressor p53 ubiquitination and degradation in gastric cancer cells by recruiting the p53-specific E3 ubiquitin ligase MDM2 (Huang et al, 2018). During viral infection, IRTKS can recruit ubiquitin-conjugating enzyme 9 (Ubc9) to SUMOylate PCBP2 within the cell nucleus, leading to cytoplasmic translocation of PCBP2 and then triggering degradation of the mitochondrial adapter MAVS to downregulate the RIG-I antiviral response (Xia et al, 2015). Interestingly, we also found that IRTKS may increase H3K9me3 level by promoting SETDB1 accumulation (Cui et al, 2023). These observations raise the possibility that IRTKS, as a nuclear protein, could play a significant role in regulating the eukaryotic genome and chromatin, a role not previously recognized.

Surprisingly, this study showed that, as *IRTKS* is lost, heterochromatin, as an essential architectural feature of the eukaryotic genome and chromatin, is dramatically decreased in mouse and human cells. Heterochromatin is essential for the regulation of genome stability, gene expression, and cellular functions (Janssen et al, 2018). The prevalent view is that the formation and maintenance of heterochromatin requires the critical heterochromatin protein 1-alpha (HP1α), which recognizes histone H3 lysine 9 trimethylation (H3K9me3) via its N-terminal chromo domain (CD) and provides a binding interface for diverse proteins, including H3K9 methyltransferase (H3K9 MT), through the C-terminal chromo shadow domain (CSD) (Allshire and Madhani, 2018; Maeda and Tachibana, 2022). Constitutive heterochromatin, the repressive and condensed state of chromatin, suppresses the transcriptional activity of many repetitive DNA sequences, including long interspersed elements (LINEs), short interspersed elements (SINEs), and long terminal repeat (LTR)-retrotransposons, for genome integrity (Nishibuchi and Dejardin, 2017; Zhang et al, 2020b). Global heterochromatin loss with abnormal activation of repetitive sequences has been associated with cellular senescence, aging, and aging-related diseases (Gorbunova et al, 2021; Hu et al, 2020; Liang et al, 2022a; Liang et al, 2021). Significantly, recent studies have reported that liquid–liquid phase separation (LLPS) was implicated in heterochromatin formation (Erdel et al, 2020; Keenen et al, 2021; Larson et al, 2017; Sanulli et al, 2019; Strom et al, 2017; Wang et al, 2019).

Interestingly, our data indicate that IRTKS condensates are indispensable for heterochromatin, where IRTKS recruits the E2 ligase Ubc9 to SUMOylate and stabilize HP1α and co-phase separates with unmodified and SUMOylated HP1α condensates. Unexpectedly, the depletion of IRTKS gives rise to global loss of heterochromatin, leading to increased chromatin accessibility and aberrant transcription of repetitive DNA sequences, which then trigger cellular senescence. Collectively, these data provide an innovative perspective on the mechanism by which LLPS drives IRTKS-mediated heterochromatin formation by cooperating with HP1α, uncovering a new role for the I-BAR domain protein IRTKS in the regulation of heterochromatin and cellular senescence.

## Results

### IRTKS is required for heterochromatin formation

To investigate whether IRTKS is essential for heterochromatin organization, we first surveyed the status of heterochromatin in tissues from wild-type (WT) and *Irtks* knockout (KO) mice.

Unexpectedly, a significantly notable reduction of heterochromatin at the nuclear periphery and around the nucleolus, characterized by electron-dense regions (EDRs) in transmission electron microscopy (TEM), was observed in the livers, kidneys and stomachs (Figs. 1A–D and EV1A–E) of *Irtks* KO mice. Consistently, TEM images also showed a visible loss of heterochromatin at the nuclear periphery and around the nucleolus in mouse embryonic fibroblasts (MEFs) of *Irtks* KO mice. In a rescue experiment, the enforced ectopic IRTKS expression restored the heterochromatin distribution in the KO MEFs (Figs. 1E,F and EV1F). Subsequently, we knocked out IRTKS in human liver endothelial (SK-Hep-1) cells using the CRISPR/Cas9 system, and a significant decrease in heterochromatin domains was also observed (Fig. EV1G–I). Conversely, overexpression of ectopic IRTKS distinctly increased the EDRs of heterochromatin, particularly at the nuclear periphery, in MEFs and SK-Hep-1 cells (Fig. EV1J–O). These results suggested that IRTKS could be required for heterochromatin formation in mouse and human cells.

To strengthen this hypothesis, we examined the subcellular co-localization of IRTKS with the typical heterochromatin markers H3K9me3 and HP1α using an immunofluorescence (IF) assay. We found that Irtks co-localized with H3K9me3 and HP1α foci in MEFs, and interestingly, as Irtks was lost, the puncta and signal intensity of H3K9me3 and HP1α were dramatically reduced in *Irtks* KO MEFs (Fig. 1G,H). Similarly, IF imaging also showed that the foci of HP1α became diffuse or disappeared in the livers, kidneys, and stomachs of *Irtks* KO mice (Figs. 1I,K and EV1P), which was consistent with the decreased level of HP1α foci by the quantification of the line scan analysis and western blotting assay (Figs. 1J,L and EV1Q–S). Moreover, the puncta and signal intensity of H3K9me3 were also obviously reduced in livers of *Irtks* KO mice (Fig. EV1T). Furthermore, we performed live-cell fluorescence imaging and fluorescence recovery after photobleaching (FRAP) experiments using enhanced green fluorescent protein (EGFP)-tagged HP1α. We showed that HP1α puncta were rapidly exchanged in the nucleus of SK-Hep-1 and MEF cells (Fig. EV1U–X). However, the recovery of HP1α in *IRTKS* KO SK-Hep-1 cells and MEF cells was dramatically slower than that in the control cells (Fig. EV1V,X and Movies EV1–4), implying that HP1α participates in IRTKS-associated heterochromatin formation. Collectively, our data indicate that IRTKS is required for heterochromatin architecture, a requirement that could be related to HP1α, a key molecule for heterochromatin formation.

### IRTKS promotes HP1α SUMOylation with larger condensates by recruiting Ubc9

To explore the mechanism by which IRTKS stabilizes heterochromatin organization, we examined heterochromatin-associated enzymes and factors in livers of *Irtks* KO mice by western blot assay and found that only HP1α was visibly reduced in the absence of Irtks (Fig. EV2A), suggesting that IRTKS is required for maintaining the levels of HP1α, an essential factor for heterochromatin formation. Subsequently, we knocked out *HP1α* or *IRTKS* in SK-Hep-1 cells using the CRISPR/Cas9 system (Fig. EV2B,C) and then observed the effect of ectopic IRTKS or HP1α on heterochromatin formation. In the HP1α-deficient cells, heterochromatin at the nuclear periphery failed to be restored by ectopic IRTKS expression (Fig. 2A). Conversely, in the IRTKS-deficient cells,

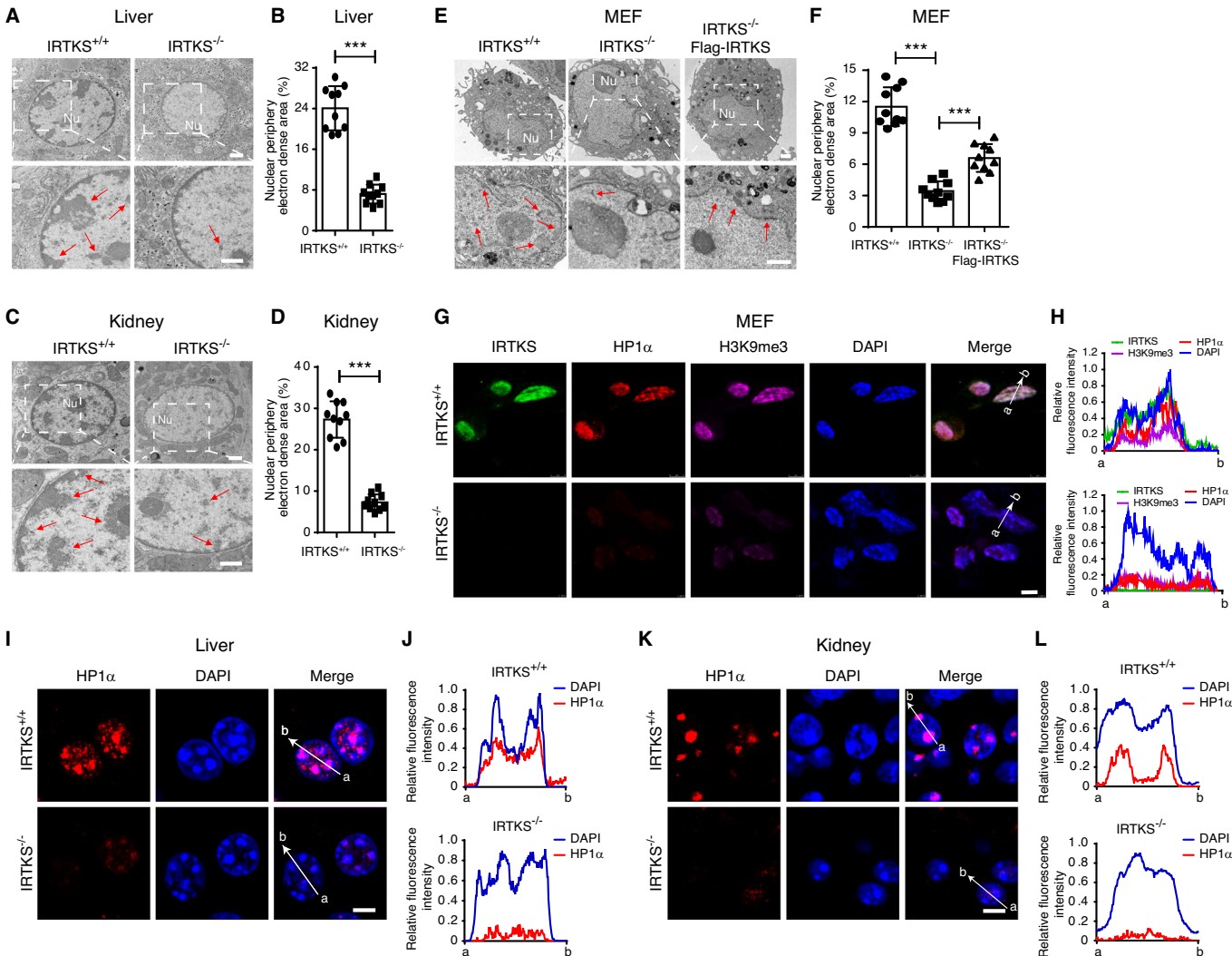

**Figure 1. IRTKS is required for heterochromatin formation.**

(A–D) Representative images of electron microscopy and statistical analysis of the electron-dense areas of the heterochromatin regions at the nuclear periphery in the liver (**A, B**) and kidney (**C, D**) tissues from WT and *Irtks* KO mice. Parts of the upper panel were enlarged and are shown in the lower panel. Red arrows indicate the electron-dense heterochromatin regions. Nu, nucleolus. $n = 10$ cells analyzed for each condition (**B**, ***$p = 8.3836 \times 10^{-8}$; **D**, ***$p = 1.17 \times 10^{-8}$). Scale bar, 1 µm. (**E**) Ectopically expressed IRTKS can rescue heterochromatin at the nuclear periphery of *Irtks* KO MEFs. Red arrows indicate the electron-dense heterochromatin regions. Nu, nucleolus. Scale bar, 1 µm. (**F**) Quantification of the electron-dense heterochromatin regions in MEFs. $n = 10$ cells analyzed for each condition. ***$p = 7.34 \times 10^{-13}$ (IRTKS$^{+/+}$ *vs* IRTKS$^{-/-}$) and $3.36 \times 10^{-5}$ (IRTKS$^{-/-}$ *vs* IRTKS$^{-/-}$-Flag-IRTKS). (**G, H**) Immunofluorescence images (**G**) of MEFs show that IRTKS (green) co-localizes with HP1α (red) and H3K9me3 (purple). Nuclei were counterstained with DAPI (blue). (**H**) Line scans of the images of a cell co-stained for IRTKS, HP1α, H3K9me3, and DAPI at the position depicted by the white arrow. Scale bar, 5 µm. (**I–L**) Representative confocal images of HP1α foci (red) and nuclei (DAPI, blue) in the livers (**I**) and kidneys (**K**) of WT and *Irtks* KO mice. Quantification of lines scanned across HP1α foci and nuclei at the position depicted by the white arrow (**J, L**). Scale bar, 5 µm. Data are presented as the mean ± SD. Figure 1F was tested by one-way ANOVA followed by Tukey's post hoc test. The remaining plots were tested by two-tailed Student's t test. Source data are available online for this figure.

heterochromatin at the nuclear periphery could be partially restored by ectopic HP1α expression (Fig. EV2D), implying that HP1α is indispensable for IRTKS-mediated heterochromatin formation.

Next, we sought to examine whether there is a direct interaction between IRTKS and HP1α through a GST pull-down assay and reciprocal co-immunoprecipitation (co-IP) assay in human HEK293T and MEF cells, showing that IRTKS can reciprocally and directly interact with HP1α (Fig. EV2E–H). To determine which domains are responsible for the interactions between IRTKS

and HP1α, a series of truncations derived from the two proteins were used to generate GST-tagged fusion proteins (Fig. EV2I). GST pull-down assays showed that the N-terminal CD and C-terminal CSD domains of HP1α can mediate the physical interaction with IRTKS, while the C-terminal SH3 and WH2 domains of IRTKS can directly bind to HP1α (Fig. EV2J,K). Subsequently, based on the 3D structures of IRTKS and HP1α predicted by AlphaFold algorithms (Tunyasuvunakool et al, 2021), we analyzed the interaction interface of the IRTKS-HP1α in detail using the Z-DOCK server. We found that key residues such as Gln 34, Asp 131, and Tyr 177

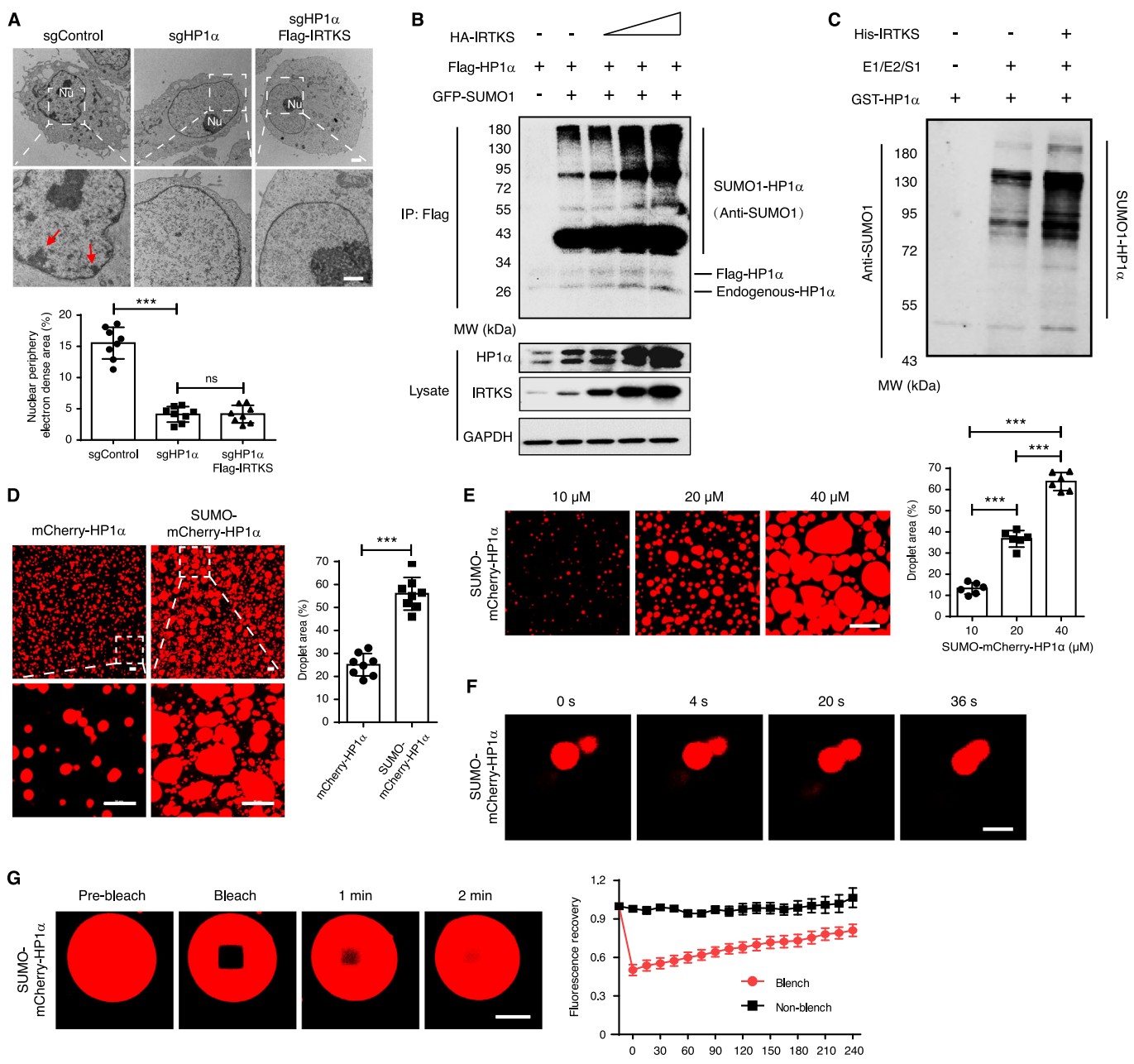

**Figure 2. IRTKS recruits Ubc9 to SUMOylate HP1α.**

(A) Electron microscopy images and quantification of the electron-dense heterochromatin regions in SK-Hep-1 cells treated with CRISPR/Cas9 single-guide RNA (sgRNA) lentivirus (sgHP1α) to knock out HP1α coupled with the Flag-IRTKS construct. Red arrows indicate the electron-dense heterochromatin regions. Nu, nucleolus. $n = 8$ cells analyzed for each condition. ***$p = 3.39 \times 10^{-11}$, ns $= 0.967612$. Scale bar, 1 μm. (B) HP1α was abundantly SUMOylated by SUMO-1 when co-expressed with IRTKS in HEK293T cells, as detected by immunoprecipitation assay. HEK293T cells transfected with HA-IRTKS, Flag-HP1α, and GFP-SUMO1 were immunoprecipitated with an anti-Flag antibody for the SUMOylation assay, followed by western blotting with the indicated antibodies. (C) In vitro SUMOylation assay showing that IRTKS visibly enhanced HP1α SUMOylation in the presence of SUMO E1, E2 and SUMO-1 proteins. (D) Droplet formation assays showing that SUMO-mCherry-HP1α forms liquid-like droplets. mCherry-HP1α or SUMO-mCherry-HP1α was added to the droplet formation buffer to 40 μM. Scale bars, 10 μm. $n = 8$ fields for each group were quantified. ***$p = 2.39 \times 10^{-7}$. (E) Representative images and quantification of droplet formation at various protein concentrations ($n = 6$ fields for each group were quantified). ***$p = 1.34 \times 10^{-8}$ (10 μM vs 20 μM), $2.58 \times 10^{-13}$ (10 μM vs 40 μM), and $1.96 \times 10^{-9}$ (20 μM vs 40 μM). SUMO-mCherry-HP1α was added to droplet formation buffer to final concentrations as indicated. Scale bar, 10 μm. (F) Time-lapse fluorescence images showing that the droplets of SUMO-mCherry-HP1α rapidly fused. Scale bar, 2 μm. (G) Representative images and the fluorescence recovery curve of the SUMO-mCherry-HP1α FRAP experiments. Scale bar, 10 μm. $n = 8$ biological replicates for the FRAP curve construction. Data are presented as the mean ± SD. Figure 2D was tested by two-tailed Student's t test. The remaining plots were tested by one-way ANOVA followed by Tukey's post hoc test. Source data are available online for this figure.

on the CD and CSD domains of HP1α are involved in the interaction with IRTKS (Fig. EV2L), which is consistent with the results of the GST pull-down experiments. Taken together, these results suggest that IRTKS-mediated heterochromatin formation occurs via direct regulation of HP1α.

To explore the molecular mechanism by which IRTKS regulates the HP1α level, we first assessed HP1α at the transcriptional level by quantitative RT-PCR. The data showed that the abundance of HP1α mRNA remained unchanged in the livers, kidneys and MEFs of *Irtks* KO mice (Fig. EV3A), excluding the possibility of transcriptional regulation. We therefore further speculated that IRTKS could positively modulate HP1α stability via posttranslational modifications. To examine this hypothesis, we evaluated HP1α stability in HEK293T cells with or without transfected ectopic IRTKS with the addition of the protein synthesis inhibitor cycloheximide (CHX). In the presence of CHX, HP1α in HEK293T cells overexpressing IRTKS maintained the same level, while HP1α was clearly decreased in control cells without ectopic IRTKS (Fig. EV3B), suggesting that IRTKS enhances HP1α stability.

To further determine how IRTKS enhances HP1α stability, we first blocked the ubiquitin–proteasome pathway with MG132, a proteasome inhibitor, in *Irtks* KO MEFs. However, treatment with MG132 failed to restore HP1α levels (Fig. EV3C), implying that the lower HP1α levels in *Irtks*-deficient cells could not be ascribed to the ubiquitin–proteasome pathway but is perhaps due to other mechanisms mediated by IRTKS. Several previous studies have demonstrated that the SUMOylation of HP1α has been considered crucial for binding to pericentric heterochromatin (Maison et al, 2011; Maison et al, 2016b). In addition, our previous study demonstrated that IRTKS can recruit Ubc9, the only known E2 ligase involved in protein SUMOylation, to SUMOylate PCBP2 (Xia et al, 2015), implying that IRTKS could act as a scaffold for SUMOylation of other protein substances by recruiting Ubc9. To test this possibility, we first confirmed the direct interaction between IRTKS and Ubc9 through a GST pull-down assay and co-IP experiments in human HEK293T cells (Fig. EV3D,E). These results were consistent with our previous study (Xia et al, 2015).

Next, to verify that IRTKS enhances HP1α stability through SUMOylation, both HA-tagged IRTKS and Flag-tagged HP1α, together with GFP-tagged SUMO1, were transiently co-transfected into HEK293T cells. Subsequent immunoblotting results showed that HP1α could be SUMOylated by SUMO-1, and ectopic IRTKS overexpression facilitated SUMOylation of HP1α, leading to an increase of HP1α level in a dose-dependent manner (Fig. 2B). Since HP1α SUMOylation was usually detected in the embryonic mouse fibroblast cell line NIH3T3 (Maison et al, 2016a; Maison et al, 2011; Maison et al, 2012), we further conducted an in vivo SUMOylation experiment in NIH3T3 cells, demonstrating that IRTKS obviously enhanced the SUMO1-mediated SUMOylation of HP1α in NIH3T3 cells (Fig. EV3F). Consistent with that result, an in vitro reconstitution assay also confirmed that SUMOylation of HP1α was distinctly enhanced by IRTKS (Figs. 2C and EV3G,H). Together, these data suggest that IRTKS can enhance HP1α SUMOylation.

Considering that both unmodified and phosphorylated HP1α have been proven to be involved in heterochromatin formation by LLPS (Larson et al, 2017; Li et al, 2020; Strom et al, 2017; Wang et al, 2019), we assessed whether SUMOylated HP1α also has the ability to drive LLPS. We first purified the SUMOylated His$_6$-mCherry-tagged HP1α fusion protein (SUMO-mCherry-HP1α), which was obtained by co-expressing His$_6$-mCherry-HP1α with the E1E2SUMO1 plasmid. SUMOylation of HP1α was confirmed by western blotting and mass spectrometry (Fig. EV3I,J). Interestingly, SUMO-mCherry-HP1α showed the formation of liquid-like condensates under crowding conditions, as shown by confocal imaging, with the size of SUMO-mCherry-HP1α droplets being markedly increased compared with that of unmodified mCherry-HP1α condensates (Fig. 2D), implying that HP1α−driven phase separation could be promoted by IRTKS-Ubc9-mediated SUMOylation.

To assess whether SUMO-mCherry-HP1α maintains liquid-like properties, we performed confocal imaging, showing a significant formation of SUMO-mCherry-HP1α droplets with a size increase in a dosage-dependent manner (Fig. 2E). Moreover, with increasing salt concentration or 1,6-hexanediol treatment, the formation of SUMO-mCherry-HP1α significantly decreased (Fig. EV3K,L). In addition, some SUMO-mCherry-HP1α droplets that were close to each other fused into larger condensates over time (Fig. 2F, Movie EV5). A FRAP experiment showed that SUMO-mCherry-HP1α droplets could be recovered after photobleaching (Fig. 2G, Movie EV6), which confirmed the dynamic characteristic of these liquid droplets. Taken together, these data demonstrate that IRTKS recruits Ubc9 to SUMOylate and stabilize HP1α, and SUMOylated HP1α possesses stronger phase-separation properties.

## IRTKS undergoes liquid–liquid phase separation

It is known that, among a limited number of recognized driving factors, the HP1α-mediated liquid droplet formation has been linked to heterochromatin formation (Larson et al, 2017; Sanulli et al, 2019; Strom et al, 2017; Zhao et al, 2019), and thus we were intrigued to investigate whether IRTKS-driven heterochromatin formation also undergoes liquid−liquid phase separation (LLPS). To investigate whether IRTKS has liquid-like properties on its own, we first analyzed vertebrate IRTKS sequences using D$^2$P$^2$ and PONDR algorithms, which were designed to predict intrinsically disordered regions (IDRs), the fragments known to be involved in LLPS, within proteins (Alberti et al, 2019). The analysis showed that IRTKS has some potential IDRs (Figs. 3A and EV4A), suggesting the possibility that IRTKS can undergo phase separation.

Next, to determine the ability of IRTKS to form phase-separated droplets, we purified recombinant His$_6$-EGFP-tagged IRTKS fusion protein (EGFP-IRTKS) and His$_6$-EGFP (EGFP) as a control (Fig. EV4B) and then mixed them with buffers containing 10% polyethylene glycol (PEG) 8000 (a molecular crowding agent). As expected, EGFP-IRTKS exhibited droplet-like morphology at room temperature in vitro, whereas EGFP remained diffuse under the same test conditions (Fig. 3B). His$_6$-mCherry and recombinant His$_6$-mCherry-HP1α fusion proteins were also purified and mixed with phase-separated buffer, and we observed that mCherry-HP1α formed droplets as well (Fig. EV4C,D), which was consistent with previous findings (Li et al, 2020; Wang et al, 2019).

To exclude the possibility that the droplet formation of EGFP-IRTKS was induced by the multivalent effect of EGFP, we purified His$_6$-IRTKS protein without EGFP (Fig. EV4E) and then performed a droplet formation assay under the same experimental conditions. Numerous liquid-like droplets were still observed (Fig. EV4F), suggesting that IRTKS has the ability to form liquid droplets on its

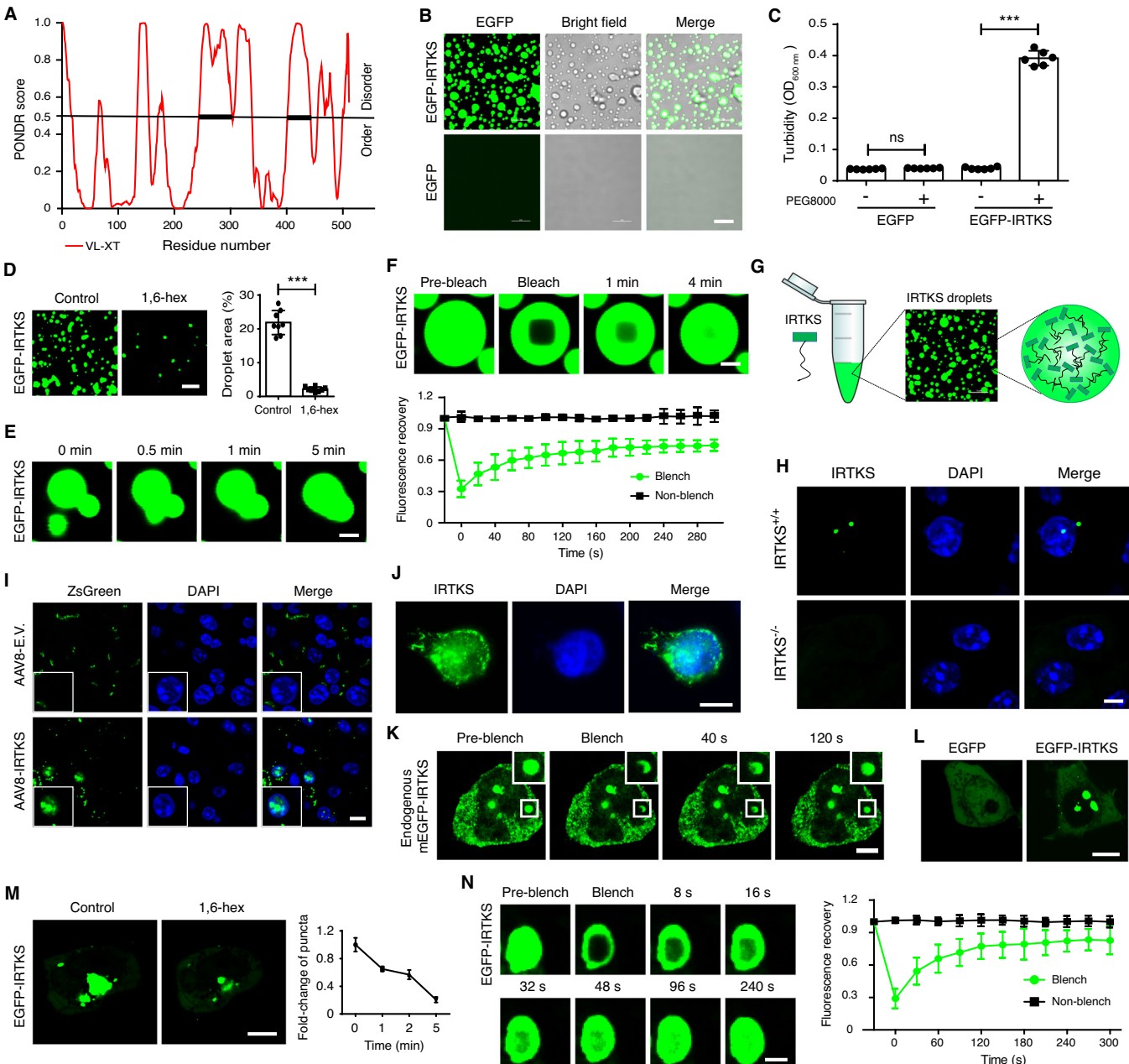

own. Here we found that the size and number of EGFP-IRTKS droplets significantly increased with concentration (Fig. EV4G). In addition, we observed that when the crowding agent 10% PEG-8000 was added to the buffer, the EGFP-IRTKS solution rapidly became turbid, whereas equivalent solutions with EGFP alone remained clear (Figs. 3C and EV4H). Moreover, we found that the ability of EGFP-IRTKS droplets to form was significantly reduced with increasing salt concentrations or the addition of 5% 1,6-hexanediol (Figs. 3D and EV4I). In addition, EGFP-IRTKS displayed other phase-separated liquid properties, including time-dependent droplet fusion and dynamic molecular exchange of droplets as measured using the FRAP assay (Figs. 3E,F and Movies EV7, 8). Taken together, these results indicate that the liquid–liquid phase separation of IRTKS can occur in vitro.

To verify which domains or fragments within IRTKS are responsible for phase separation, we purified three EGFP-tagged IRTKS fragments containing I-BAR, SH3, and WH2 domains and then tested their ability to form droplets by using an in vitro phase-separation assay. Our data showed that the purified SH3 and WH2 domains containing larger IDRs, called IRTKS-IDR, underwent phase separation (Fig. EV4J,K), consistent with the prediction of the $D^2P^2$ and PONDR algorithms (Figs. 3A and EV4A). Based on the above data, we infer that the IRTKS-mediated phase-separated droplets could form due to the interactions among these IRTKS-IDRs (Fig. 3G), which is in agreement with the widely accepted idea that IDRs are sufficient to drive the formation of phase separation though homotypic interactions.(Alberti et al, 2019; Shapiro et al, 2021; Tsang et al, 2020; Zhang et al, 2020a).

**Figure 3. IRTKS undergoes LLPS in vivo and in vitro.**

(A) Prediction of disordered regions by PONDR algorithms. (B) Representative images of droplet formation showing that EGFP-IRTKS has liquid-like properties. EGFP served as a negative control. EGFP-IRTKS and EGFP were added to droplet formation buffer to 20 μM. Scale bar, 5 μm. (C) Turbidity assay demonstrating phase separation of IRTKS. $n = 6$ (two independent experiments, each with three technical repeats). ns $= 0.7066$, ***$p = 1.5 \times 10^{-22}$. (D) Representative images and quantification of the droplet formation ability of EGFP-IRTKS in the absence or presence of 5% 1,6-hexanediol ($n = 8$ fields for each group were quantified). EGFP-IRTKS was added to droplet formation buffer to 10 μM. ***$p = 6.19 \times 10^{-7}$. Scale bar, 5 μm. (E) Time-lapse fluorescence images showing that the EGFP-IRTKS droplets rapidly fused. Scale bar, 1 μm. (F) Representative FRAP images showing that the EGFP-IRTKS signal within the puncta recovered within a few minutes. The fluorescence recovery curve of the EGFP-IRTKS FRAP experiments ($n = 8$ biological replicates for the FRAP curve construction). Scale bar, 1 μm. (G) A model showing that the interactions among IDRs (black tails) of IRTKS mediate the formation of phase-separated liquid droplets in vitro. (H) Immunofluorescent staining of IRTKS (green) and DAPI (blue) in liver sections of WT and *Irtks* KO mice. Scale bar, 5 μm. (I) Immunofluorescent staining of IRTKS (green) and DAPI (blue) in liver sections of mice infected with empty vector AAV8 (AAV-E.V.) (control) or AAV8-IRTKS. Scale bar, 5 μm. (J) Immunofluorescence imaging of endogenously mEGFP-tagged IRTKS (green) and DAPI (blue) in HEK293T cells. Scale bar, 10 μm. (K) Representative live-cell images of the endogenous mEGFP-tagged IRTKS puncta shown by FRAP experiments in HEK293T cells. Scale bar, 5 μm. (L) Representative images of live cell imaging of EGFP (control) or EGFP-IRTKS-expressing HEK293T cells. Scale bar, 10 μm. (M) Representative live-cell images and quantification of EGFP-IRTKS-expressing HEK293T cells before and after treatment with 5% 1,6-hexanediol ($n = 3$ biological replicates for each group). Scale bar, 10 μm. (N) Representative live-cell images and the fluorescence recovery curve of the EGFP-IRTKS FRAP experiments ($n = 8$ biological replicates for the FRAP curve construction). Scale bar, 2 μm. Data are presented as the mean ± SD or mean ± SEM. Figure 3C was tested by one-way ANOVA followed by Tukey's post hoc test. Figure 3D was tested by two-tailed Student's t test. Source data are available online for this figure.

To further determine which residues of IRTKS-IDR are required for LLPS, we first examined the amino acid content within IRTKS-IDR, finding that IRTKS-IDR has an abundance of serine, proline, and tyrosine (Fig. EV4L). To test the role of these copious residues, we replaced these residues of IRTKS-IDR with alanine and then predicted the impact of these mutants on the ability to form phase-separated droplets using PONDR algorithms. We found that the P-to-A mutations dramatically destroyed the IRTKS-IDR, whereas the T-to-A and S-to-A mutations had less effect on the IRTKS-IDR (Fig. EV4M). In accordance with the prediction of the PONDR algorithms, the in vitro phase separation assay confirmed that the P-to-A mutant of IRTKS-IDR was unable to form liquid-like puncta, but the wild-type IRTKS-IDR quickly formed droplets under the same experimental conditions (Fig. EV4N). These results suggest that the proline residues within IRTKS-IDR are indispensable for phase-separated droplet formation.

To determine whether endogenous IRTKS indeed forms condensates in vivo, we first observed livers and kidneys from WT mice by IF with an anti-IRTKS antibody. We observed liquid-like puncta of endogenous Irtks in the cytoplasm and nucleus of these tissues from WT mice in comparison to those from *Irtks* KO mice (Figs. 3H and EV4O–Q). Furthermore, we injected mice with adeno-associated virus serotype 8 (AAV8) vector containing ZsGreen-tagged IRTKS (AAV8-IRTKS) to confirm whether IRTKS can form condensates in vivo. Interestingly, the conditionally overexpressed IRTKS in mouse livers exhibited liquid-like puncta, whereas the empty AAV8 vector only expressing ZsGreen did not result in liquid-like puncta (Figs. 3I and EV4R).

In the subsequent experiment, we aimed to investigate the presence of IRTKS puncta in human cells. We engineered HEK293T cells by using the CRISPR–Cas9 system to label endogenous IRTKS with monomeric enhanced green fluorescent protein (mEGFP) (Fig. EV4S,T). Confocal fluorescence imaging showed that the mEGFP-tagged endogenous IRTKS formed condensates within the cytoplasm or nucleus, and these endogenous mEGFP-IRTKS puncta exhibited dynamic fusion and splitting behaviors (Figs. 3J and EV4U,V, Movies EV9, 10). Furthermore, the FRAP experiment showed that the endogenous mEGFP-IRTKS puncta could recover after photobleaching (Figs. 3K and EV4W, Movie EV11), according with the phase-separated liquid properties.

Together, these results demonstrated that endogenous IRTKS can form phase-separated condensates.

To further assess the liquid-like characteristics of IRTKS puncta in vivo, EGFP-tagged IRTKS was transfected into HEK293T cells. We observed that EGFP-IRTKS formed puncta, particularly in the nucleus, as shown by confocal microscopy (Fig. 3L), whereas the empty vector expressing EGFP failed to lead to formation of puncta, ruling out the possibility that the puncta were artificially formed by the EGFP tag. Then, HEK293T cells expressing EGFP-IRTKS were treated with 1,6-hexanediol, and the sizes and numbers of EGFP-IRTKS puncta significantly decreased (Fig. 3M, Movie EV12). Furthermore, time-lapse fluorescence images showed that EGFP-IRTKS condensates underwent fusion over time (Fig. EV4X, Movie EV13). In addition, a FRAP experiment indicated that such EGFP-IRTKS puncta displayed the dynamic features of liquid droplets (Fig. 3N, Movie EV14), in accordance with phase-separated behavior. Taken together, these data strongly support the idea that IRTKS undergoes liquid-liquid phase separation, and the formed condensates display phase-separated properties.

## IRTKS droplets infiltrate the phase-separated heterochromatin compartment

Numerous studies have shown that heterochromatin can exist as a dynamic phase-separated condensate (Erdel et al, 2020; Li et al, 2020; Sanulli et al, 2019; Zhang et al, 2022). Based on our findings that IRTKS is crucial for heterochromatin formation and has the capacity to form phase-separated condensates, we speculated that IRTKS could function as a key component of heterochromatin condensates. To examine this hypothesis, we performed droplet assays in vitro to detect whether IRTKS droplets could be incorporated into heterochromatin condensates composed of HP1α, nucleosomal arrays and various DNAs. We first assessed the ability of EGFP-IRTKS droplets to be concentrated and incorporated into mCherry-HP1α condensates. Notably, the phase-separated droplets formed by mCherry-HP1α were visibly enhanced by EGFP-IRTKS in a dosage-dependent manner, and vice versa (Fig. EV5A), indicating that IRTKS can incorporate into and promote the phase separation of HP1α, a known key component for heterochromatin formation.

A similar reciprocal incorporation between EGFP-IRTKS and SUMO-mcherry-HP1α droplets was also observed, along with an increase in both size and number of these co-condensates in a manner that depended on the dosage of the two proteins (Fig. EV5B), which was consistent with the above finding that SUMOylated HP1α possesses stronger phase-separated properties (Fig. 2D). Next, we observed the incorporation of EGFP-IRTKS droplets into nucleosomal arrays and various DNAs. Interestingly, the incorporation into Cy5-labeled DNA, nucleosomal DNA, twelve native nucleosomal arrays or H3K9me3-marked nucleosomal arrays (Figs. 4A–D and EV5C–F) was also markedly promoted by EGFP-IRTKS in a dosage-dependent fashion. More notably, the condensates composed of EGFP-IRTKS and mCherry-HP1α droplets could be further infiltrated by diverse DNAs and nucleosomal arrays, including Cy5-labeled DNA, nucleosomal DNA or H3K9me3-marked nucleosomal arrays (Figs. 4E,F and EV5G–J), implying that IRTKS, HP1α and various DNA or H3K9me3-marked nucleosomal arrays can be dramatically incorporated into heterochromatin condensates (Fig. 4G). Overall, these observations suggest that IRTKS can participate in heterochromatin-associated phase separation.

Subsequently, we sought to investigate whether the ability of IRTKS to form condensates is involved in heterochromatin formation. Accordingly, we transfected the enhanced green fluorescent protein (EGFP)-tagged IRTKS into mouse fibroblast NIH3T3 cells and human liver endothelial SK-Hep-1 cells, and the IRTKS condensates were directly visualized using live-cell fluorescence microscopy. We observed that EGFP-IRTKS, but not EGFP alone, predominantly localized in the nucleus exhibited significant overlap with Hoechst-dense regions in NIH3T3 cells, as indicated by the line scan analysis (Fig. EV5K,L). Similarly, IRTKS puncta were observed in both the cytoplasm and nucleus of SK-Hep-1 cells, showing co-localization with the intensely Hoechst-stained regions (Fig. EV5M,N). Furthermore, the EGFP-tagged IRTKS truncations containing I-BAR, IDR, and IDR-mutant (P to A mutant of IRTKS-IDR) were overexpressed in NIH3T3 cells. Compared to full-length IRTKS, the IDR of IRTKS was also capable of forming puncta and co-localized effectively with the Hoechst signals, as shown by the line scan analysis in NIH3T3 cells. In contrast, I-BAR and IDR-mutant of IRTKS were dispersed without the ability to form puncta (Fig. EV5K,L). Collectively, these results suggest that IRTKS condensates are associated with heterochromatin organization.

To further confirm the impact of IRTKS condensates on heterochromatin formation, the EGFP-tagged IRTKS with various truncations (full-length, I-BAR, IDR, and IDR-mutant) or EGFP control were overexpressed in MEF cells lacking the endogenous IRTKS. It was observed that re-introduction of IRTKS-full length and IDR mutant markedly restored the distribution of heterochromatin at the nuclear periphery and around the nucleolus, as visualized by transmission electron microscopy (Fig. EV5O,P), while IRTKS-I-BAR only slightly rescued the EDRs of heterochromatin. In contrast, LLPS-defective IRTKS-IDR-mutant and EGFP control failed to restore the heterochromatin distribution in *Irtks* KO MEFs (Fig. EV5O,P).

In addition, we performed chromatin immunoprecipitation-quantitative polymerase chain reaction (ChIP-qPCR) to investigate the effect of IRTKS condensates on regulating heterochromatin-associated repetitive sequences, showing that the occupation of H3K9me3 and HP1α (heterochromatin markers) on known

genomic repetitive sequence subtypes, including SINE, LINE1 and intracisternal A particle (IAP) (an LTR retrotransposon), was significant decreased in MEFs from *Irtks* KO mice, as compared to those from WT mice (Fig. EV5Q). Notably, in IRTKS-KO MEF cells, the ectopic expression of IRTKS-full length and IRTKS-IDR, but not IRTKS-IDR-mutant and EGFP control, substantially increased the enrichment of H3K9me3 and HP1α on repetitive sequences (Fig. EV5Q). Taken together, these findings strongly support the critical role of LLPS-forming ability of IRTKS in the regulation of heterochromatin.

## IRTKS deficiency leads to genome-wide epigenetic alterations

Given that heterochromatin is characterized by abundant repetitive sequences that are commonly depressed (Liang et al, 2022b), we analyzed various repetitive sequences for occupation by H3K9me3 and HP1α, the known typical markers of heterochromatin, through chromatin immunoprecipitation sequencing (ChIP-seq) of livers and MEFs from WT and *Irtks* KO mice. As expected, the enrichment of H3K9me3 and HP1α on REs was visibly decreased in livers and MEFs (Fig. 5A,B; Appendix Fig. S1A,B) from *Irtks* KO mice compared with those from WT mice. ChIP-qPCR analysis further confirmed the significant reduction in H3K9me3 and HP1α bound at diverse repetitive element (RE) in these mouse livers and MEFs (Fig. 5C,D; Appendix Fig. S1C,D) without endogenous Irtks. Interestingly, in the absence of IRTKS, the enrichment of H3K9me3 and HP1α is obviously reduced at the same chromatin binding sites across the genome in mouse livers and MEFs (Appendix Fig. S1E,F), especially at repetitive sequences, including simple DNA repeats, LINE and LTR retrotransposons (Fig. 5E; Appendix Fig. S1G). Together, these findings demonstrate that IRTKS is crucial to the enrichment of H3K9me3 and HP1α on REs.

In addition, we further analyzed the enrichment of H3K9me3 and HP1α on diverse non-repetitive sequences (non-REs), including transcription termination site (TTS), intergenic, intron, exon and untranslated regions of mRNA, showing that the enrichment of H3K9me3 on most of non-RE regions was significantly decreased in livers and MEFs from *Irtks* KO mice, as compared with those from WT mice (Appendix Fig. S1H–K), suggesting that IRTKS deficiency could affect nearly all chromatin regions, encompassing both heterochromatic and euchromatic regions.

We further assessed the genome-wide changes in IRTKS-mediated chromatin accessibility by using transposase-accessible chromatin sequencing (ATAC-seq) analysis. As expected, ATAC-seq analysis demonstrated that chromatin accessibility showed no visible changes at *Actb* locus as a negative control (Appendix Fig. S1L,M), but significantly increased in repetitive DNA sequence regions, including SINEs, LINEs, LTRs, transposable elements and simple DNA repeats, in *Irtks*-deficient MEFs and livers, as compared to those in WT controls (Fig. 5F; Appendix Fig. S1N,O). These results suggest that in the absence of IRTKS, condensed heterochromatin, especially that harboring regions of repetitive DNA sequences, is dramatically altered to be easily accessible.

In addition, ATAC-seq analysis demonstrated that chromatin accessibility was also significantly increased in most non-RE regions in MEFs and livers from *Irtks* KO mice, as compared with those from WT mice (Appendix Fig. S1P,Q). Consistent with these

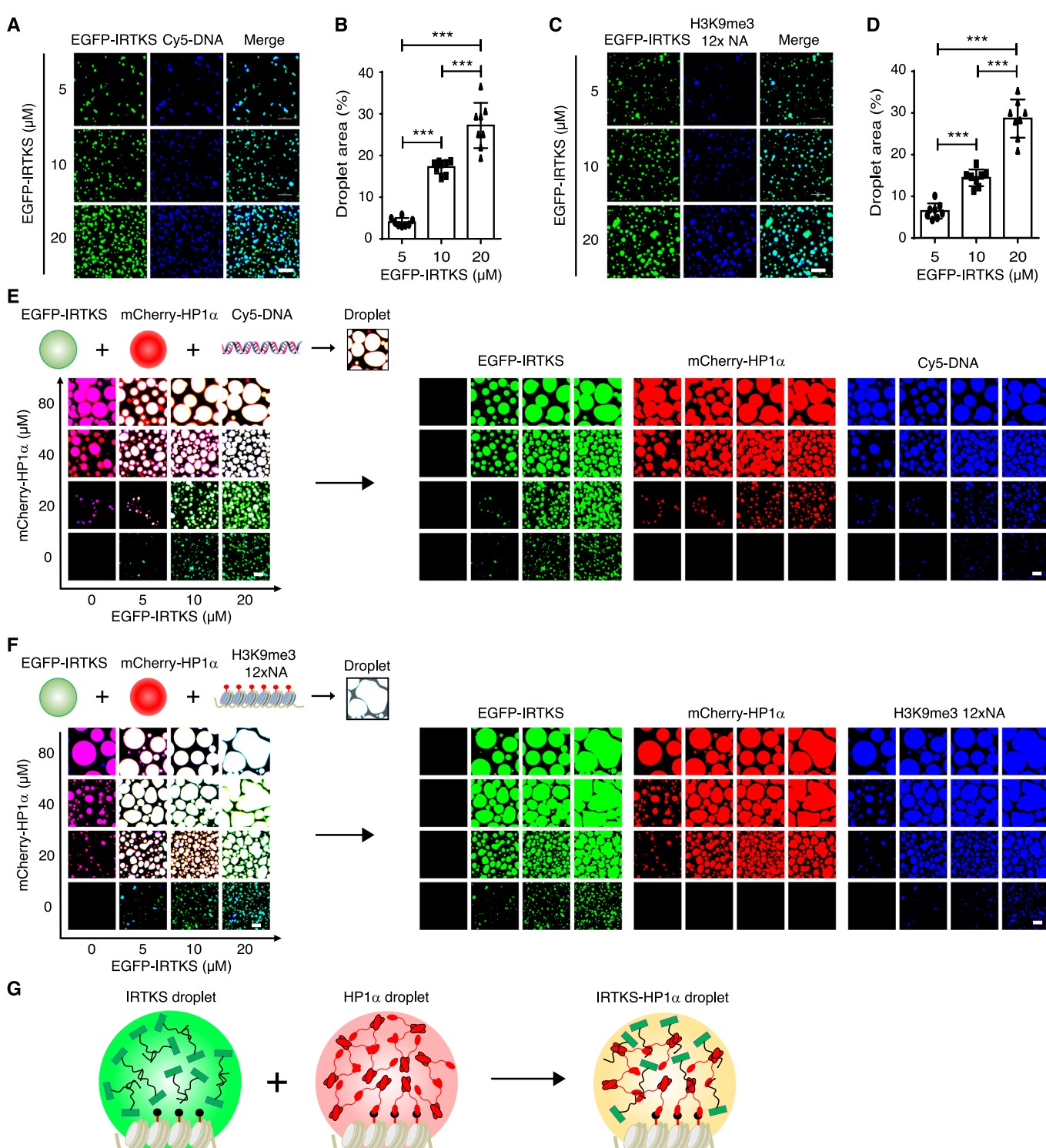

changes in chromatin accessibility, our RNA sequencing (RNA-seq) and RT-qPCR analyses also showed notable increases in the transcript levels of repetitive DNA sequences and non-REs in the livers and MEFs from *Irtks* KO mice in comparison to those from WT mice (Fig. 5G; Appendix Fig. S1R). Collectively, these results indicate that IRTKS-mediated heterochromatin harbors regions of repetitive DNA sequences and represses their transcriptional activity across the whole genome, which is in accordance with the hallmarks of constitutive heterochromatin.

## IRTKS deficiency accelerates cellular senescence

Heterochromatin loss and elevated transcriptional activity of REs are associated with aging and cell senescence (Gorbunova et al,

**Figure 4. IRTKS participates in heterochromatin-associated phase separation.**

(A, B) Liquid–liquid phase separation assay of EGFP-IRTKS proteins at different concentrations mixed with DNA oligos. A total of 160 nM fluorescent DNA for the droplet assay was labeled with Cy5 fluorophore modifications ($n = 8$ fields for each group were quantified). ***$p = 8.44 \times 10^{-8}$ (5 µM vs 10 µM), $3.96 \times 10^{-12}$ (5 µM vs 20 µM), and $5.56 \times 10^{-6}$ (10 µM vs 20 µM). Scale bar, 5 µm. (C, D) In vitro phase separation assay of EGFP-IRTKS proteins at various concentrations mixed with reconstituted H3K9me3 12× nucleosomal arrays (NA). Reconstituted H3K9me3 12× NA (330 nM) for the droplet assay was stained using DAPI ($n = 8$ fields for each group were quantified). ***$p = 3.97 \times 10^{-5}$ (5 µM vs 10 µM), $2.38 \times 10^{-12}$ (5 µM vs 20 µM), and $7.89 \times 10^{-9}$ (10 µM vs 20 µM). Scale bar, 5 µm. (E) Droplet experiments with DNA oligos examining the ability of IRTKS to form condensates with HP1α. Concentrations of IRTKS and HP1α are indicated at the bottom and left of the images, respectively. Fluorescent naked DNA for droplet assays was labeled with a Cy5 fluorophore modification. Scale bars, 5 µm. (F) Liquid–liquid phase separation assay with reconstituted H3K9me3 12× NA to examine the ability of IRTKS to form condensates with HP1α and reconstituted H3K9me3 12× NA stained using DAPI. Scale bar, 5 µm. (G) A schematic model showing that IRTKS and HP1α droplets can be co-incorporated into the phase-separated heterochromatin condensates with a nucleosomal array. Data are presented as the mean ± SD and the $p$ value of one-way ANOVA followed by Tukey's post hoc test. Source data are available online for this figure.

2021). Interestingly, our above data indicate that IRTKS deficiency may lead to loss of heterochromatin at the nuclear periphery and increased transcriptional activity of various repetitive DNA elements at the genome-wide level, which raises the possibility that IRTKS-mediated heterochromatin could be involved in cellular senescence, a main hallmark of aging. To address this issue, we first analyzed the gene expression patterns in kidneys and livers of *Irtks* KO mice by RNA-seq data. Notably, the differentially expressed genes (DEGs) between WT and *Irtks* KO mice were significantly enriched in gene sets associated with cellular senescence and aging, including DNA damage, inflammation, cell cycle and reactive oxygen species, as shown by Gene Ontology (GO) analysis, in these mouse kidneys and livers (Fig. 6A; Appendix Fig. S2A), suggesting that IRTKS may be involved in the regulation of cellular senescence.

To further explore the role of IRTKS in cellular senescence, we examined Irtks expression in 2- to 12-month-old mice by western blotting analysis. The expression of IRTKS was significantly decreased in the kidneys and livers (Fig. 6B; Appendix Fig. S2B) of 12-month-old mice compared to those of 2-month-old mice, indicating that Irtks expression is gradually depressed during the normal aging process. Furthermore, increased senescence-associated-β-galactosidase (SA-β-gal) activity, shortened telomere length and upregulation of the cell cycle arrest-related molecules P16 and P21 were detected in kidneys, livers, and MEFs of *Irtks* KO mice (Fig. 6C–F; Appendix Fig. S2C–I). In addition, we found that lamin B1, a senescence-associated biomarker, was significantly decreased in kidneys and livers of *Irtks*-deficient mice (Fig. 6F; Appendix Fig. S2I), indicating that Irtks deficiency disrupts nuclear integrity, consistent with cell senescence.

Importantly, mounting evidence suggests that senescent cells exhibit loss of nuclear envelope integrity and reactivation of retrotransposons, leading to cytoplasmic translocation of these small DNA and RNA fragments, which can activate the cGAS-STING signaling pathway and trigger type-I interferon (IFN-I) and senescence-associated secretory phenotype (SASP) responses (Miller et al, 2021). As expected, cGAS and STING, two key molecules involved in the pathway, were distinctly elevated in kidneys and livers of IRTKS-deficient mice (Fig. 6F; Appendix Fig. S2I), suggesting that the cGAS–STING pathway is activated in the absence of Irtks. Correspondingly, known SASP-associated genes, including CXCL1, IL-6, TNFα, and IL-1β, were significantly upregulated in the kidneys, livers, and MEFs (Fig. 6G; Appendix Fig. S2J,K) of *Irtks* KO mice compared to those of WT mice.

In addition, we measured the protein products of these crucial factors of SASP in sera from these mice. We found that CXCL1, IL-6,

TNFα, and IL-1β were significantly increased in *Irtks* KO mice compared with those in WT mice (Fig. 6H–K), indicating that IRTKS deficiency indeed leads to the appearance of SASP. Taken together, these results demonstrate that IRTKS deficiency facilitates cellular senescence, possibly through global heterochromatin loss, reactivation of REs through derepression, and disruption of nuclear envelope integrity, which can activate the cGAS-STING pathway to trigger the SASP response (Fig. 6L).

## Discussion

IRTKS, a member of the IRSp53/MIM homology domain family that is well known to play crucial roles in the formation of plasma membrane protrusions (Ahmed et al, 2010; Hu et al, 2000; Millard et al, 2007), is located in the cell membrane, cytoplasm and nucleus under certain conditions. In addition to plasma membrane protrusions, IRTKS also plays roles in regulating the insulin, EGF, FGF, SRC, PIP3, and AKT–mTOR signaling pathways in the cell membrane and cytoplasm (Chen et al, 2011; Wang et al, 2013; Williams et al, 2013). However, the function of IRTKS in the cell nucleus has been largely unknown. Intriguingly, our previous studies demonstrated that IRTKS facilitated MDM2-mediated p53 degradation via the ubiquitin–proteasome pathway (Wang et al, 2011), and SUMOylation of PCBP2 by recruiting the E2 ligase Ubc9 to the cell nucleus. Recently, we also demonstrated that IRTKS may elevate the level of H3K9me3 by promoting the accumulation of SETDB1 (Cui et al, 2023). Consistent with this finding, IRTKS, alongside other known heterochromatin regulators such as HP1α, was identified through mass spectrometry following NeutrAvidin pulldown as an endogenous retrovirus (ERVs)-bound protein engaged in regulating H3K9me3 levels (Zhao et al, 2023).

Inspired by these diverse functions of IRTKS in the cell nucleus (Xia et al, 2015), we set out to explore a non-canonical nuclear role of IRTKS in the regulation of heterochromatin, a fundamental architecture of chromatin. In this study, we provide several lines of evidence unveiling an unexpected role of IRTKS, which undergoes phase separation involving heterochromatin formation and protects cells against senescence. Interestingly, in the new working model, IRTKS is vital for maintaining a higher level of the key protein HP1α to enhance the formation and maintenance of constitutive heterochromatin, with IRTKS recruiting the E2 ligase Ubc9, the only enzyme known to be responsible for SUMOylation, to SUMOylate HP1α, thus stabilizing it. More significantly, IRTKS, based on both in vivo and in vitro experimental evidence, has the capacity for phase separation and concentrates heterochromatin

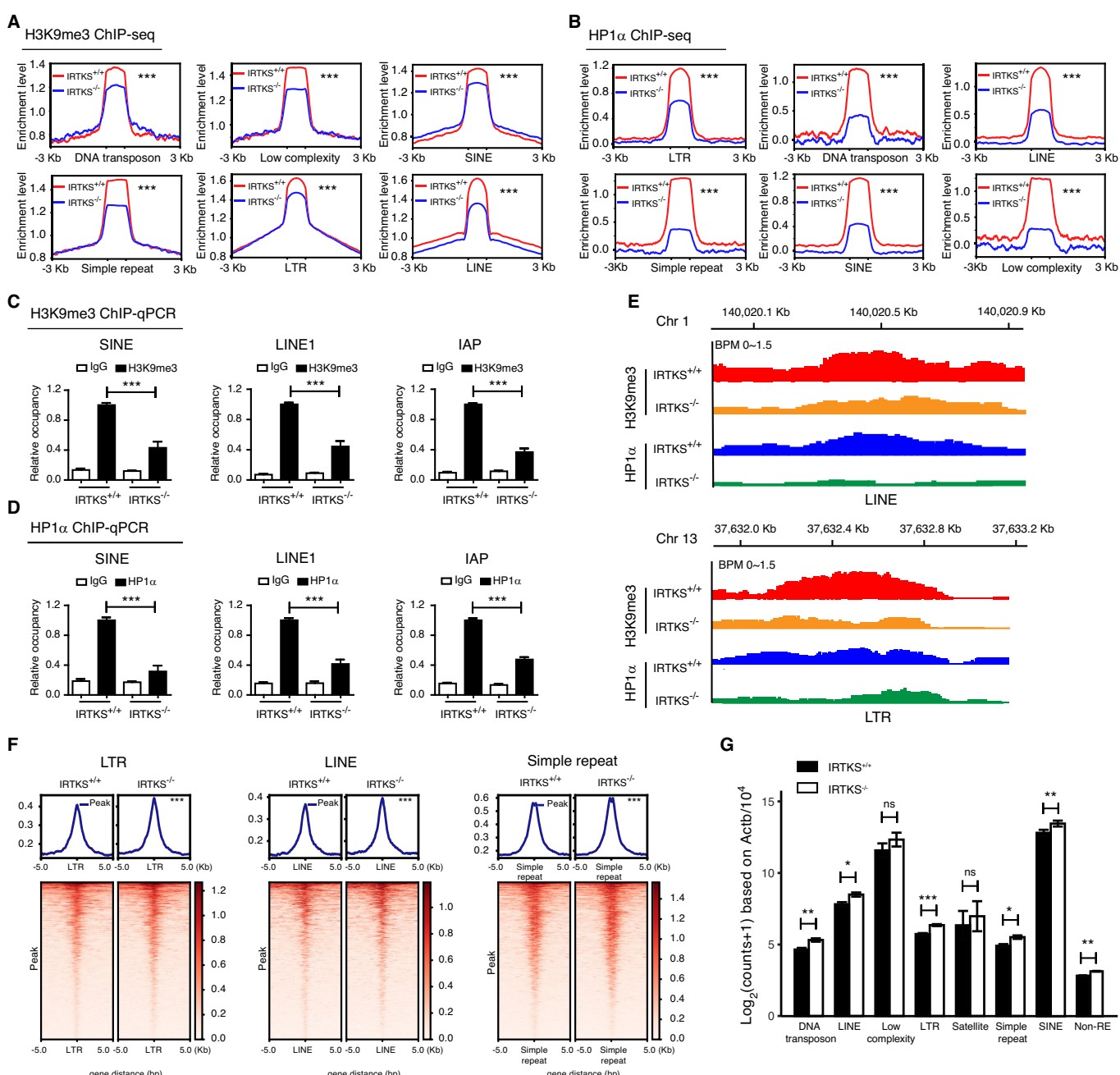

**Figure 5. IRTKS deficiency leads to the genome-wide epigenetic alterations.**

(A, B) ChIP-seq enrichment profiles of H3K9me3 (A) and HP1α (B) peaks showing the reduction of H3K9me3 and HP1α signals at repetitive sequence regions (DNA transposon, low complexity, simple repeat, SINE, LINE, and LTR) in the livers of *Irtks* KO mice. (C, D) Enrichment of H3K9me3 (C, ***$p = 7.12 \times 10^{-5}$ (SINE), $3.52 \times 10^{-5}$ (LINE1), and $1.93 \times 10^{-7}$ (IAP)) and HP1α (D, ***$p = 6.23 \times 10^{-6}$ (SINE), $1.61 \times 10^{-6}$ (LINE1), and $8.73 \times 10^{-10}$ (IAP)) within the regions of repetitive sequences (LINE1, SINE and IAP) in the livers of WT and *Irtks* KO mice as measured by ChIP–qPCR. $n = 3$ animals for each condition. (E) Visualization of the co-localization of H3K9me3 and HP1α on representative genomic regions corresponding to the indicated repetitive sequences in livers from WT and Irtks-KO mice. (F) Heatmaps showing ATAC signals ranging from 5 kb upstream to 5 kb downstream of ATAC-seq peaks of repetitive sequence regions (LTR, LINE, and simple repeats) in livers from WT and *Irtks* KO mice. (G) The expression levels of repetitive sequences and non-repetitive sequences (non-REs) in the livers of WT and *Irtks* KO mice, as shown by RNA-seq analysis. Y-axis indicated the log₂ counts, as normalized by the respective expression of *Actb* with alignment to $10^4$. $n = 4$ animals for each condition. Data are presented as the mean ± SD or mean ± SEM. The *p* values of Fig. 5A, B, F and G were provided in Source data. Figure 5C and D were tested by one-way ANOVA followed by Tukey's post hoc test. The remaining plots were tested by Student's t test. Source data are available online for this figure.

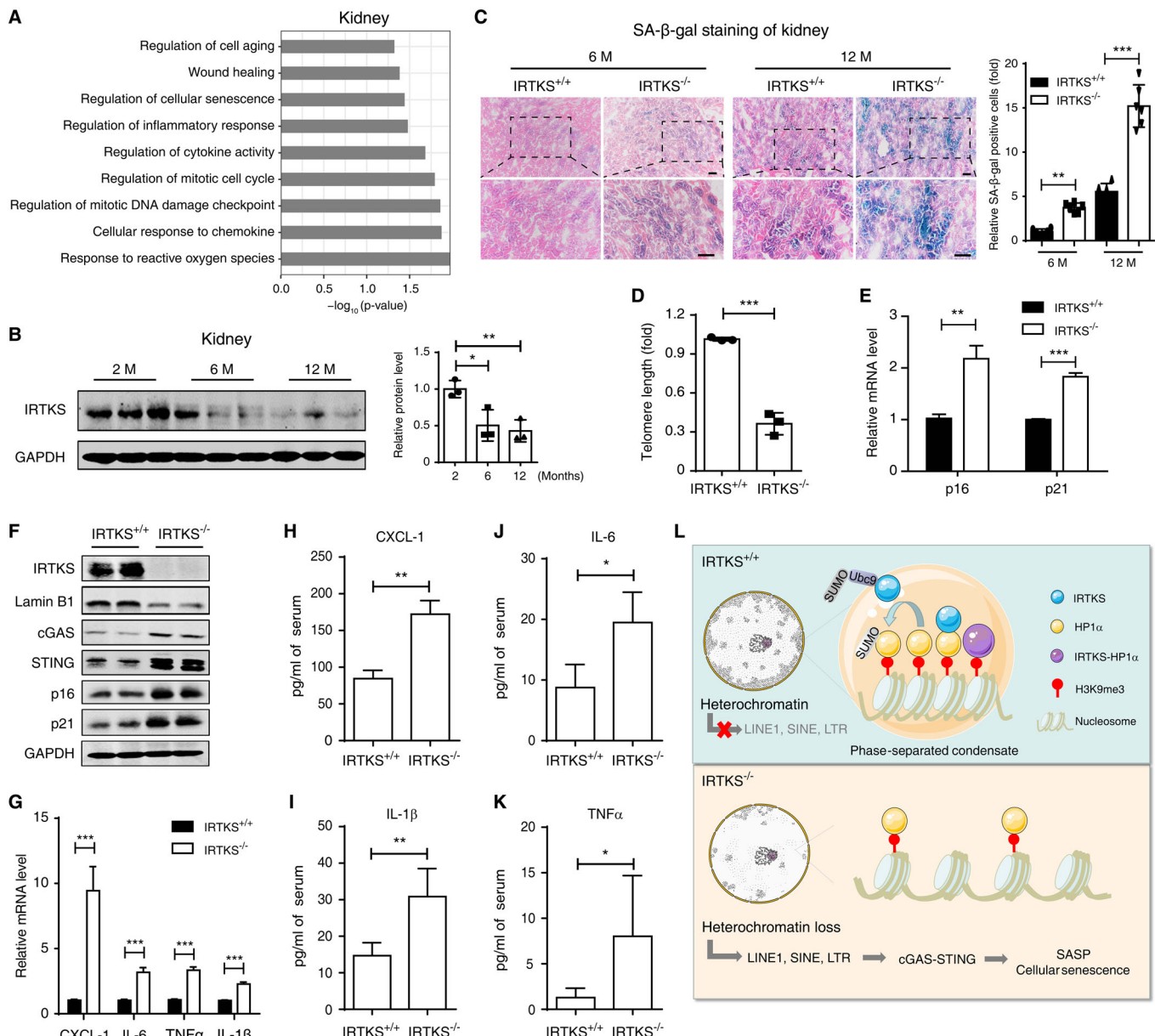

**Figure 6. IRTKS deficiency results in cellular senescence.**

(A) GO analysis of DEGs in the kidneys of Irtks-KO mice. (B) Western blotting images and quantification showing that IRTKS was downregulated in the kidneys of mice at the indicated ages ($n = 3$ animals for each condition). *$p = 0.0105$, **$p = 0.0056$. Fold change represents the normalized IRTKS signal (IRTKS/GAPDH). (C) Representative images and quantification of SA-β-gal staining in the cortex and medulla cells of kidneys of WT and *Irtks* KO mice at the indicated ages. Scale bars, 100 μm. $n = 6$ animals for each condition. **$p = 0.0017$, ***$p = 4.65 \times 10^{-11}$. (D) Telomere length analysis in kidneys of WT and Irtks-KO mice by qPCR. $n = 3$ animals for each condition. ***$p = 1.94 \times 10^{-4}$. (E) p21 and p16 transcriptional expression in kidneys of WT and Irtks-KO mice by qPCR. $n = 3$ animals for each condition. **$p = 0.0015$, ***$p = 2.19 \times 10^{-6}$. (F) Western blotting analyses of cellular senescence-related molecules in kidneys of WT and Irtks-KO mice. GAPDH was used as the loading control. (G) SASP-associated genes are significantly upregulated in kidneys of *Irtks* KO mice compared to those of WT mice. $n = 6$ animals for each condition. ***$p = 2.76 \times 10^{-4}$ (CXCL-1), $1.78 \times 10^{-5}$ (IL-6), $3.23 \times 10^{-8}$ (TNFα), $3.46 \times 10^{-7}$ (IL-1β). (H–K) Key SASP-associated molecules, including CXCL1 (H, **$p = 0.0058$), IL1β (I, **$p = 0.0034$), IL6 (J, *$p = 0.0161$), and TNFα (K, *$p = 0.0472$), were significantly elevated in sera from *Irtks* KO mice compared with those in sera from WT mice, as measured by ELISA. $n = 4$ animals for each condition. (L) A working model illustrating the mechanism by which IRTKS deficiency increases chromatin accessibility and repetitive DNA sequence reactivation, thus accelerating cellular senescence. Data are presented as the mean ± SD or mean ± SEM. Figure 6D,H–K were tested by two-tailed Student's t test. The remaining plots were tested by one-way ANOVA followed by Tukey's post hoc test. Source data are available online for this figure.

condensates, converging with HP1α liquid droplets and diverse nucleosomal arrays, which results in a decrease in chromatin accessibility and repression of repetitive DNA elements.

Unexpectedly, the IRTKS level gradually decreased during physiological aging in mice. Accordingly, we observed that IRTKS deficiency leads to global heterochromatin loss, an increase in chromatin accessibility, and derepression of repetitive DNA sequences, all of which accelerate cellular senescence and trigger cytoplasmic DNA or RNA sensors such as cGAS and RIG-I molecules to generate SASP. Taken together, our work unveils a non-canonical role of IRTKS, an I-BAR domain-containing protein, in regulating heterochromatin formation and cellular senescence, broadening our fundamental understanding of the molecular mechanisms involved in the basic biological processes of heterochromatin formation and maintenance as well as cellular senescence related to aging.

Heterochromatin organization, formation, maintenance and function have been investigated for a long time. HP1α, known as the H3K9me3 "reader", is a highly conserved protein that is well known to be the most crucial factor for heterochromatin formation and maintenance (Becker et al, 2016; Janssen et al, 2018; Millan-Zambrano et al, 2022; van de Werken et al, 2014). Previous studies have indicated the significance of HP1α posttranslational modifications (PTMs), including phosphorylation and SUMOylation, for the regulation of heterochromatin formation and functions; of these, SUMOylation seems to have a particular role in promoting the marking of pericentric heterochromatin with HP1α (Maison et al, 2011; Maison et al, 2016b; Maison et al, 2012). However, the PTMs related to the stability of HP1α through avoidance of degradation are still unclear. Interestingly, our data demonstrated that IRTKS functions as a new mediator in the recruitment of the E2 ligase Ubc9 for HP1α SUMOylation, which is a critical step for maintaining the higher HP1α levels responsible for heterochromatin formation and maintenance. Given the importance of HP1α SUMOylation in heterochromatin organization and function, future work should address the SUMOylated amino acid residues, the features of HP1α SUMOylation-mediated heterochromatin and the occupied DNA regions across genomes.

Phase separation has recently received notable attention in various biological processes, including its participating in heterochromatin formation (Larson et al, 2017; Li et al, 2020; Sanulli et al, 2019; Strom et al, 2017). However, recent reports on the topic suggest that the role of HP1α in heterochromatin formation could be independently of droplet formation under certain circumstances (Erdel et al, 2020). Moreover, it is proposed that LLPS may not be the sole mechanism for heterochromatin regulation, and other regulatory factors may also be critical (McSwiggen et al, 2019). Additional emerging evidences suggest that heterochromatin formation could be partly facilitated by some known components of heterochromatin, including MeCP2 and linker histone H1, through LLPS, besides HP1α (He et al, 2024; Larson et al, 2017; Li et al, 2020; Strom et al, 2017; Wang et al, 2020).

Significantly, it remains largely unclear whether other unknown regulators are required for LLPS-mediated heterochromatin formation. Interestingly, this study demonstrates that IRTKS functions as a new critical regulator involved in LLPS-mediated heterochromatin formation. Based on our data, IRTKS, which possesses the capability to undergo phase separation and form liquid-like condensates, plays a role in stabilizing heterochromatin

organization. However, we cannot rule out alternative mechanisms that could participate in the IRTKS-mediated heterochromatin formation in the complex environment within living cells. Therefore, it is becoming more valuable to establish the standards in research on LLPS both in vitro and in vivo, to investigate how various regulatory factors participate or fail to participate in the process of IRTKS-mediated heterochromatin formation under physiological conditions or pathological states. This exploration will provide profound and distinctive insights into the intricate regulation of heterochromatin dynamics. In addition, it should be pointed out that IRTKS condensates display an irregular, non-spherical structure within the intricate cellular milieu, resembling these 53BP1, NLRP6, and SAFB condensates as previously reported (Huo et al, 2020; Shen et al, 2021a; Zhang et al, 2022). We assume that the aspherical structure might be formed by the following possibilities: (1) Anisotropic protein-protein (eg: IRTKS-HP1α) or protein-nucleotide complexes driven by heterotypic electrostatic interactions among various molecules within the condensates in a complicated cellular environment; (2) Time-dependent changes within IRTKS condensates, where certain molecules or unknown factors within these condensates could have partial gel-like properties. Furthermore, we also noticed that the subcellular location of IRTKS puncta were slightly variant among different types of cells, and speculate that these differences might arise from these possibilities: (1) Unlike IRSp53, IRTKS lacks a CRIB domain that binds specifically to Cdc42, a small GTPase associated with the plasma membrane. This distinction could confer the different subcellular locations of IRTKS to exert diverse functions. (2) The subcellular location of IRTKS could be influenced by numerous factors under specific conditions. We believe that these significant observations could be reasonably explained by further investigation and experiments in future research, including our own.

Moreover, these IRTKS droplets can recruit and concentrate HP1α condensates through co-phase separation to drive the formation of larger heterochromatin condensates (Fig. 4). Similar to the LLPS activity of HP1α droplets (Larson et al, 2017; Wang et al, 2019), IRTKS droplets also possess the capacity to infiltrate heterochromatin condensates, which aggregate key components, including diverse DNA and nucleosomal arrays (Figs. 4 and EV5). Importantly, IRTKS is required for HP1α stability and its liquid-like properties as well as heterochromatin formation. After IRTKS depletion, in addition to heterochromatin loss in various tissues (livers, kidneys, and stomachs) and MEF cells, HP1α puncta became smaller or more diffuse, and their mobility was markedly depressed (Figs. 1 and EV1). These data reveal for the first time that IRTKS acts as a novel phase-separated component to reshape heterochromatin condensates, by converging diverse DNA and nucleosomal arrays and stabilizing HP1α droplets.

From an epigenetic perspective, condensed constitutive heterochromatin is characterized by abundant repetitive sequences, which are strictly suppressed to safeguard genomic integrity (Allshire and Madhani, 2018). Activation of repetitive sequences is associated with cellular senescence and aging-related disorders (Hu et al, 2020; Liang et al, 2022b). Strikingly, IRTKS-mediated constitutive heterochromatin occupies genomic repetitive sequences, including SINEs, LINEs, and LTRs, and therefore, the depletion of IRTKS attenuates H3K9me3 and HP1α enrichment on these repetitive sequences, leading to an aberrant reactivation of repetitive

sequences that accelerates cellular senescence. Interestingly, IRTKS deficiency also led to the significant reduction of occupation of H3K9me3 and HP1α on non-RE regions, along with an obvious increase of chromatin accessibility and transcription activity of non-repetitive DNA sequences, suggesting that IRTKS deficiency could affect almost all chromatin regions. This also implied that further investigation is needed to fully understand the effect of IRTKS on whole genomic regulation.

In addition, our data are consistent with previous observations in other cellular senescence models (Hu et al, 2020; Liang et al, 2022b), indicating that IRTKS functions as a novel epigenetic regulator to counteract cellular senescence by stabilizing heterochromatin architecture, a finding that emphasizes the link between global heterochromatin loss and cellular senescence. Cellular senescence is closely related to aging (Schmeer et al, 2019). Although the decreased Irtks levels in old mice and IRTKS deficiency-induced cellular senescence imply the possibility that this gene in involved in the physiological aging process and aging-related diseases, it should be further investigated whether and how IRTKS alleviates aging and aging-related diseases.

# Methods

### Reagents and tools table

| Reagent/Resource | Reference or Source | Identifier or Catalog Number |
|---|---|---|
| **Experimental Models** | | |
| IRTKS^f/f mice (*M. musculus*) | (Huang et al, 2018) | N/A |
| IRTKS^−/− mice (*M. musculus*) | (Huang et al, 2018) | N/A |
| C57BL/6 mice (*M. musculus*) | The SLAC Laboratory | N/A |
| **Recombinant DNA** | | |
| pGEX4T-1-IRTKS | This study | N/A |
| pGEX4T-1-IRTKS-D1 (1–249) | This study | N/A |
| pGEX4T-1-IRTKS-D2 (250–402) | This study | N/A |
| pGEX4T-1-IRTKS-D3 (403–511) | This study | N/A |
| PET28a-IRTKS | This study | N/A |
| HA-IRTKS | This study | N/A |
| Flag-IRTKS | This study | N/A |
| PET28a-EGFP-IRTKS | This study | N/A |
| PET28a-EGFP-IRTKS-D1 (1–249) | This study | N/A |
| PET28a-EGFP-IRTKS-D2 (250–402) | This study | N/A |
| PET28a-EGFP-IRTKS-D3 (403–511) | This study | N/A |
| PET28a-EGFP-IRTKS-IDR (250–511) | This study | N/A |
| PET28a-EGFP-IRTKS-IDR-Mut | This study | N/A |

| Reagent/Resource | Reference or Source | Identifier or Catalog Number |
|---|---|---|
| PET28a-EGFP | This study | N/A |
| PET28a-mCherry | This study | N/A |
| HA-EGFP | This study | N/A |
| HA-EGFP-IRTKS | This study | N/A |
| pGEX4T-1-HP1α | This study | N/A |
| pGEX4T-1-HP1α-D1 (1–78) | This study | N/A |
| pGEX4T-1-HP1α-D2 (79–121) | This study | N/A |
| pGEX4T-1-HP1α-D3 (122–191) | This study | N/A |
| PET28a-HP1α | This study | N/A |
| Flag-HP1α | This study | N/A |
| PET28a-mCherry-HP1α | This study | N/A |
| PET30a-H2A | Gift from Guohong Li, Chinese Academy of Sciences, Beijing, China. | N/A |
| PET30a-H2B | Gift from Guohong Li, Chinese Academy of Sciences, Beijing, China. | N/A |
| PET30a-H3 | Gift from Guohong Li, Chinese Academy of Sciences, Beijing, China. | N/A |
| PET3a-H4 | Gift from Guohong Li, Chinese Academy of Sciences, Beijing, China. | N/A |
| pWM530 (12 × 177 bp) | Gift from Guohong Li, Chinese Academy of Sciences, Beijing, China. | N/A |
| H3K9C/C110A | Gift from Haitao Li, Tsinghua University, Beijing, China. | N/A |
| GFP-SUMO1 | Gift from Jiemin Wong, East China Normal University, Shanghai, China. | N/A |
| GST-SUMO1 | This study | N/A |
| GST-SAE1/2 | Gift from Ping Wang, Tongji University, Shanghai, China. | N/A |
| GST-UBC9 | Gift from Ping Wang, Tongji University, Shanghai, China. | N/A |
| pE1/E2/SUMO1 | Gift from Ping Wang, Tongji University, Shanghai, China. | N/A |
| **Antibodies** | | |
| Rabbit Anti-Histone H3, trimethyl(Lys9) | Abcam | Cat# ab8898; RRID: AB_306848 |
| HP1α antibody | Cell Signaling Technology | Cat#2616S; RRID: AB_2070987 |
| HP1β antibody | Abcam | Cat# ab10811; RRID: AB_297490 |

| Reagent/Resource | Reference or Source | Identifier or Catalog Number |
|---|---|---|
| HP1γ antibody | Abcam | Cat# ab154871; RRID: AB_2924364 |
| Lamin B1 Rabbit pAb | Abclonal | Cat# A1910; RRID: AB_2862592 |
| p21 antibody | Abcam | Cat #ab109199; RRID: AB_10861551 |
| SUMO1 Rabbit pAb | Abclonal | Cat# A2130; RRID: AB_2764149 |
| IRTKS Rabbit pAb | Homemade | (Huang et al, 2018) |
| DNMT3A Rabbit mAb | Abclonal | Cat# A19659; RRID: AB_2862720 |
| DNMT3B Rabbit pAb | Abclonal | Cat# A2899; RRID: AB_2764719 |
| Anti-UBE2I/UBC9 antibody | Abcam | Cat# ab75854; RRID: AB_1310787 |
| MBD2 Rabbit pAb | Abclonal | Cat# A2241; RRID: AB_2764245 |
| SUV39H1 Rabbit pAb | Abclonal | Cat# A3277; RRID: AB_2765020 |
| Anti-CDKN2A/ p16INK4a Antibody (F-12) | Santa Cruz Biotechnology | Cat #sc-1661; RRID: AB_628067 |
| cGAS Rabbit pAb | Abclonal | Cat# A8335; RRID: AB_2770305 |
| STING/TMEM173 Rabbit pAb | Abclonal | Cat# A3575; RRID: AB_2765161 |
| **Oligonucleotides and other sequence-based reagents** | | |
| Plasmid cloning primers | This study | Dataset EV2 |
| qPCR primers | This study | Dataset EV2 |
| ChIP-qPCR primers | This study | Dataset EV2 |
| Telomere length detection | This study | Dataset EV2 |
| gRNA sequences | This study | Dataset EV2 |
| **Chemicals, Enzymes and other reagents** | | |
| 1,6-hexanediol | Sigma-Aldrich | 240117; CAS: 629-11-8 |
| Guanidine hydrochloride | Diamond | A100287; CAS: 50-01-1 |
| Bromocholine bromide | Macklin | B802596; CAS: 2758-06-7 |
| Isopropyl-β-D-thiogalactopyranoside (IPTG) | BBI | A600168; CAS: 367-93-1 |
| D/L-methionine | Merck | M9500; CAS:59-51-8 |
| 1,4-Dithiothreitol (DTT) | Diamond | A100281; CAS: 3483-12-3 |
| Nonidet P-40 | Sigma-Aldrich | I3021; CAS: 9002-93-1 |
| Puromycin | BBI | A610593; CAS: 58-58-2 |
| PEG8000 | BBI | A600433; CAS: 25322-68-3 |
| RNase A | TransGen Biotech | GE101-01 |
| Hoechst 33342 | Beyotime | C1028 |

| Reagent/Resource | Reference or Source | Identifier or Catalog Number |
|---|---|---|
| **Software** | | |
| Image J | (Schneider et al, 2012) | https://imagej.nih.gov/ij/ |
| PONDR algorithms | (Sabari et al, 2018) | http://www.pondr.com/ |
| D²P² algorithms | (Oates et al, 2013) | https://d2p2.pro/ |
| PyMOL | Schrodinger | https://pymol.org/2/ |
| Bowtie 2 | (Langmead and Salzberg, 2012) | http://bowtie-bio.sourceforge.net/bowtie2/index.shtml |
| RepEnrich2 | (Criscione et al, 2014) | https://github.com/nerettlab/RepEnrich2 |
| Deeptools | (Ramirez et al, 2016) | https://deeptools.readthedocs.io/en/develop/index.html |
| AlphaFold algorithms | (Jumper et al, 2021) | https://alphafold.ebi.ac.uk/ |
| **Other** | | |
| PrimeScript RT reagent Kit with gDNA Eraser | Takara | Cat#RR047B |
| SA-β-gal staining kit | Beyotime | Cat# C0602 |

## Cell lines

Mouse embryonic fibroblasts (MEFs) were isolated from E13.5 mouse embryos. HEK293T cells were provided by prof. Yujia Cai (Shanghai Jiao Tong University). Embryonic mouse fibroblast NIH3T3 cells were provided by prof. Yan Zhang (Shanghai Jiao Tong University). SK-Hep-1 cells were obtained from National Collection of Authenticated Cell Cultures. MEFs, NIH3T3, SK-Hep-1 and HEK293T were cultured in DMEM (Dulbecco's modified Eagle medium; GIBCO) supplemented with 10% fetal bovine serum (GIBCO), penicillin (GIBCO), and streptomycin (GIBCO). All cells were incubated at 37 °C in 5% $CO_2$.

## Experimental animals

IRTKS-knockout mice were previously generated and were housed in Shanghai Jiao Tong University (Huang et al, 2013). For liver-specific IRTKS overexpression, recombinant adeno-associated virus serotype 8 (AAV8) vectors containing a liver-specific TBG promoter and mouse IRTKS were injected into wild-type C57BL/6 mice (SLAC Laboratory, Shanghai) by tail vein at a dose of $1 \times 10^{12}$ viral titer/mL in a total volume of 100 μl/mouse at the age of 8 weeks, where empty AAV8 vectors was used as negative control (ZsGreen). These AAV8 vectors were constructed, amplified, and purified by Hanbio Biotechnology (Shanghai, China). Mice were sacrificed after six months of feeding, and then livers were collected for analysis. All animal experiments were approved by the Animal Use and Care Committee of Shanghai Jiao Tong University. The mice were housed under specific pathogen-free (SPF) conditions with a 12 h light/dark cycle, and allowed free access to food and water during the study.

## Immunofluorescence imaging

Cells were cultured on the coverslip, fixed by 4% paraformaldehyde for 10 min at room temperature. Tissues were freshly frozen in O.C.T. compounds, cryosectioned and fixed with 4% paraformaldehyde for 15 min at room temperature, and then washed twice with PBS and treated by 0.5% Triton X-100 for 10 min, blocked by 1–4% BSA (based on different antibodies) for 1 h, cultured by diluted antibody overnight at 4 °C, followed by adding second-fluorescence antibody for 1 h at room temperature.

Nuclei were stained with DAPI. The slides were imaged with a NIKON A1 microscope. NIS-Elements AR Analysis was used to analyze these images.

## Generation of stable cell lines

The Irtks and HP1α KO cell lines were generated by CRISPR/Cas9 system. Three gRNA oligos for each target gene were designed (http://www.e-crisp.org/E-CRISP/index.html). Lentiviral shuttle plasmid containing sgRNA targeting human Irtks and HP1α together with helper vectors of psPAX2 and PMD2.G were co-transfected to HEK293T using lipofectamine 2000 (Invitrogen) according to the manufacturer's instructions. After 48 and 72 h transfection, the viral supernatant was collected and passed through 0.45 μm filter diluted 1:1 with fresh medium containing 8 μg/ml polybrene and used to infect the target cells. Stable cells were selected in 2 μg/ml puromycin in culture medium. Stably knockouts of IRTKS and HP1α were verified by western blotting.

## Western blotting

The collected cells or tissues were washed with cold PBS, then lysed in lysis buffer (50 mM Tris/HCl, pH 7.5, 150 mM NaCl, 0.1% NP40, 1 mM EDTA) with protease inhibitors for 30 min on ice. The isolated proteins were separated on SDS-PAGE gels and transferred to nitrocellulose. The membrane was blocked with 5% nonionic-Skimmed milk in TBS/0.1% Tween 20 (TBST), and then incubated with primary antibodies and secondary antibodies for protein detection. All antibodies are listed in Dataset EV2.

## Quantitative RT-PCR

The RNA samples were reversely transcribed via PrimeScript RT reagent Kit with gDNA Eraser (Takara). Quantitative PCR was conducted with ChamQ Universal SYBR qPCR Master Mix (Vazyme) on the Roche LightCycler 96 Real-time System. Gene expression was normalized to expression of GAPDH in the same sample using the ΔCt method. The primers used for the indicated gene products are described in Dataset EV2.

## Immunoprecipitation assay

Immunoprecipitation assay was performed as previously described (Huang et al, 2018). Briefly, cells were collected and lysed in lysis buffer (50 mM Tris-HCl, pH 7.5, 150 mM NaCl, 1 mM EDTA, 1 mM DTT, 1% Triton X-100 and protease inhibitor cocktail (Sigma-Aldrich, St. Louis, Missouri, USA), following which the samples were centrifuged at $12,000 \times g$ at 4 °C for 30 min. The supernatant was incubated with the indicated antibodies at 4 °C for

2 h and then incubated with Protein A/G PLUS-Agarose (Santa Cruz, sc-2003) at 4 °C for overnight. After centrifugation at $900 \times g$ at 4 °C for 2 min, the supernatant was discarded and the beads were washed with lysis buffer, and then analyzed by immunoblotting.

## Electron microscopy

Briefly, cells or tissues were fixed with 2.5% glutaraldehyde overnight at 4 °C, and then treated with 1% osmic acid, followed by gradient dehydration with alcohol. Then, samples were treated with a mixture of epoxypropane and resin, embedded with pure resin, and sliced with a diamond tool kit. Transmission electron microscopy (TEM) analyses were conducted with a FEI Tecnai G2 Spirit 120 kV transmission electron microscope in Instrumental Analysis Center of Shanghai Jiao Tong University.

## GST pull-down assay

Plasmid constructs were expressed in BL21 (DE3) cells and purified using glutathione sepharose beads or Ni-NTA beads. Concentration of purified protein was estimated by coomassie staining. Beads coated with GST or GST fusion proteins were incubated with His-tagged fusion proteins under rotation at 4 °C and the beads were washed with ice-cold PBS. The resin was eluted with 2× SDS loading buffer and analyzed by western blotting.

## Protein expression and purification

The corresponding expression plasmids were transformed into E. coli strain BL21 (DE3) cells or *E. coli* strain Rosetta (DE3) cells, then induced with 0.5 mM isopropyl-β-D-thiogalactopyranoside (IPTG) for overnight at 18 °C. The bacteria were collected by centrifugation at 8000 rpm for 10 min at 4 °C, then resuspended with lysis buffer (20 mM HEPES pH 7.5, 500 mM NaCl, 1 mM PMSF) followed by a high-pressure homogenizer before centrifugation. The supernatant was purified with Ni NTA beads (smart-lifesciences) or Glutathione-Sepharose agarose (GE Healthcare, 17-0756-01) before being stored in 20 mM Tris (pH 7.4), 200 mM NaCl, 10% glycerol and 1 mM DTT at −80 °C.

## Phase separation assays

In-cell assays were carried out on glass bottomed 35 mm dishes (Cellvis), which were coated with 3% BSA for 15 min and then washed with MilliQ $H_2O$. For in vitro experiments such as FRAP and time-lapse imaging experiments, phase separation was recorded on 384 low-binding multi-well 0.17 mm microscopy plates (Cellvis). Imaging was performed with a NIKON A1 microscope equipped with a 60× or 100× oil immersion objective. NIS-Elements AR Analysis was used to analyze these images. Quantification of the droplet area of randomly selected liquid-like condensates was analyzed by ImageJ. In brief, the droplet fluorescence intensity thresholds were set to identify droplet boundaries. Background intensity was subtracted, and the droplet area was automatically measured by ImageJ. Statistical significance was evaluated by GraphPad Prism software.

## Fluorescence recovery after photobleaching (FRAP)

The FRAP was performed as described with minor modifications (Strom et al, 2017). In vivo and in vitro FRAP experiments were

performed with a NIKON A1 microscope equipped with a 100× oil immersion objective. Droplets were bleached with a 488- or 561-nm laser pulse every 4 s. Recovery from photobleaching was recorded for the indicated time.

## Fluorescent DNA production

Fluorescent DNA production was performed as previously described (Li et al, 2020). Fluorescent DNA for droplet assays was produced by amplifying plasmid DNA using oligonucleotide primers with 5'-Cy5 fluorophore modifications (GENEWIZ). Fluorescent PCR products were gel purified using the TIANgel midi purification Kit (TIANGE, DP209).

## Poly-nucleosome purification

Poly-nucleosome arrays were purified from 293T using a protocol as previously reported with minor modifications (Li et al, 2020). Briefly, nuclei were isolated from 293T cells by resuspending cells in lysis buffer (10 mM MES pH 6.5, 15 mM NaCl, 0.5 mM sodium metabisulfite, 60 mM KCl, 5 mM MgCl₂, 1 mM CaCl₂, 10 mM sodium butyrate, 0.5 mM Benzamidine-HCl, 0.5 mM DTT, 0.5% Triton X-100, 0.25 M sucrose, 0.1 mM PMSF) and douncing with a dounce homogenizer (20 strokes with pestle B). The nuclei were then and digested with a limited amount of micrococcal nuclease, and then the samples were centrifuged at maximum speed for 10 min. To purify poly-nucleosome arrays, the supernatant was loaded on a sucrose gradient and centrifuged for 15 h at 4 °C in a SW32-Ti rotor (Beckman Coulter) at 24,000 rpm. The sucrose gradients were 5–45% in a base buffer of HEPES pH 7.5 and 200 mM NaCl. Individual fractions corresponding to poly-nucleosome arrays were collected. To determine the length distribution of the poly-nucleosome arrays in each faction, DNA was purified from each fraction and analyzed on an agarose gel. Fractions containing nucleosomal arrays ranging between 7 and 20 nucleosomes in length were pooled and dialyzed against buffer (20 mM HEPES pH 7.5, 200 mM NaCl, 5 mM MgCl₂). Purified poly-nucleosomes were stored in liquid nitrogen until ready to use in droplet assays.

## Nucleosome assembly

Plasmid of DNA template harboring 12 × 177 bp tandem repeats of Widom 601 sequence was a gift from Dr. Guohong Li at Institute of Biophysics, Chinese Academy of Sciences. Preparation of 12 × 177 bp 601 DNA template followed the method described previously (Dyer et al, 2004). Histone octamer assembly was performed the method of serial dialysis. Briefly, four histones at equal molar amounts in unfolding buffer (20 mM Tris-HCl, pH 7.5, 7 M guanidine hydrochloride, 10 mM DTT) were dialyzed into refolding buffer (2 M NaCl, 10 mM Tris-HCl, pH 8.0, 1 mM EDTA, 5 mM β-mercaptoethanol) and purified by a Superose 6 column (GE Healthcare, USA). Nucleosomal arrays were assembled using the salt dialysis method as previously described (Wang et al, 2020).

## UPLC/VION mass spectrometry

UPLC/VION mass spectrometry was carried out using Acquity UPLC I-class/VION IMS QTOF instrument (Waters, USA) in

Instrumental Analysis Center of Shanghai Jiao Tong University. Prior to analysis, mCherry-HP1α and SUMO-mCherry-HP1α proteins desalted with ACQUITY UPLC BEH C4 column. Protein sample was introduced into the instrument, then the multiply charged mass spectrum was acquired and analyzed on a UNIFI software to evaluate the average mass molecular weight of protein samples.

## In vitro sumoylation reconstitution assay

The in vitro sumoylation assays were carried out as previously described (Xia et al, 2015). In brief, 3 μg GST-HP1α was incubated in 20 μL reactions containing 50 mM Tris-HCl pH 7.5, 5 mM MgCl₂, 2 mM ATP, 1 mM DTT, 2 μg SAE1/2, 1 μg SUMO-1, 1.5 μg Ubc9. The reactions were incubated at 37 °C for 3 h, stopped by addition of the 2× SDS loading buffer and followed by western blotting.

## SA-β-Gal staining

Cultured cells were fixed in SA-β-galactosidase (SA-β-gal) staining fix solution for 20 min at room temperature, then the cells were incubated with SA-β-gal staining solution (Beyotime Biotechnology, C0602) overnight at 37 °C. The cells were washed with PBS and visualized under a bright field microscope. For SA-β-gal staining of tissues, the frozen sections were fixed in SA-β-gal staining fixative solution for 15 min at room temperature. The frozen sections were incubated with SA-β-gal staining solution overnight at 37 °C. After incubation, the sections were counter-stained with Eosin. Finally, samples were observed under a microscope.

## Telomere length analysis

The analysis of telomere length by quantitative real-time PCR was performed as previously described (Hu et al, 2020). The primers used for detection of telomere length are listed in Dataset EV2.

## Endogenously-tagged cell line generation

The endogenously-mEGFP-tagged IRTKS cell lines were generated by CRISPR/Cas9 system. Oligos coding for guide RNAs targeting IRTKS was cloned into a px330 vector expressing Cas9 and mCherry. The sequence that was targeted for IRTKS was 5' ATTCGATGAGAGGACAGCCA 3'. To generate the donor plasmid, repair templates containing mEGFP, a GS linker and 800 bp homology arms of targeted gene were amplified. To obtain the mEGFP-KI HEK293T cell lines, cells were transfected with 1 μg px330 vector and 2 μg the corresponding donor plasmid. 1 week later, the cells were inspected by fluorescence microscopy and FACS (BD LSRFortessa). About two weeks after single-cell sorting, mEGFP-positive single colonies were picked up, and then the mEGFP-KI HEK293T cell lines were further confirmed by sequencing and western blotting.

## ELISA analysis

Blood samples from WT and Irtks-KO mice were collected and stewed 1 h at room temperature, and then centrifuged (3000 rpm,

15 min) to gain serum. Secretion of mouse IL1β, IL6, CXCL1 and TNFα was measured using Mouse Interleukin 1β (IL1β) ELISA Kit (ABclonal, RK00006), Mouse Tumor Necrosis Factor alpha (TNFα) ELISA Kit (ABclonal, RK00027) kit, Mouse CXCL1/KC ELISA Kit (ABclonal, RK00038) and IL-6 (Interleukin-6) Mouse ELISA Kit (ABclonal, RK00008) according to the manufacturers' instructions.

## Strand-specific total RNA-seq

Total RNA was isolated using TRIzol reagent (Invitrogen). Specific total RNA library preparation and raw data quality control were performed by Personal Biotechnology Co., Ltd. (Shanghai, China).

## RNA-seq data processing

Low-quality sequencing reads were removed by FastQC (version 0.11.9) (http://www.bioinformatics.babraham.ac.uk/projects/fastqc/). After subtracting adapters, filtered reads of liver tissues and kidney tissues were mapped to the GRCm38 genome by TopHat (v2.1.1) (Trapnell et al, 2009). PCR duplicates were removed by SAMtools (Danecek et al, 2021). Gene expression was subsequently estimated by annotation reference from GenCode (vM23) and featureCounts (version 2.0.3) (Liao et al, 2014). For analysis of expression on repetitive elements (REs), multi-reads were aligned to REs and counted using RepEnrich2 (https://github.com/nerettilab/RepEnrich2) with recommended parameters. The RE annotation file was obtained from UCSC RepeatMasker where REs labeled with different level names. We defined non-repetitive sequences (non-REs) as those mRNA sequences that did not contain any of the repeat sequences. After merging the counts of these genes and REs based on different samples, the differential expression and Gene Ontology (GO) enrichment analyses were operated by using R packages DESeq2 (version 1.34.0) and clusterProfiler (version 4.2.2). As for the classic types of REs, we calculated their relative expression in these paired liver samples, by normalizing with the respective expression levels of *Actb* as a control locus. *p* values were calculated in the paired knockout and wild-type samples. These data were shown in Dataset EV1.

## Chromatin immunoprecipitation (ChIP)-qPCR and ChIP-seq

ChIP was carried out previously with minor modification (Shen et al, 2021b). Briefly, Cells were crosslinked with 1% formaldehyde for 10 min at room temperature and then quenched by Glycine, the cell pellets were lysed on ice and sonicated with a Bioruptor® Plus sonication device (Diagenode). And then the supernatant subjected to immunoprecipitations with 2 μg of anti-HP1α, H3K9me3 antibody or control IgG conjugated with Protein A/G PLUS-Agarose (Santa Cruz, sc-2003) at 4 °C for overnight. The beads were then washed 7 times, followed by elution and reverse cross-link at 65 °C for overnight. The ChIP and input were then purified and used for qPCR analysis. The primers used are listed in Dataset EV2. ChIP sequencing and raw data quality control were performed by Personal Biotechnology Co., Ltd. (Shanghai, China).

## ChIP-seq data processing

For the processing of H3K9me3 and HP1α ChIP-seq date, raw reads, from both MEF cells and liver tissues, were trimmed using by Fastp (version 0.23.2) and removed adapters (Chen et al, 2018). Trimmed reads were aligned to the UCSC mouse genome build mm10 (http://hgdownload.cse.ucsc.edu/goldenPath/mm10/bigZips/chromFa.tar.gz) using Bowtie2 (version 2.4.4) (Langmead and Salzberg, 2012). SAM files were then sorted and operated duplicates marked to form mature BAM with bai format index by SAMtools (version 1.9) and picard (version 2.26.11) (https://broadinstitute.github.io/picard/) (Danecek et al, 2021). Subsequently, Peaks were called using MACS2 software (version 2.2.7) according to the relative loose parameters stated below (*p* < 0.01), and set to detect narrow peaks for both H3K9me3 and HP1α (Zhang et al, 2008). Peaks were operated by annotatePeaks.pl of HOMER software (version 4.11) (Heinz et al, 2010) to get the annotation on non-REs. Peaks enrichment at repetitive element were generated by using BEDTools intersect command of peaks against seven types of repeat annotations from UCSC RepeatMasker (https://hgdownload.soe.ucsc.edu/goldenPath/mm10/database/rmsk.txt) (Quinlan and Hall, 2010). To compare the peaks of a certain REs or non-RE regions, we merged the peak files of these samples to generate the unbiased genome positions for visualization. Then, packaged commands from deepTools were used coherently for downstream analysis (Ramirez et al, 2016). BamCompare was used generate bigwig (bw) files of ChIP reads normalized to paired input files, while genome coverage bigwig files for IGV were built by bamCoverage with the parameter "--binSize 10 --normalizeUsing BPM". Co-location of H3K9me3 and HP1α was determined by merged wild-type MEF cells and liver tissues ChIP annotated peaks. Finally, plotHeatmap and plotProfile were used to plot multiple ChIP-seq samples onto +/−3 kb of appointed locations, encompassing repetitive sequences peaks and co-location, after calculating read enrichment scores at the regions or points by using computeMatrix. *p* values with adjusted false discovery rate (FDR) were generated by paired-*t* test on the above enrichment score matrix.

## ATAC-seq and data processing

MEF cells were washed with ice-cold PBS and lysed in buffer containing 10 mM Tris-HCl, pH 7.4, 3 mM MgCl$_2$, 10 mM NaCl, 0.1% NP40. The cell pellets were incubated in transposition mix containing Tn5 transposase at 37 °C for 30 min. The purified DNA was ligated with adapters and PCR amplified. And the sequencing libraries were purified with AMPure beads and sequenced on the Illumina NovaSeq 6000 platform. Basic data processing was similar with ChIP-seq analysis. We used "bigwigCompare –operation mean" command to merge the biological replicates of liver samples. Peaks were identified by MACS2 with *p*-value < 0.01. HOMER help to annotate peaks and analysis motif with default setting. Because there were large number of peaks without significant difference within a given group, we merged the different peaks from wild-type and knockout samples, respectively. The differential analysis of peaks was performed by using command makeTagDirectory and getDifferentialPeaks in HOMER with parameter "-F 1.0 -P 0.0001" for livers and "-F 2.0 -P 0.01" for MEFs, to get relatively consistent amount of genome positions. Peak signals were aligned to bigwig tracks for figures plotted by IGV (Integrated Genomic Viewer) and deepTools packages (Ramirez et al, 2016). Similarly, *p* values with adjusted false discovery rate were figured out by paired-t test.

## Statistics and reproducibility

The data are presented as the means ± standard deviation (SD) or means ± standard error of the mean (SEM). The mean is calculated from truly independent experiments. Statistical analyses were performed with a two-tailed unpaired Student's t-test (for two-sample comparison and certain pairwise comparisons to a specific sample in a multiple sample group) or one-way ANOVA followed by Tukey post hoc tests (for multiple groups), Statistical analysis was performed using GraphPad Prism, and the values of $p < 0.05$ were considered statistically significant.

Sample numbers are indicated in the figure legends. Results of images, staining, or gels were reproducible with at least two independent experiments or prepared samples, with the representative ones shown in the figures. All in vitro experiments were replicated at least three times, and in vitro measurement was taken from a distinct condensate.

## Data availability

The ChIP-seq, RNA-seq and ATAC-seq datasets generated during this study are available at the National Center for Biotechnology Information (NCBI) Sequence Read Archive (SRA) database: PRJNA888247 (PRJNA888247 - SRA - NCBI (nih.gov)) and PRJNA714184 (PRJNA714184 - SRA - NCBI (nih.gov)). The scripts for genomic data analyses and all other data are available from the corresponding author upon request.

The source data of this paper are collected in the following database record: biostudies:S-SCDT-10_1038-S44318-024-00212-3.

## Peer review information

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

## Acknowledgements

We sincerely thank Dr. Xiaotao Li (School of Life Sciences of East China Normal University) for helping with histone octamers purification. This work was supported by the National Key Research and Development Program of China (2022YFA1302700 to Z-GH), the National Natural Science Foundation of China (82073116 and 82272969 to Z-GH), National Key Research and Development Program of China (2020YFC2002705 to Z-GH), National Science and Technology Major Project (2017ZX10203207 to Z-GH), Natural Science Foundation of Shanghai (21JC1403200 to Z-GH), Shanghai Jiao Tong University Scientific and Technological Innovation Funds (2019TPA09 to Z-GH.) and the National Natural Science Foundation of China (82203262 to Z-NL).

## Author contributions

**Jia Xie**: Data curation; Formal analysis; Validation; Investigation; Visualization; Methodology; Writing—original draft; Writing—review and editing. **Zhao-Ning Lu**: Formal analysis; Funding acquisition; Investigation. **Shi-Hao Bai**: Software; Investigation. **Xiao-Fang Cui**: Resources. **He-Yuan Lian**: Investigation. **Chen-Yi Xie**: Resources. **Na Wang**: Resources. **Lan Wang**: Resources. **Ze-Guang Han**: Conceptualization; Supervision; Funding acquisition; Project administration; Writing —review and editing.

Source data underlying figure panels in this paper may have individual authorship assigned. Where available, figure panel/source data authorship is listed in the following database record: biostudies:S-SCDT-10_1038-S44318-024-00212-3.

## Disclosure and competing interests statement

The authors declare no competing interests.

# Expanded View Figures

**Figure EV1. IRTKS regulates heterochromatin formation.** ▶

(A, B) Quantification of electron-dense regions (EDRs) around the nucleolus of the livers (**A**, \*\*\*$p = 2.72 \times 10^{-10}$) and kidneys (**B**, \*\*\*$p = 2.73 \times 10^{-9}$) from WT and *Irtks* KO mice. $n = 10$ cells analyzed for each condition. (**C–E**) Representative images (**C**) and quantification of EDRs at the nuclear periphery (**D**, \*\*\*$p = 3.09 \times 10^{-9}$) and around the nucleolus (**E**, \*\*\*$p = 5.49 \times 10^{-11}$) of the stomach tissues of WT and *Irtks* KO mice. Red arrows indicate the electron-dense heterochromatin regions. Nu, nucleolus. $n = 10$ cells analyzed for each condition. Scale bar, 1 μm. (**F**) Quantification of EDRs around the nucleolin in MEF cells. $n = 10$ cells analyzed for each condition. \*\*\*$p = 1.22 \times 10^{-13}$, \*$p = 0.0134$. (**G, H**) Electron microscopy images (**G**) and quantification of EDRs at the nuclear periphery (**H**, \*\*\*$p = 2.14 \times 10^{-8}$) and around the nucleolus (**I**, \*\*\*$p = 1.19 \times 10^{-10}$) in WT and Irtks-KO SK-Hep-1 cells. Red arrows indicate the electron-dense heterochromatin regions. Nu, nucleolus. $n = 10$ cells analyzed for each condition. Scale bar, 1 μm. (**J–O**) Electron microscopy images and quantification of the electron-dense heterochromatin regions at the nuclear periphery and around the nucleolus in MEFs (**J–L**, respectively) and SK-Hep-1 cells (**M–O**, respectively) that were transfected with empty vector or Flag-IRTKS construct. Red arrows indicate the electron-dense heterochromatin regions. Nu, nucleolus. $n = 10$ cells analyzed for each condition (**K**, \*\*\*$p = 1.39 \times 10^{-9}$; **L**, \*$p = 0.0370$; **N**, \*\*\*$p = 5.68 \times 10^{-6}$; **O**, \*\*\*$p = 6.90 \times 10^{-8}$). Scale bar, 1 μm. (**P, Q**) Representative confocal images (**P**) and line scan analysis (**Q**) of HP1α foci (red) and nuclei (DAPI, blue) in the stomach tissues of WT and *Irtks* KO mice. Quantification of lines scanned across HP1α foci and nuclei at the position depicted by the white arrow. Scale bar, 5 μm. (**R, S**) Western blotting analyses of HP1α expression in the livers (**R**) and kidneys (**S**) of WT and Irtks-KO mice. GAPDH was used as the loading control. (**T**) Representative confocal images and line scan analysis (right) on H3K9me3 (green) and nuclei (DAPI, blue) in the livers of WT and *Irtks* KO mice. Quantification of lines scanned across H3K9me3 foci and nuclei at the position depicted by the white arrow. Scale bar, 5 μm. (**U**) Representative confocal microscopy and line scan analysis of SK-Hep-1 cells showing the location of EGFP-HP1α foci. Nuclei were labeled with Hoechst 33342 (blue). Quantification of lines scanned across HP1α foci and nuclei at the position depicted by the white arrow. Scale bar, 5 μm. (**V**) Live-cell images and fluorescence recovery curves of FRAP experiments of EGFP-HP1α in SK-Hep-1 cells. Red arrow indicates the bleached point, which is boxed and amplified in the images on the right. $n = 8$ biological replicates for the FRAP curve construction. \*\*\*$p = 3.10 \times 10^{-9}$. Scale bar, 5 μm. (**W**) Representative confocal microscopy and line scan analysis of MEF cells showing the location of EGFP-HP1α foci. Nuclei were labeled with Hoechst 33342 (blue). Quantification of lines scanned across HP1α foci and nuclei at the position depicted by the white arrow. Scale bar, 5 μm. (**X**) Live-cell images and fluorescence recovery curves of FRAP experiments of EGFP-HP1α in MEF cells. Red arrow indicates the bleached point, which is boxed and amplified in the images on the right. $n = 8$ biological replicates for the FRAP curve construction. \*\*\*$p = 1.03 \times 10^{-6}$. Scale bar, 5 μm. Data are presented as the mean ± SD. Figure EV1V and y were tested by two-way ANOVA. The remaining plots were tested by two-tailed unpaired Student's t test. Source data are available online for this figure.

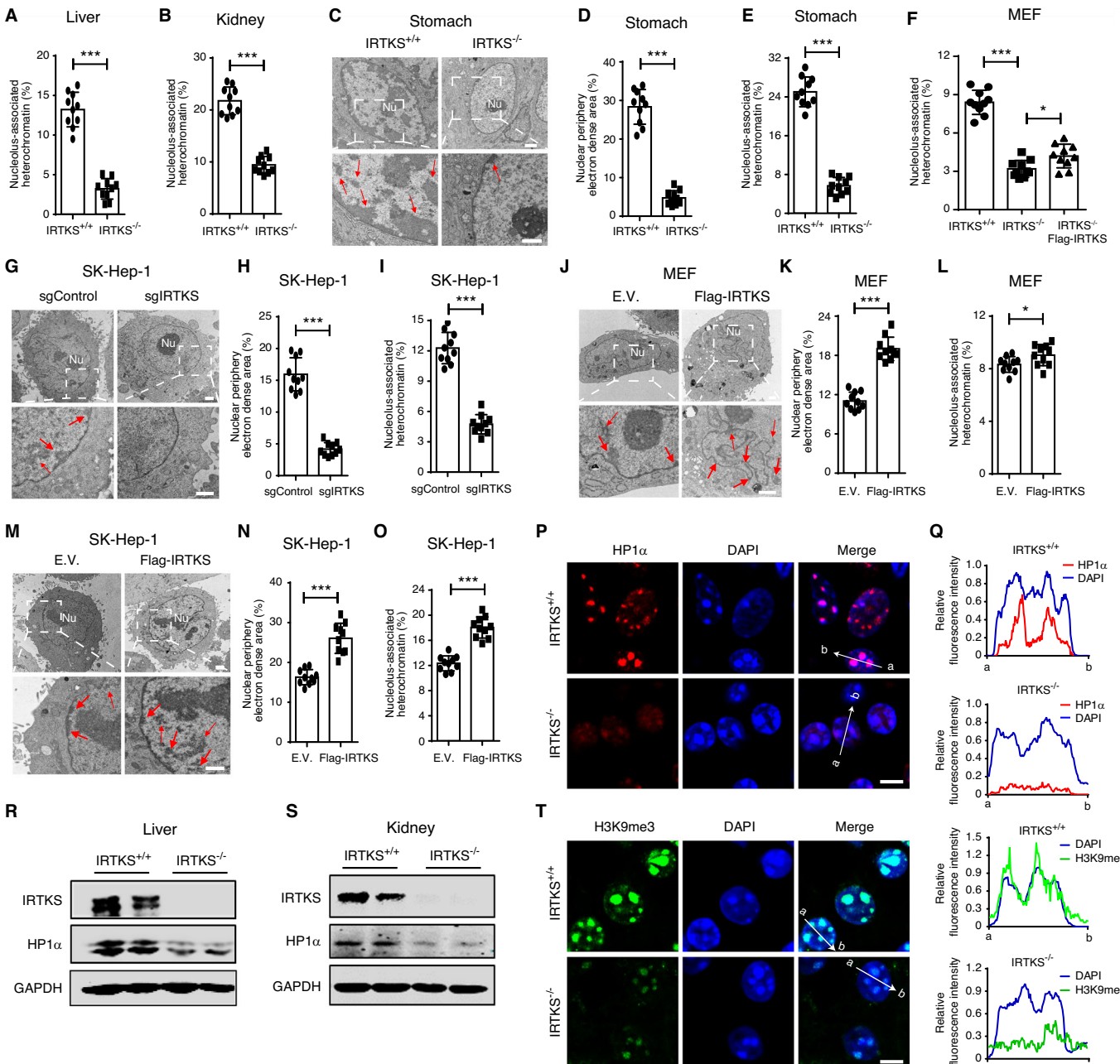

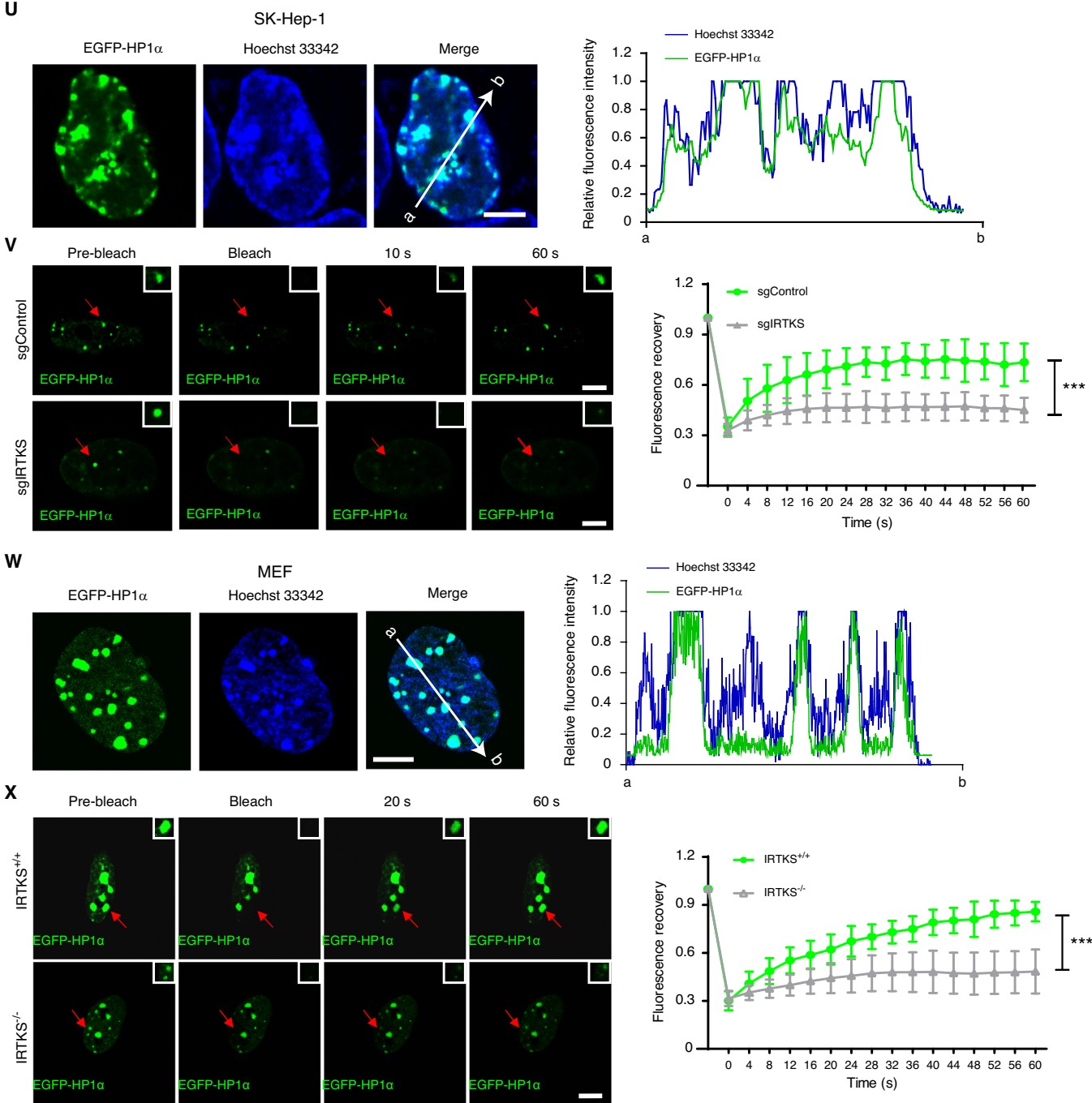

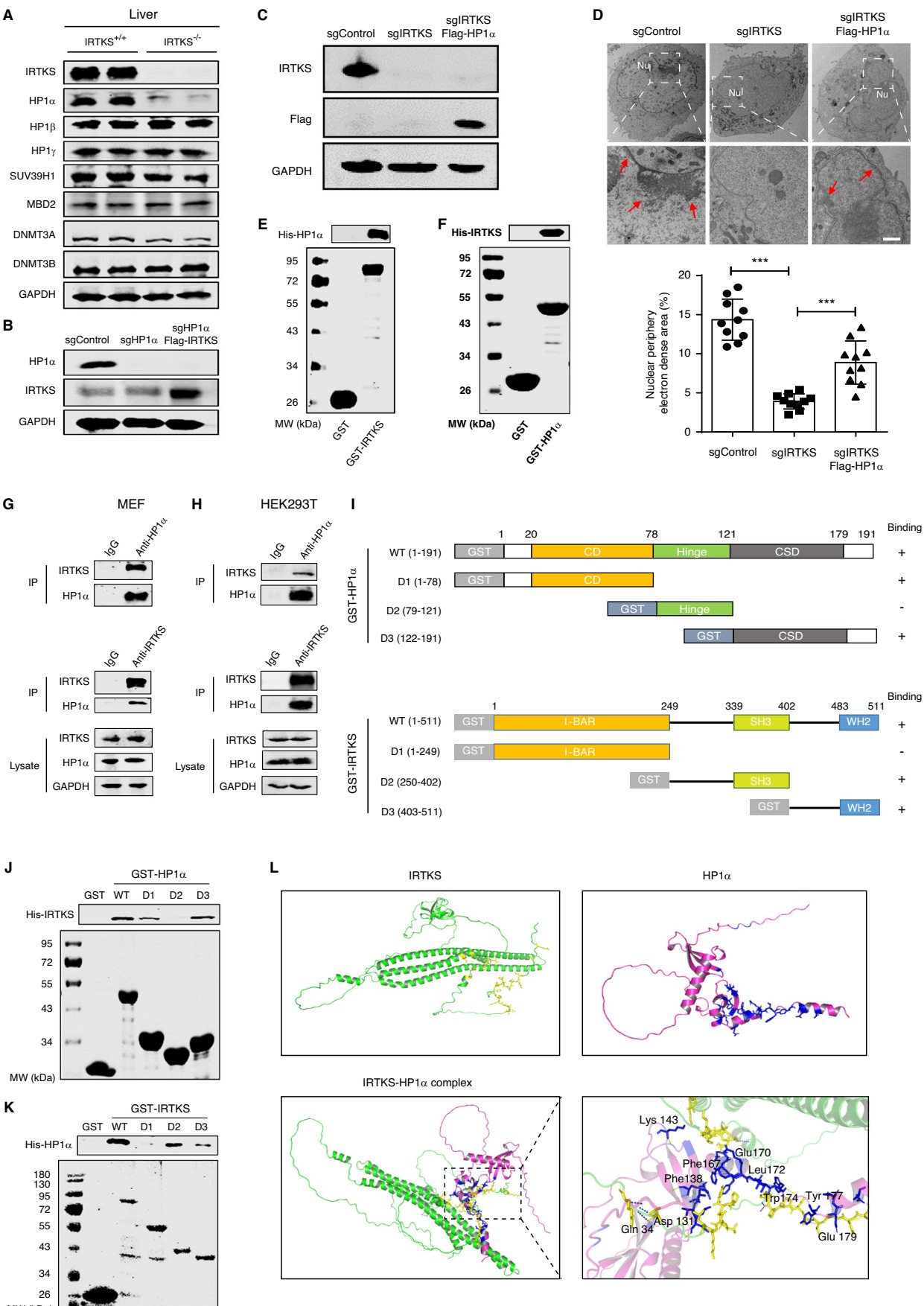

**Figure EV2. IRTKS directly associates with HP1α.**

(A) Western blotting analysis of several important epigenetic factors associated with heterochromatin formation in liver tissues from WT and *Irtks* KO mice. GAPDH was used as the loading control. (B) Western blotting analysis of HP1α and IRTKS expression in SK-Hep-1 cells treated with CRISPR/Cas9 single-guide RNA (sgRNA) lentivirus (sgHP1α), to knock out HP1α, coupled with the Flag-IRTKS construct. GAPDH was used as the loading control. (C) Western blotting analysis of HP1α and IRTKS expression in these SK-Hep-1 cells treated with CRISPR/Cas9 single-guide RNA (sgRNA) lentivirus (sgIRTKS) to knock out IRTKS, and then coupled with the Flag-HP1α construct. GAPDH was used as the loading control. (D) Electron microscopy images and quantification (bottom) of the electron-dense heterochromatin regions in SK-Hep-1 cells that were genetically engineered with CRISPR/Cas9 single-guide RNA (sgRNA) lentivirus (sgIRTKS) to knock out IRTKS, and then transfected with the Flag-HP1α construct. Red arrows indicate the electron-dense heterochromatin regions. Nu, nucleolus. $n = 10$ cells analyzed for each condition. ***$p = 7.52 \times 10^{-11}$ (sgControl *vs* sgIRTKS) and $4 \times 10^{-5}$ (sgIRTKS *vs* sgIRTKS-Flag-HP1α). Scale bar, 1 μm. (E, F) IRTKS and HP1α reciprocally interact directly in a GST pull-down assay. Equivalent amounts of His-HP1α were incubated with either GST (negative control) or GST-IRTKS. After GST pulldown, HP1α was detected by western blotting. The mirror experiment was performed using His-IRTKS and GST-HP1α. (G, H) The reciprocal interaction between IRTKS and HP1α was detected by co-immunoprecipitation (co-IP) with anti-IRTKS and anti-HP1α antibodies in MEFs (G) and HEK293T cells (H). The immunoglobulin G (IgG) group was the negative control. (I) Schematic summaries of the interactions between diverse IRTKS and HP1α truncations. (J) Direct interaction between full-length or truncated GST-HP1α proteins and His-IRTKS revealed by GST pulldown assay. Full-length and truncated GST-HP1α proteins were visualized by Coomassie blue staining. (K) GST pulldown assays were performed with recombinant His-HP1α and full-length or truncated GST-IRTKS. The pulldown samples were analyzed by western blotting. (L) A 3D structural model of the IRTKS-HP1α complex was constructed using the Z-DOCK server. The 3D structures of IRTKS and HP1α were predicted by AlphaFold algorithms (upper panel). The interaction between IRTKS and HP1α is depicted by the yellow and blue colors, respectively. Details of the key residues of HP1α that interact with IRTKS are also shown in the lower panel. Data are presented as the mean ± SD. Figure EV2D was tested by one-way ANOVA followed by Tukey's post hoc test. Source data are available online for this figure.

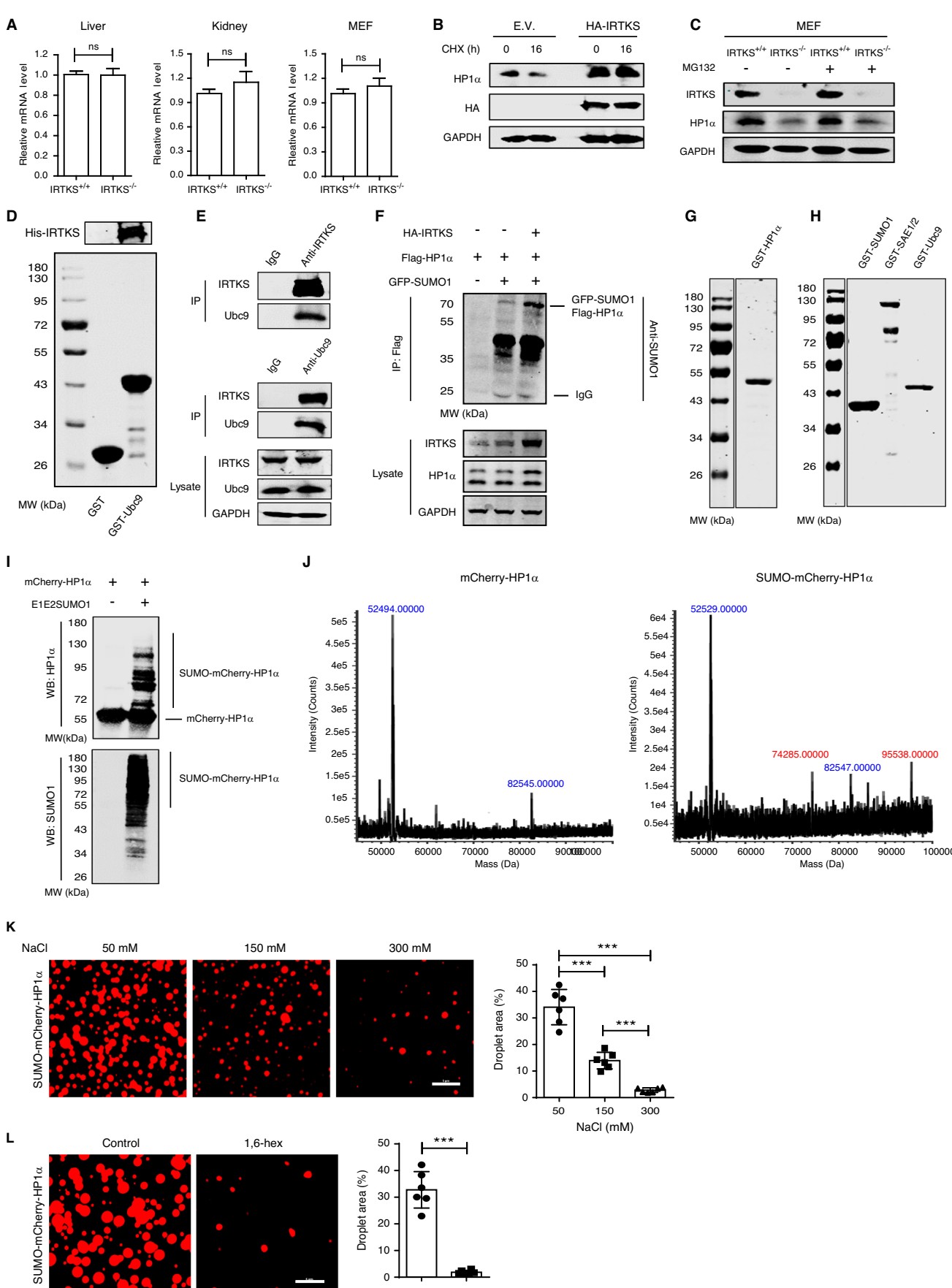

◀ **Figure EV3.  SUMOylated HP1α follows the principle of phase separation.**

(A) RT–qPCR showing RNA levels of HP1α in the livers, kidneys, and MEFs of WT and Irtks-KO mice. Relative RNA levels are normalized to GAPDH. $n = 3$ biological replicates. ns = 0.9397 (liver), 0.3482 (kidney), and 0.4195 (MEF). (B) Western blotting analysis showed the stability of HP1α in HEK293T cells that were transfected with empty vector or Flag-IRTKS construct after treatment with the protein synthesis inhibitor cycloheximide (CHX, 100 μg/ml) at the indicated time. (C) The HP1α level was not restored by MG132 (10 μM, 9 h), a proteasome inhibitor, in MEFs. (D) IRTKS can directly interact with Ubc9 in a GST pull-down assay in vitro. (E) The reciprocal interaction between IRTKS and Ubc9 was detected by co-immunoprecipitation (co-IP) with anti-IRTKS or anti-Ubc9 antibodies in HEK293T cells. The immunoglobulin G (IgG) group was the negative control. (F) The SUMO-1-mediated SUMOylation of HP1α was obviously enhanced by the co-expressed IRTKS in NIH3T3 cells, as detected by immunoprecipitation assay. NIH3T3 cells were transfected with HA-IRTKS, Flag-HP1α, and GFP-SUMO1, and then were immunoprecipitated with an anti-Flag antibody for the SUMOylation assay, followed by western blotting with the indicated antibodies. (G, H) Coomassie blue-stained images of purified GST-HP1α (G), GST-SUMO1, SAE1/2, and Ubc9 (H). (I) Western blotting analysis showing that mCherry-HP1α protein was SUMOylated by coexpression with E1E2SUMO1. (J) Results of Acquity UPLC I-class/VION IMS QTOF analysis for mCherry-HP1α and SUMO-mcherry-HP1α. The observed masses of mCherry-HP1α and SUMO-mCherry-HP1α proteins are also shown. The observed masses of SUMO-mCherry-HP1α proteins are increased compared with that of unmodified mCherry-HP1α. (K) Representative images and quantification of droplet formation at various salt concentrations. SUMO-mCherry-HP1α was added to droplet formation buffer to achieve a 20 μM protein concentration with a final NaCl concentration as indicated ($n = 8$ fields for each group were quantified). ***$p = 6.6 \times 10^{-7}$ (50 mM *vs* 150 mM), $2.03 \times 10^{-9}$ (50 mM *vs* 300 mM), and $4.21 \times 10^{-4}$ (150 mM *vs* 300 mM). Scale bar, 5 μm. (L) The droplet formation ability of SUMO-mCherry-HP1α was significantly depressed by 5% 1,6-hexanediol treatment. $n = 8$ fields for each group were quantified. ***$p = 9.36 \times 10^{-5}$. Scale bar, 5 μm. Data are presented as the mean ± SD or mean ± SEM. Figure EV3K was tested by one-way ANOVA followed by Tukey's post hoc test. The remaining plots were tested by two-tailed Student's t test. Source data are available online for this figure.

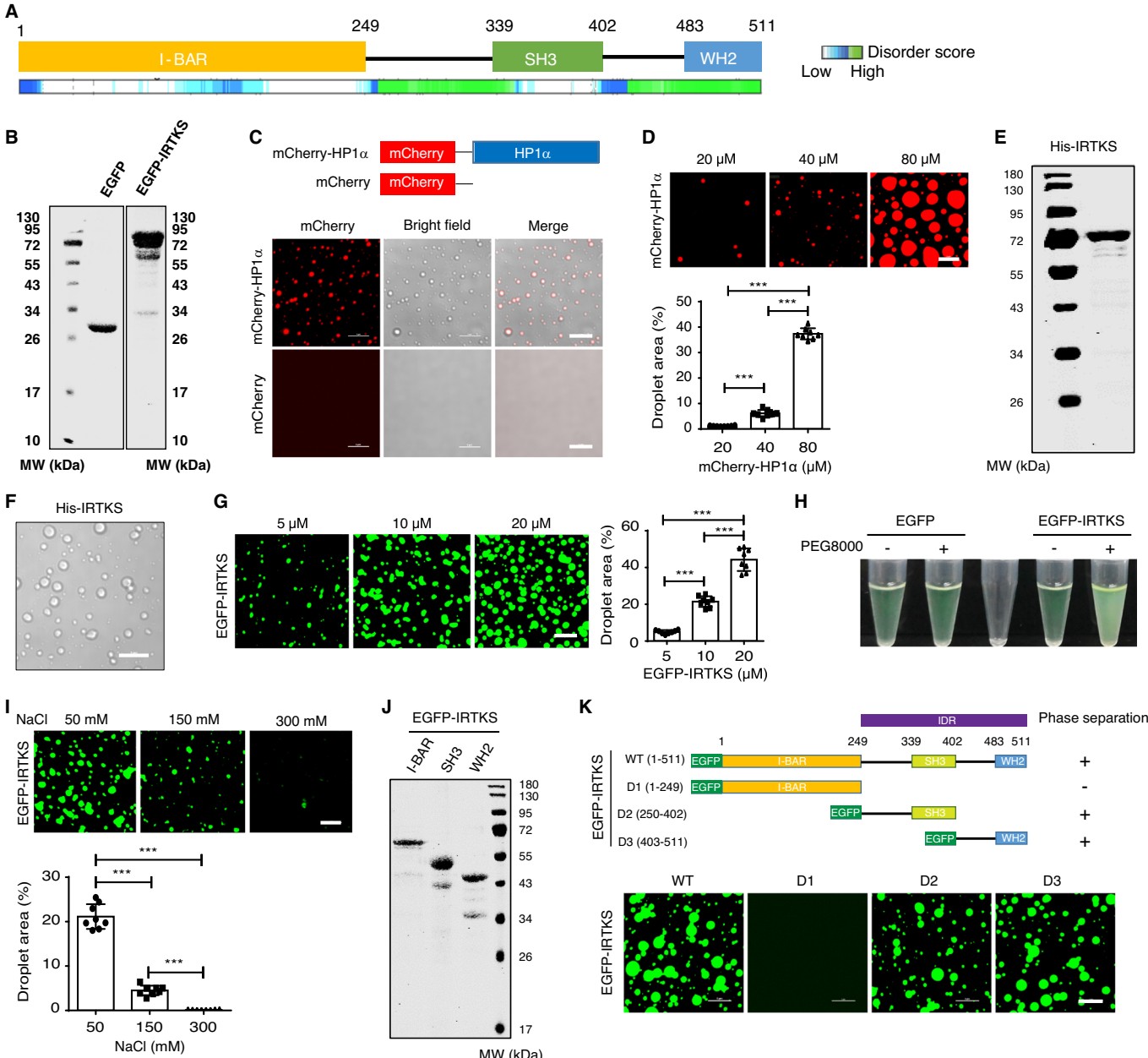

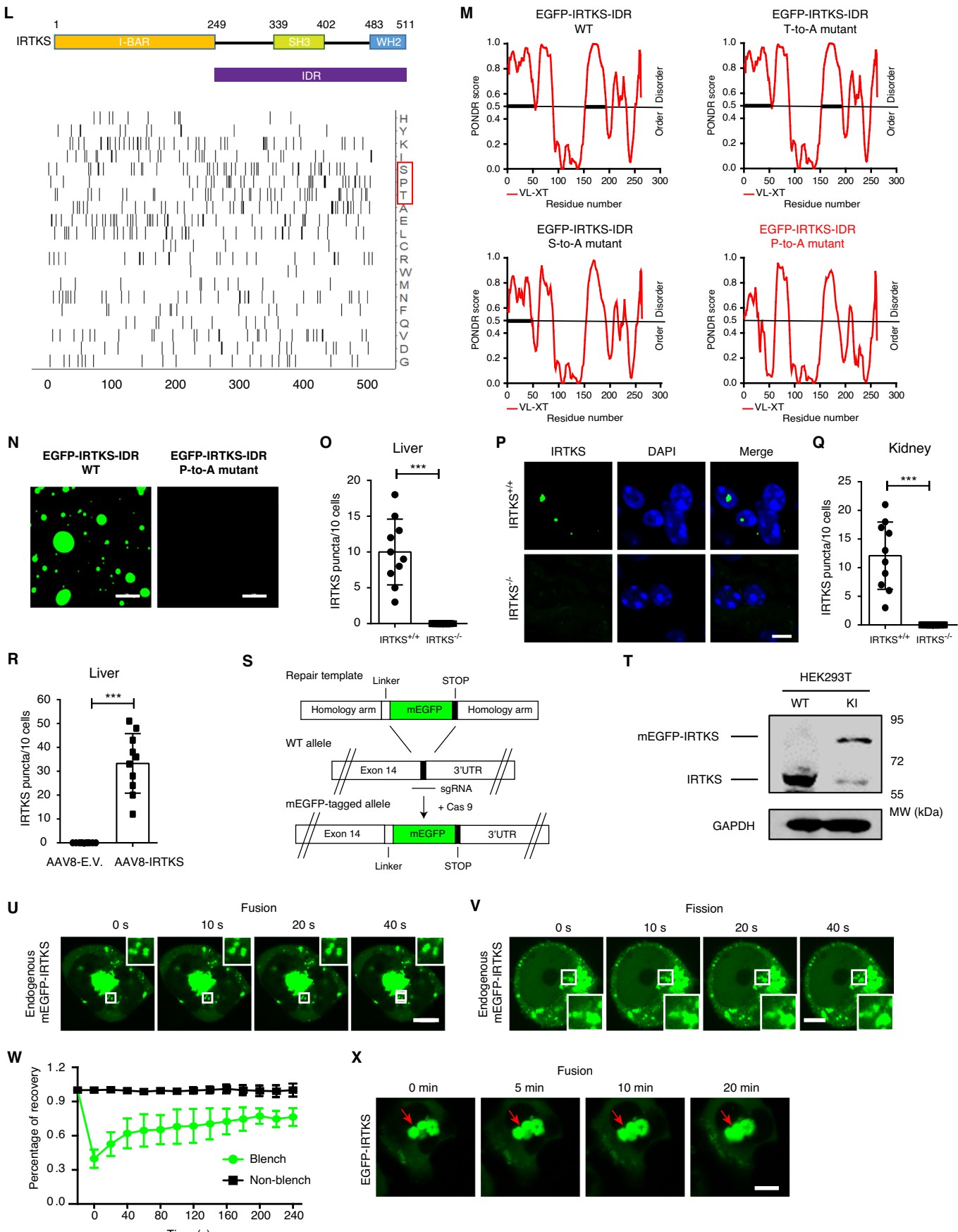

◀

**Figure EV4.   IRTKS possesses phase-separated properties.**

(A) Schematic diagram of IRTKS domains and intrinsically disordered region (IDR) prediction by the $D^2P^2$ algorithm. (B) Coomassie blue staining image of purified EGFP and EGFP-IRTKS proteins. (C) Representative images of droplet formation showing that mCherry-HP1α has liquid-like properties. mCherry and mCherry-HP1α were added to droplet formation buffer to 40 μM. Scale bar, 5 μm. (D) In vitro droplet assay of mCherry-HP1α at the indicated concentrations ($n = 8$ fields for each group were quantified). ***$p = 1.55 \times 10^{-6}$ (20 μM vs 40 μM), $6.48 \times 10^{-23}$ (20 μM vs 80 μM), and $1.42 \times 10^{-21}$ (40 μM vs 80 μM). Scale bar, 5 μm. (E) Coomassie blue staining image of purified His-IRTKS. (F) Representative DIC images of droplet formation showing that His-IRTKS has liquid-like properties. His-IRTKS was added to droplet formation buffer to 20 μM. Scale bar, 5 μm. (G) Representative images and quantification of droplet formation at various protein concentrations. EGFP-IRTKS was added to droplet formation buffer to final concentrations as indicated ($n = 8$ fields for each group were quantified). ***$p = 4.13 \times 10^{-8}$ (5 μM vs 10 μM), $4.29 \times 10^{-15}$ (5 μM vs 20 μM) and $1.55 \times 10^{-10}$ (10 μM vs 20 μM). Scale bar, 5 μm. (H) Visualization of turbidity associated with droplet formation. Tubes containing EGFP (left pair) and EGFP-IRTKS (right pair) in the presence (+) or absence (–) of PEG-8000 are shown. Blank tubes are included between pairs for contrast. (I) Representative images and quantification of droplet formation at various salt concentrations. EGFP-IRTKS was added to droplet formation buffer to achieve a 10 μM protein concentration with a final NaCl concentration as indicated ($n = 8$ fields for each group were quantified). ***$p = 7.87 \times 10^{-15}$ (50 mM vs 150 mM), $6.92 \times 10^{-17}$ (50 mM vs 300 mM), and $4.65 \times 10^{-5}$ (150 mM vs 300 mM). Scale bar, 5 μm. (J) Coomassie blue staining image of purified EGFP-IRTKS truncations (I-BAR, SH3, and WH2). (K) A schematic summary of the droplet formation ability of full-length or truncated EGFP-IRTKS and representative images of droplet formation of various EGFP-IRTKS truncations. Scale bar, 5 μm. (L) Heatmap analyzing the amino acid composition and position of IRTKS. Each row represents information for a single amino acid. The length of the row corresponds to the length of the IRTKS protein. The purple bar represents the IDR of IRTKS shown in Extended Data Fig. 4a. (M) Predictions of IDRs of IRTKS with mutation of all prolines (P), serines (S) or threonines (T) to alanine (A) using the PONDR algorithm. (N) Mutating all prolines to alanine (P to A) disrupts phase separation. Representative images of droplet formation by wild-type IRTKS-IDR or the IRTKS-IDR P-to-A mutant fused to EGFP. Scale bar, 5 μm. (O) Quantification of IRTKS puncta number per 10 cells was analyzed in the livers from WT and *Irtks* KO mice. $n = 10$ for each group. ***$p = 7.21 \times 10^{-5}$. (P, Q) Immunofluorescent staining (P) of IRTKS (green) and DAPI (blue) and quantification (Q, ***$p = 1.1 \times 10^{-4}$) of IRTKS puncta number per 10 cells in kidney sections of WT and *Irtks* KO mice. Scale bar, 5 μm. $n = 10$ for each group. (R) Quantification of IRTKS puncta number per 10 cells was analyzed in the livers of these mice infected with AAV8-IRTKS and empty vector AAV8 (AAV-E.V.) as control. ***$p = 1.43 \times 10^{-5}$. $n = 10$ for each group. (S) Schematic of the strategy used to generate endogenously mEGFP-tagged IRTKS HEK293T cells. (T) Western blotting analysis of HEK293T cells with mEGFP knock-in at the endogenous IRTKS locus. (U) Time-lapse fluorescence images showing that the endogenous mEGFP-tagged IRTKS puncta rapidly fused in HEK293T cell. The fused puncta are boxed and enlarged in the images on the right. Scale bar, 5 μm. (V) Live-cell imaging of endogenous mEGFP-tagged IRTKS puncta in HEK293T cells. The white box indicated the fission events of IRTKS puncta and enlarged in the images on the right. Scale bar, 5 μm. (W) The fluorescence recovery curves of endogenous mEGFP-tagged IRTKS puncta shown by FRAP experiments ($n = 8$ biological replicates for the FRAP curve construction). (X) Fusion events of droplets over time indicate liquid-like material properties of EGFP-IRTKS in HEK293T cells. Red arrows indicate fusion events. Scale bar, 10 μm. Data are presented as the mean ± SD. Figures EV4O, Q and R were tested by two-tailed Student's t test. The remaining plots were tested by one-way ANOVA followed by Tukey's post hoc test. Source data are available online for this figure.

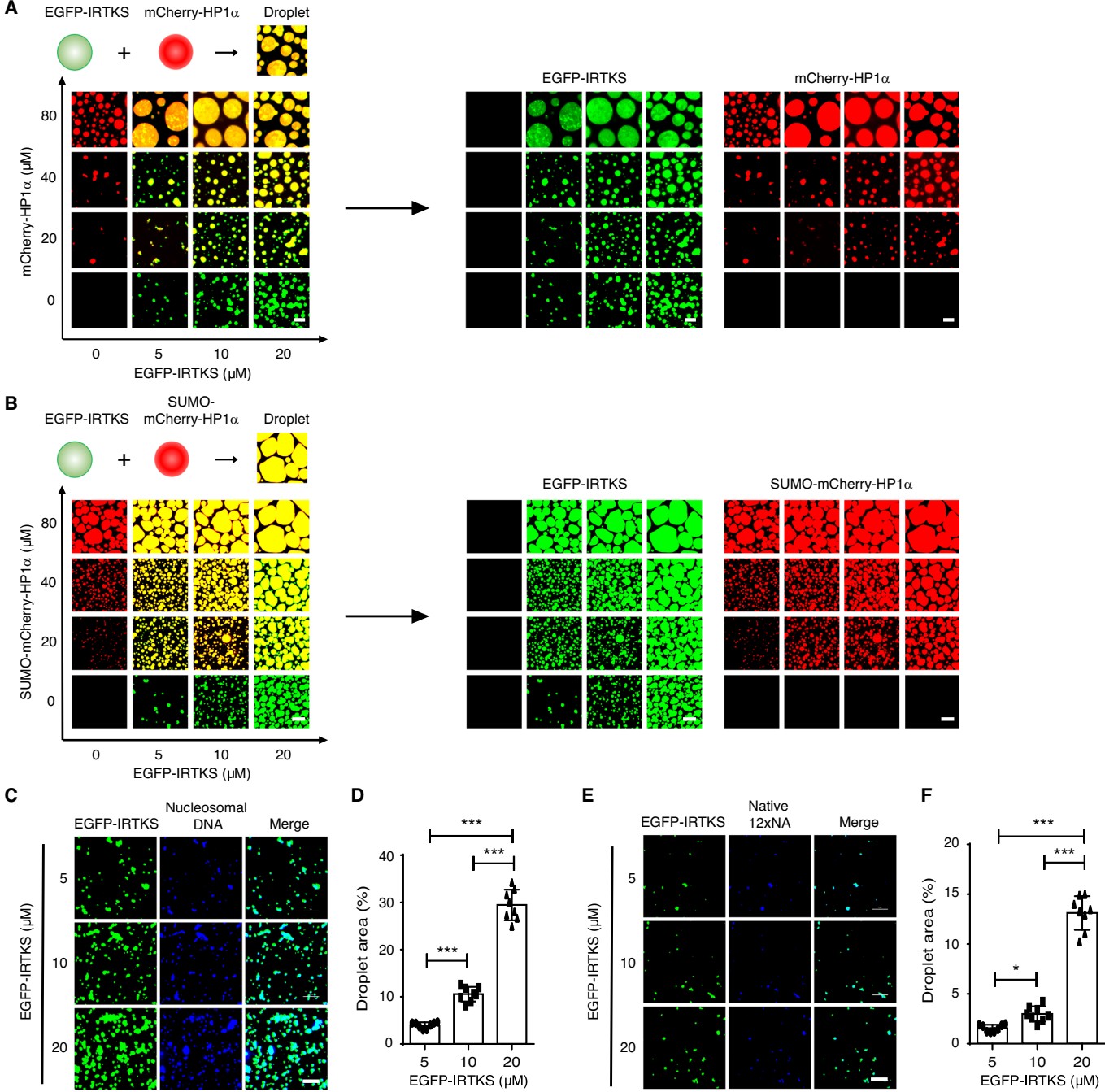

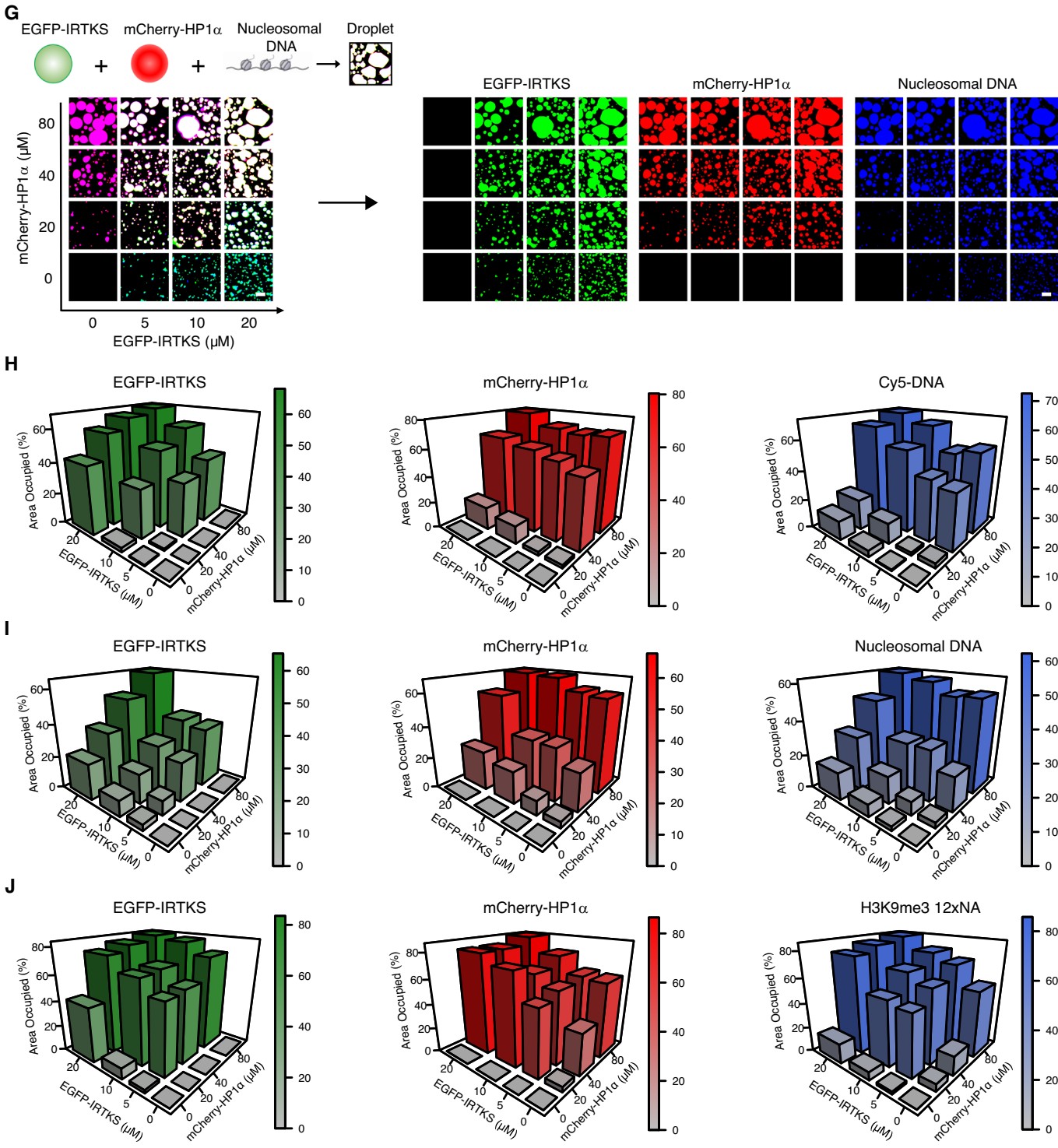

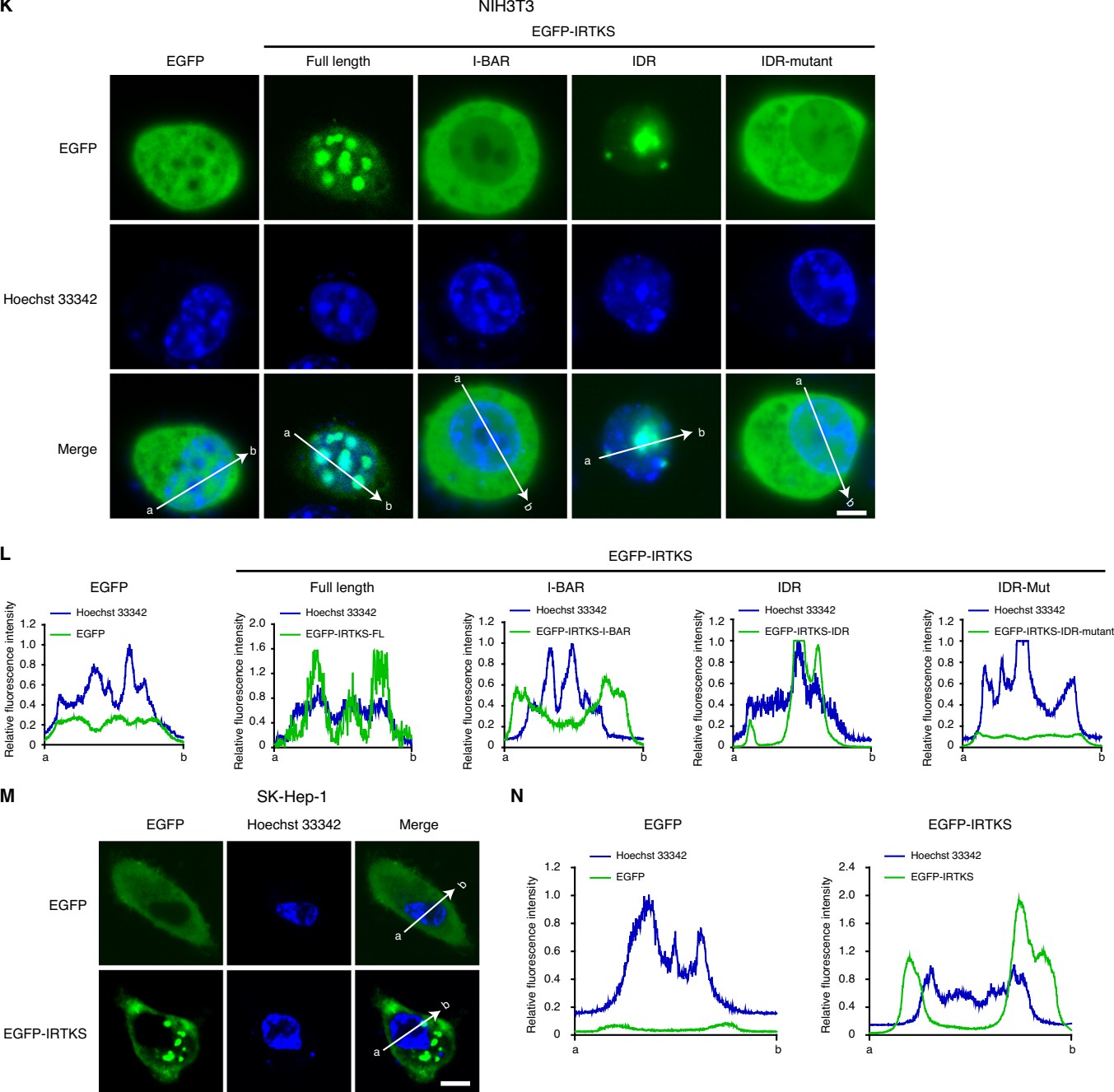

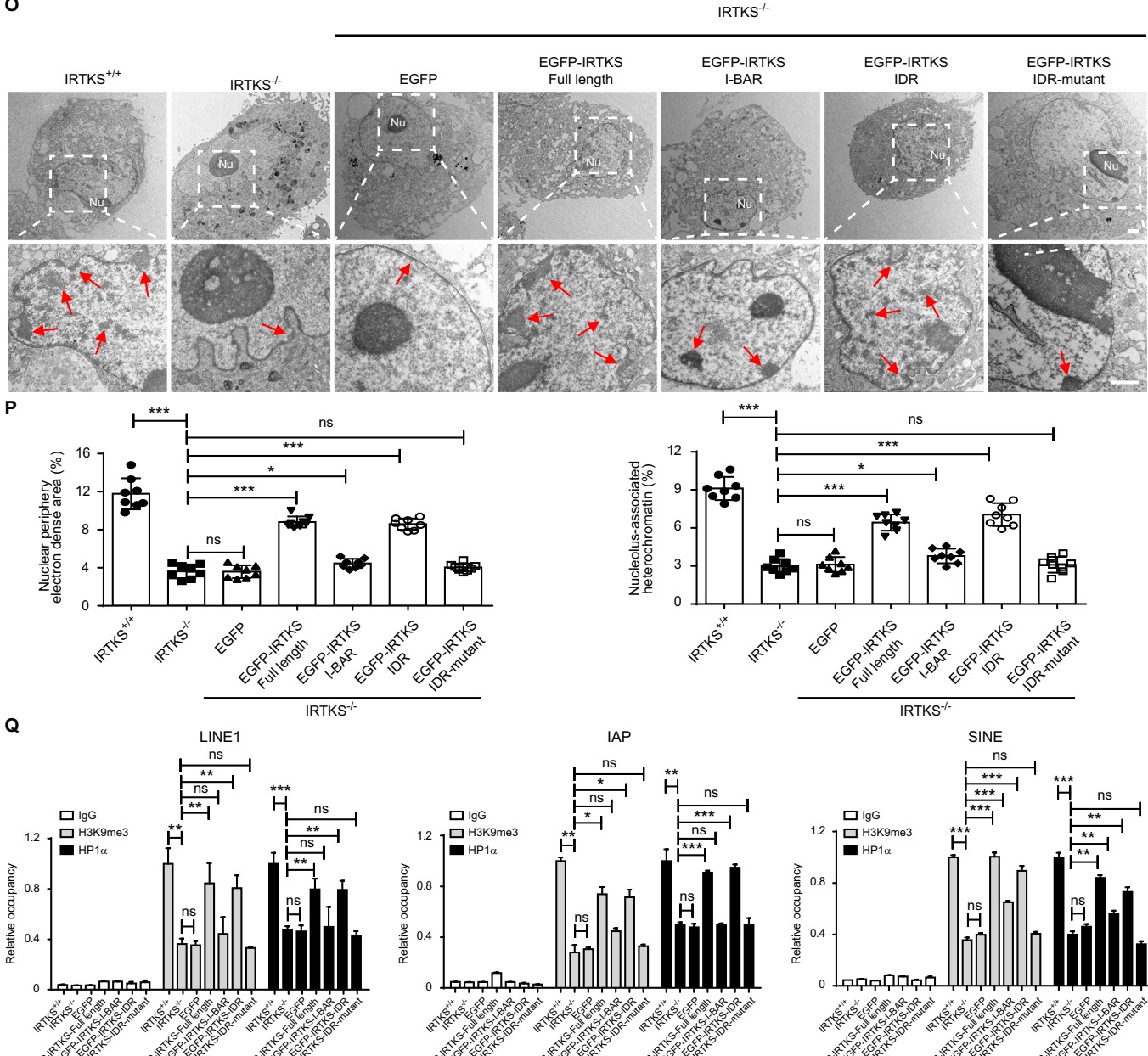

◀ **Figure EV5. Liquid–liquid phase separation of IRTKS and HP1α droplets with diverse DNA-containing substrates.**

(A) Representative images of droplet formation at different concentrations of IRTKS and HP1α protein. Concentrations of IRTKS and HP1α are indicated at the bottom and left of the images, respectively. Scale bars, 5 μm. (B) Representative images of droplet formation at various concentrations of EGFP-IRTKS and SUMO-mCherry-HP1α protein. Concentrations of EGFP-IRTKS and SUMO-mCherry-HP1α are indicated at the bottom and left of the images, respectively. Scale bars, 10 μm. (C, D) In vitro phase separation assay of EGFP-IRTKS protein at various concentrations mixed with nucleosomal DNA. A total of 6 nM nucleosomal DNA for the droplet assay was stained using DAPI ($n = 8$ fields for each group were quantified). ***$p = 3.3 \times 10^{-6}$ (5 μM *vs* 10 μM), $8.34 \times 10^{-17}$ (5 μM *vs* 20 μM), and $3.42 \times 10^{-14}$ (10 μM *vs* 20 μM). Scale bar, 5 μm. (E, F) Droplet formation of various concentrations of EGFP-IRTKS protein mixed with reconstituted native 12× nucleosomal arrays (NA). A total of 330 nM reconstituted native 12× NA for the droplet assay was stained using DAPI ($n = 8$ fields for each group were quantified). *$p = 0.0162$ (5 μM *vs* 10 μM), ***$p = 1.22 \times 10^{-15}$ (5 μM *vs* 20 μM), and $1.71 \times 10^{-14}$ (10 μM *vs* 20 μM). Scale bar, 5 μm. (G) Liquid–liquid phase separation assay with nucleosomal DNA to examine the ability of IRTKS to form condensates with HP1α and nucleosomal DNA stained using DAPI. Scale bar, 5 μm. (H–J) A phase diagram of IRTKS and HP1α mixed with Cy5-labeled DNA oligos (H), nucleosomal DNA (I), and reconstituted H3K9me3 12× NA (J). $n = 8$ fields for each group were quantified. (K, L) Representative images (K) and line scan analysis (L) of EGFP or EGFP-IRTKS with different truncations (full-length, I-BAR, IDR and IDR-mutant) in NIH3T3 cells. Nuclei were labeled with Hoechst 33342. Quantification of lines scanned across EGFP or EGFP-IRTKS with different truncations and nuclei at the position depicted by the white arrow. Scale bar, 2 μm. (M, N) Live-cell images (M) and line scan analysis (N) of EGFP or EGFP-IRTKS-expressing SK-Hep-1 cells. Nuclei were labeled with Hoechst 33342. Quantification of lines scanned across EGFP or EGFP-IRTKS and nuclei at the position depicted by the white arrow. Scale bar, 5 μm. (O, P) Electron microscopy images (O) and quantification of the electron-dense heterochromatin regions (P) in MEF cells overexpressed EGFP or EGFP-IRTKS with different truncations (full-length, I-BAR, IDR, and IDR-mutant). Red arrows indicate the electron-dense heterochromatin regions. Nu, nucleolus. $n = 8$ cells analyzed for each condition. Scale bar, 1 μm. (Q) Enrichment of H3K9me3 and HP1α within the regions of repetitive sequences (LINE1, IAP, and SINE) in the MEFs of WT and *Irtks* KO mice as measured by ChIP-qPCR. $n = 3$ technical replicates in independent experiments. The p values of Fig. EV5P and Q were provided in Source data. Data are presented as the mean ± SD or mean ± SEM and the p value of one-way ANOVA followed by Tukey's post hoc test. Source data are available online for this figure.

