## [Peer Review File · The EMBO Journal]

Heterochromatin formation and remodeling by IRTKS condensates counteract cellular senescence

Zeguang Han, Jia Xie, Zhao-Ning Lu, Shi-Hao Bai, Xiao-Fang Cui, He-Yuan Lian, Chenyi Xie, Na Wang, and Lan Wang

Corresponding author(s): Zeguang Han (hanzg@sjtu.edu.cn)

Review Timeline:

Submission Date:	4th Apr 24
Editorial Decision:	5th Jun 24
Revision Received:	16th Jun 24
Editorial Decision:	25th Jun 24
Revision Received:	30th Jun 24
Accepted:	16th Jul 24

Editor: *Cornelius Schneider*

Transaction Report:

This manuscript was transferred to The EMBO JOURNAL following peer review at another journal.

The point-by-point response to the editor and referee comments

We thank all the reviewers for their thorough analysis on our manuscript, and these constructive comments are very helpful to improve our study. In the revised version, we have endeavored to address all concerns raised by all reviewers with additional experiments, reanalysis of data and textual changes.

Before we provide the point-by-point response to all reviewers in detail, we highlight the major revisions as the editor suggested:

First-round review

A) Experimentally address concerns about IRTKS perturbations on global vs. local changes to heterochromatin (Reviewer #1 and Reviewer #3).

R: To address the concerns raised by Reviewers #1 and #3, we re-analyzed the transmission electron microscopy (TEM) images of livers, kidneys, stomachs, MEFs from WT and *Irtks* knockout mice, as well as human SK-Hep-1 cells with or without IRTKS, and then quantified the heterochromatin domains (**Fig. 1b, d, f, Extended Data Fig. 1a, b, d-f, h, i, k, l, o and p**), which are located at the nuclear periphery or around the nucleolus. The new results, combined with the observation and analysis in the previous version, indicate that IRTKS could globally regulate heterochromatin domains.

B) Carefully characterize biomolecular condensation in-cell with further experiments (Reviewer #1 and Reviewer #3).

R: In the revised version, we further performed the fluorescence recovery after photobleaching (FRAP) experiments to examine the dynamic feature of IRTKS puncta (**Fig. 3k and Extended Data Fig. 4w**). In addition, we also employed the time-lapse microscopy to explore the fusion and fission events of IRTKS puncta within cells (**Extended Data Fig. 4u, v**). These new data, as response to the concerns raised by Reviewers #1 and #3, further strengthen the characteristics of IRTKS condensates within cells.

C) Address with new experiments whether there are different changes in chromatin accessibility based on different cell types, such as with primary cells (Reviewer #2), as well as account for differences in mouse vs. human cell lines in heterochromatin (Reviewer #3).

R: As suggested by Reviewer #2, we further performed ATAC-seq to evaluate the genome-wide changes in livers from WT and *Irtks* KO mice (**Fig. 5f, Extended Data**

Fig. 6m, n and q), the new data suggest that the changing chromatin accessibility based on different cell types, including mouse livers and MEFs, was similar. In addition, to address the concerns raised by Reviewer #3, we also performed a number of new experiments to investigate the role of IRTKS in heterochromatin formation in mouse and human cell lines, including: (1) Observing the live-cell fluorescence imaging and fluorescence recovery after photobleaching (FRAP) experiments using enhanced green fluorescent protein (EGFP)-tagged HP1 α in human liver endothelial (SK-Hep-1) cells and mouse embryonic fibroblast (MEF) cells (**Extended Data Fig. 1w, y**); (2) Analyzing the DAPI-stained regions using Hoechst 33342 in SK-Hep-1 cells and MEF cells, respectively (**Extended Data Fig. 1v, x**); (3) Performing *in vivo* SUMOylation experiments in human HEK293T cells and mouse NIH3T3 cells, respectively (**Fig. 2b and Extended Data Fig. 3f**). These data indicate that the role of IRTKS in heterochromatin formation was similar and conserved in mouse and human cell lines.

D) Perform appropriate controls, quantifications, and statistical analyses of the sequencing datasets (Reviewer #2 and Reviewer #3).

R: In the light of the Reviewers #1 and #3, we provided more detail about the quantifications and statistical analysis of the sequencing datasets (**Fig. 5a, b, f, g, Extended Data Fig. 6a, b, h-k and n-r**), as well as the additional appropriate control (**Extended Data Fig. 6l, m**).

E) Assess localization and effects of IRTKS on HP1 α localization (Reviewer #1 and Reviewer #3)) and expression (Reviewer #2), controlling for potential pleiotropic effects of IRTKS perturbation given its known effects on multiple cellular activities (Reviewer #3).

R: To address the concerns raised by Reviewers #1 and #3, we conducted a kind of additional experiments to define the location of IRTKS (**Fig. 3k, Extended Data Fig. 4u and v**), and then evaluate the effect of IRTKS on HP1 α location, including: (1) Observing the HP1 α effect on heterochromatin formation by TEM, as *IRTKS* was knocked out in SK-Hep-1 cells using the CRISPR/Cas9 system (**Extended Data Fig. 2d**); (2) Re-analyzing statistically the ChIP-seq data with anti-HP1 α antibody of livers and MEF cells from WT and *Irtks* KO mice (**Fig. 5b, Extended Data Fig. 6b, i and k**); (3) Transfecting MEF cells with EGFP-tagged HP1 α to assess the IRTKS-mediated HP1 α dynamic changes by using FRAP experiments, where we analyzed the location of HP1 α puncta in the nucleus of SK-Hep-1 and MEF cells, respectively (**Extended Data Fig. 1v-y**). To address the effect of IRTKS on HP1 α expression, we further performed *in vivo* SUMOylation experiment to investigate the effect of IRTKS on HP1 α

SUMOylation in NIH3T3 cells (**Extended Data Fig. 3f**), suggesting that IRTKS-mediated HP1 α SUMOylation was conserved in mouse and human cells. These direct evidences derived from these new and previous data may exclude the possibility that the known pleiotropic effects of IRTKS perturbation could be involved in HP1 α and heterochromatin.

F) Appropriately cite the known literature on IRTKS and HP1alpha that could have effects independently of any direct causal relationship and independently of biomolecular condensation (Reviewer #3).

R: In the revised version, as Reviewer #3 suggested, we cite some known references about IRTKS and HP1 α , and then discuss the relationship between them and independently of biomolecular condensation.

G) All other referee concerns pertaining to strengthening existing data, providing controls, methodological details, clarifications and textual changes, should also be addressed.

R: In the revised version, we address all other concerns to strengthen our data (see point-by-point response below). We also carefully checked the manuscript thoroughly, and rectify some inaccuracies we made, providing controls, methodological details, clarifications and textual changes. These changes are highlighted in blue in the revision.

Second-round review

1. As suggested by Reviewer #4, we re-performed the ChIP-seq data and ATAC-seq data analysis with setting false discovery rate (**Fig. 5a, b, f, Extended Data Fig. 6a, b, h-k and n-q**), and modified some sentences for clear description in the corresponding Methods section.

2. In response to Reviewer #4, we re-analyzed the ATAC-seq data to confirm the impact of IRTKS on the chromatin accessibility of RE regions. Additionally, we conducted the statistical analyses on the chromatin accessibility of *Actb* gene as a control locus, ruling out the experimental artefact (**Fig. 5f, Extended Data Fig. 6m, n**).

3. In addressing the concerns raised by Reviewer #4, We conducted more additional experiments to investigate the significance of the liquid-liquid phase separation (LLPS) capability of IRTKS in the regulation of heterochromatin, including: (1) Transfecting mouse fibroblast NIH3T3 cells and human liver endothelial SK-Hep-1 cells with EGFP or EGFP-tagged IRTKS to investigate whether IRTKS puncta are

localized in heterochromatin regions (**Extended Data Fig. 5k-n**); (2) Observing the live-cell fluorescence imaging utilizing the enhanced green fluorescent protein (EGFP)-tagged IRTKS truncations containing I-BAR, IDR and IDR-mutant (P to A mutant of IRTKS-IDR) in NIH3T3 cells, and verifying the localization of IRTKS puncta in Hoechst-dense regions using Hoechst 33342 through line scan analysis (**Extended Data Fig. 5k, l**); (3) Performing a rescue experiment by overexpressing the EGFP-tagged IRTKS with various truncations (full-length, I-BAR, IDR and IDR-mutant) or EGFP control in MEF cells lacking endogenous IRTKS to survey the effect of IRTKS puncta on the status of heterochromatin by transmission electron microscopy (TEM) (**Extended Data Fig. 5o, p**); (4) Performing chromatin immunoprecipitation-quantitative polymerase chain reaction (ChIP-qPCR) to examine the roles of IRTKS condensates in regulating heterochromatin-associated repetitive sequences (**Extended Data Fig. 5q**).

4. As suggested by Reviewer #4, we performed the live-cell imaging and line scan analysis to verify the accumulation of IRTKS condensates within the heterochromatin foci in both NIH3T3 cells and SK-Hep-1 cells (**Extended Data Fig. 5k-n**).

5. In response to Reviewer #1, we have carefully checked the manuscript thoroughly, and rectify any inaccuracies we made. And the changes are highlighted in blue in the revision.

Point-by-point response to Reviewers

Reviewer: 1

First-round review

Remarks to the Author:

The manuscript by Xie, et al., proposes that insulin receptor tyrosine kinase substrate (IRTKS) undergoes liquid-liquid phase separation (LLPS) in cells to promote heterochromatin formation. The authors first showed that knock out of IRTKS resulted in depletion of electron-dense regions at the nuclear periphery, as visualized by TEM, which is an indicator of loss of heterochromatin. Using immunofluorescence, they showed that IRTKS co-localizes with known heterochromatin markers, HP1 α and H3K9me3. Further, the authors showed that loss of IRTKS results in reduction of HP1 α protein levels and they determined that this is due to loss of recruitment of Ubc9 by IRTKS to SUMOylate and stabilize HP1 α . They also showed that SUMO-HP1 α displayed a greater propensity for LLPS, which has been shown to be important for heterochromatin formation and maintenance. Using purified protein in vitro, the authors determined that IRTKS can also undergo condensation and LLPS. They further showed

that in cells IRTKS also displayed condensation behavior, although these data are questionable (as discussed below). Importantly, using in vitro droplet experiments, the authors showed that IRTKS, HP1 α , and various forms of DNA/chromatin could all co-phase separate within droplets, suggesting a LLPS-dependent role for IRTKS-assisted heterochromatin formation. To show functional consequences of loss of IRTKS-assisted heterochromatin formation, the authors probed repetitive element (RE) sequence region occupancy of HP1 α and H3K9me3 and determined that the heterochromatin marks were significantly depleted in these REs. RNAseq analysis showed that this resulted in transcription of these normally repressed regions and subsequent activation of cellular senescence-associated pathways, suggesting a new role for IRTKS in countering senescence. The manuscript is well-written and the data presented are of generally high quality and definitive. The findings presented are novel and will be of interest to the cell biology community. With revision, the manuscript may be a strong candidate for consideration for publication at Nature Cell Biology.

R: We sincerely appreciate the reviewer's comprehensive summary and thank for the helpful and constructive comments on our findings. Here we provide our full point-by-point responses below.

Main comments:

1. Heterochromatin visualization by TEM:

a. Figure 1A-F shows loss of peripheral heterochromatic regions upon IRTKS knockout, but immunofluorescence images show that IRTKS is distributed throughout entire nuclei, similar to HP1 α and H3K9me3. Can the authors comment on why only heterochromatin at the periphery is lost, when IRTKS is localized throughout the nucleus. Shouldn't its effects be seen throughout heterochromatin globally?

R: We appreciate the professional comment and suggestion of the reviewer. As correctly pointed by the reviewer, IRTKS is distributed throughout entire nuclei, similar to HP1 α and H3K9me3. In the revised version, as the reviewer suggested, in addition to heterochromatin domains at the nuclear periphery, we also focused on the changes of heterochromatin domains at other regions within nuclei, especially around the nucleolus.

It is believed that heterochromatin domains can be detected as electron-dense regions (EDRs), which are located at the nuclear periphery or around the nucleolus, by transmission electron microscopy (TEM) (PMIDs: 36533441, 33307247, 30140741 and 25479748). To address this point, we re-analyzed the TEM images of livers, kidneys, stomachs and MEFs from WT and *Irtks* knockout mice, and human SK-Hep-1 cells with or without *IRTKS*, and then quantified the heterochromatin domains based on EDRs around the nucleolus. Interestingly, the quantificational analysis showed that, in the absence of *Irtks*, the heterochromatic regions around the nucleolus were dramatically reduced in these mouse livers, kidneys and stomachs. Consistently, we also observed a slight loss of heterochromatin around the nucleolus in the IRTKS-KO MEF and SK-Hep-1 cells, respectively. Conversely, ectopic IRTKS overexpression

increased the EDRs of heterochromatin around the nucleolus in these MEFs and SK-Hep-1 cells. These new results, combined with our previous data, suggest that IRTKS could globally regulate heterochromatin domains.

Accordingly, in the revised version, we added these new results as **Extended Data Fig. 1a, b, e, f, i, l and p**, and modified some sentences for clear description in the corresponding Results section (**page 5, line 25; page 6, lines 1-14**).

b. Additionally, Figure 2A shows TEM images upon HP1 α knockout but the overall conclusions stated by the authors are not fully supported by the provided images. While the zoomed inset from the sgControl (left) and sgHP1 α +Flag-IRTKS (right) conditions illustrate long regions of nuclear membrane to assess, the sgHP1 α condition (center) illustrates very little nuclear periphery in comparison. Can the authors include a different sgHP1 α image that shows more uninterrupted nuclear periphery for inspection?

R: We are grateful for the professional and valuable suggestion. According to the suggestion, in the new version, we display a new sgHP1 α image (**Fig. 2a**) that shows more continuous nuclear periphery, which fully conforms our overall conclusion.

2. Quantitation:

a. The authors quantified many of their imaging results but do not fully explain how this was done in the Methods section. This applies to result from the droplet experiments shown in the following figures: Figure 2D, E; Figure 3D; Figure 4B, D. How was droplet area determined? What programs were used for these analyses?

R: Many thanks for the professional suggestion, and we apologize for not providing sufficient methodological detail in the previous version of our manuscript. In the revised version, we have carefully gone through the manuscript, and explained in detail how droplet area was determined and describe the programs to analyze them in the Methods section (**page 41, lines 4-8**).

b. In Figure 3H, the authors conclude that endogenous IRTKS forms puncta in the nucleus and cytoplasm, but the image includes only a single nuclear punctum and single cytoplasmic punctum. This statement should be supported by quantitative data. The authors should quantify the number of puncta across a population of cells. At present, the presented data do not support the authors' statement.

R: We appreciate the important suggestion from the reviewer. As suggested by the reviewer, we quantified IRTKS puncta number in the nucleus and cytoplasm per 10 cells of liver and kidney from WT mice as well as liver from mice with adeno-associated virus serotype 8 (AAV8) vector containing ZsGreen-tagged IRTKS (AAV8-IRTKS). In the revised version, we added these new results as **Extended Data Fig. 4o, q and r**. These quantitative data fully support our conclusion.

3. The in cell LLPS data presented by the authors is questionable. Figure 3 includes

images of endogenous IRTKS as well as over-expressed IRTKS. It is notable that the structures shown in Figure 3, I-M are not demonstrably round as is seen with condensates formed through LLPS. The authors should be cautious to not over-interpret their cellular data and they should comment on the irregular nature of the structures seen in many of their cellular images. Irregular structures are also clear in Figure 1, I and K.

R: We thank the reviewer for these insightful questions and professional suggestions.

- 1) We fully agree with the reviewer that typical liquid droplets are considered to be spherical condensates, however, we also note that there are a number of studies showing aspherical morphology within the complex cellular environment, for instance, those formed by 53BP1 (PMID: 35042897), NLRP6 (PMID: 34678144), FXR1 (PMID: 32328638), SAFB (PMID: 31677973) and TIS granules (PMID: 33650968). Therefore, the aspherical structures of IRTKS could not be something that is totally unexpected, and we assume that they might be formed by the following possibilities: (1) anisotropic protein-protein (eg: IRTKS-HP1 α) or protein-nucleotide complexes that are probably driven by heterotypic electrostatic interactions among these different molecules within the condensate complex in a complicated cellular environment; (2) IRTKS condensates may undergo time-dependent changes, some of them or other unknown molecules within the condensates could have partial gel-like properties. In the revised version, as the reviewer suggested, we mentioned the aspherical morphology of IRTKS condensates and possible explanation in the Discussion sections (**page 24, lines 16-24**).
- 2) Additionally, we also performed more experiments to support the phase-separated properties of IRTKS puncta in cells. These include: (1) Time-lapse fluorescence images and fluorescence recovery after photobleaching (FRAP) experiments, showing that EGFP-IRTKS puncta underwent fusion over time and displayed the dynamic feature of liquid droplets (**Fig. 3n and Extended Data Fig. 4x**, in the revised version), in accordance with phase-separated behavior. (2) The engineered HEK293T cells by using the CRISPR–Cas9 system to tag endogenous IRTKS with monomeric enhanced green fluorescent protein (mEGFP), showing that the mEGFP-tagged endogenous IRTKS formed the condensates in the cytoplasm or nucleus, and that the endogenous mEGFP-IRTKS puncta could rapidly fuse and fission (**Extended Data Fig. 4u, v**). (3) FRAP experiment to examine the dynamic characteristic of these endogenous mEGFP-IRTKS puncta, showing that endogenous mEGFP-IRTKS puncta could be recovered after photobleaching (**Fig. 3k and Extended Data Fig. 4w**). These new data further strengthen our previous conclusion that IRTKS possesses phase-separated properties in cells.
- 3) We thank the reviewer for pointing out that the irregular structures of HP1 α were shown in Fig. 1i and k in the previous version. According to the previous studies (PMIDs: 28636597 and 31543422), the similar aspherical puncta of HP1 α were also observed in some cells, suggesting that HP1 α puncta also exhibit the aspherical and irregular structures in cells. We believe that these important observations could be

reasonably explained by more investigation and experiments in future research including ours.

4. Figure 5 shows RE DNA occupancy of HP1 α and H3K9me3 is strongly depleted upon IRTKS knockout. These experiments would benefit from a negative control. Perhaps the analysis of a euchromatic genomic region or non-RE region where the loss of IRTKS is not expected to have a strong effect? This control should be carried out through the entirety of Figure 5.

R: We are grateful for the reviewer's informative comments and helpful suggestions. As the reviewer's advice, we performed the statistical analyses on ATAC-seq and ChIP-seq data of livers and MEFs from *Irtks* KO mice, as compared with those from WT mice, to assess the effect of IRTKS on the euchromatic genomic region or non-RE region, including transcription termination site (TTS), intergenic, intron, exon, 5'UTR and 3'UTR regions.

ATAC-seq analysis demonstrated that the chromatin accessibility on most of non-RE regions was also visibly increased in livers and MEFs from *Irtks* KO mice, as compared with those from WT mice, similar to the effect of other heterochromatin stabilizers such as SIRT3 and ZKSCAN3 on non-RE regions as previously reported (PMIDs: 33706382, 32427330). In the revised version, these new results were added as **Extended Data Fig. 6p and q**.

In addition, we also re-analyzed the ChIP-seq data, showing that the enrichment of H3K9me3 and HP1 α on most of non-RE regions was obviously decreased in livers and MEFs from *Irtks* KO mice, as compared with those from WT mice (**Extended Data Fig. 6h-k**). Consistently, IRTKS deficiency led to the increase of the transcript levels of non-RE, as shown by RNA-seq analysis (**Fig. 5g**). Collectively, these new results indicate that IRTKS deficiency almost affects all chromatin regions, including heterochromatic and euchromatic region. The underlying mechanisms by which IRTKS affects these euchromatic regions will be a direction of our future study.

In the revised version, we have modified some sentences for clear description related to new data **on pages 18-19** in the Results section. Additionally, we also briefly discuss the effect of IRTKS on chromatin regions, including euchromatic regions, in Discussion section (**page 25, lines 21-25; page 26, lines 1-2**).

Minor comments

1. Figure 2B includes a Western blot image of SUMOylated HP1 α . Can the authors annotate the anti-SUMO1 blot with band assignments? Is the large band at ~43 kDa mono-SUMO HP1 α ? Some annotation will help readers that are not familiar with analyzing SUMOylation blots to interpret the data.

R: We applaud the reviewer for careful inspection of the data and apologize for any confusion we caused. As suggested by the reviewer, we have annotated the anti-SUMO1 blot with band assignment according to previous reports (PMIDs: 29588524 and 29506078). The result was shown in **Fig. 2b** in the revised version.

2. The use of a knockout mouse model throughout this paper is very powerful. Can the authors comment on the influence of IRTKS knockout on the overall health of the mouse? Does it seem to be healthy or sick? Is there a noticeable phenotype? Has this mouse previously been published?

R: We thank the reviewer for raising this important point. We previously generated *Irtks* knockout mice. The *Irtks* KO mice did not exhibit developmental problems and were phenotypically normal, however, *Irtks* deficiency may lead to insulin resistance, including hyperglycemia, hyperinsulinemia and glucose intolerance (Cell Research, 2013; PMID: 23896986). In addition, *Irtks* KO mice also exhibit enhanced antiviral activity against RNA viruses, suggesting that IRTKS functions as a negative modulator of excessive inflammation (Nature Communications, 2015; PMID: 26348439). Recently, our data showed that heterozygous *p53*^{+/-} mice with IRTKS deficiency display markedly delayed tumorigenesis and an extended tumour-free survival time, suggesting IRTKS provides a promising prognostic factor for patients with gastric cancer (GUT, 2018; PMID: 28647685). We have briefly mentioned these studies in the Introduction section (**page 3, lines 12-23; page 4, lines 9-14**).

3. Do the authors think that the sole function of IRTKS in heterochromatin formation is to stabilize HP1 α levels? In other words, is IRTKS knockout rescued by over-expressing HP1 α ? They show that knockout of HP1 α is not rescued by over-expressing IRTKS, but is the reverse also true?

R: Many thanks the reviewer for these insightful suggestions. In the light of the reviewer, we knocked out *IRTKS* in human SK-Hep-1 cells using the CRISPR/Cas9 system and then observed the effect of ectopic HP1 α on heterochromatin formation. Base on TEM images, we found that heterochromatin at the nuclear periphery could be partially restored by ectopic HP1 α expression, implying that IRTKS-mediated heterochromatin formation, at least partially, depends on HP1 α . And these new results were shown in **Extended Data Fig. 2c and d** in the revised version

4. Figure 6G assesses mRNA levels of various SASP-associated genes. Can the authors comment on the ~10-fold increase in CXCL-1 mRNA compared to Figure 6H's only ~2-fold increase in CSCL-1 protein levels?

R: We thank the reviewer for pointing this out, we have repeated this experiment (include CXCL-1, IL-6, TNF α and IL-1 β) in three additional kidneys of WT and *Irtks* KO mice. Similar to the previous conclusion, our new results also showed that these known SASP-associated genes (CXCL-1, IL-6, TNF α and IL-1 β) were markedly upregulated in the kidneys from *Irtks* KO mice, as compared with those of WT mice (n=6). Additionally, CXCL-1 mRNA level increased about 10-fold, while protein level increased about 2-fold. We speculated that the potential reasons for the discrepancy between CXCL-1 mRNA and protein levels are: (i) post-transcriptional or translational regulation, such as CXCL-1 mRNA decay or translation efficiency, may play a role in regulating CXCL-1 protein level. (ii) CXCL-1 protein with shorter *in vivo* half-lives

may lead to a relatively slight increase in protein level as compared to mRNA level. We thank the reviewer for raising this question, which will be helpful to our future exploration. The new results were shown in **Fig. 6g** in the revised version.

5. Figure 2 shows Western blots that highlight the amount of over-expression of IRTKS at play in many of the experiments in this paper. The authors should comment on this.

R: We thank the reviewer for raising this important point. As shown by Fig. 2b, HP1 α could be SUMOylated by SUMO-1, and ectopic IRTKS overexpression also obviously enhanced SUMOylation of HP1 α by SUMO1 in a dose-dependent manner. In the revised version, we briefly comment on this (**page 9, lines 23-25**).

Second-round review

Remarks to the Author:

The revised manuscript from Xie, et al., thoroughly addresses the editor's and reviewers' requests for revision and is suitable for publication. The manuscript presents compelling data for a role of IRTKS in "strengthening" HP1a-associated heterochromatin through a mechanism involving phase separation and will be of broad interest in the chromatin biology and biomolecular condensates fields.

R: We sincerely thank the reviewer for the positive comments on our revision. We also thank the reviewer's professional suggestions, which are helpful to improve the quality of the manuscript.

The reviewer does have a few minor comments/suggestions:

1. Ext. Data Fig. 2i, j. The lower scheme should be labeled GST-IRTKS, not GST-HP1 α . And the labels across the top of panel j are incorrect; they should be the domains of HP1 α .

R: We applaud the reviewer for careful inspection of the data and apologize for any confusion we caused. The labels have been revised as suggested by the reviewer (**Extended Data Fig. 2i, j**)

2. Page 11. "The analysis showed that IRTKS has some potential IDRs (Fig. 3a, Extended Data Fig. 4a), implying that IRTKS potentially undergoes phase separation." Not all IDRs undergo phase separation. Maybe rephrase this sentence as follows: "The analysis showed that IRTKS has some potential IDRs (Fig. 3a, Extended Data Fig. 4a), suggesting the possibility that IRTKS can undergo phase separation."

R: We are grateful for the professional and valuable suggestion. According to the suggestion, we have rephrased this sentence as follows: "The analysis showed that IRTKS has some potential IDRs (Fig. 3a, Extended Data Fig. 4a), suggesting the possibility that IRTKS can undergo phase separation." (**Page 11, lines 14-16**).

Reviewer: 2

Remarks to the Author:

In this paper, the authors state that insulin receptor tyrosine kinase substrate (IRTKS from here on) is indispensable for constitutive heterochromatin via a liquid-liquid phase separation mechanism (LLPS). Further, the loss of IRTKS can alter genome-wide levels of heterochromatin, chromatin accessibility and expression of repetitive elements that promote cellular senescence. The authors use a diverse set of approaches that work well together to support their overall hypothesis. Overall, the work is strong and well supported by the chosen experiments. Below I enumerate a number of concerns, these concerns include more rigorous analysis of the ATAC-seq, ChIP-seq and RNA-seq data.

R: We are deeply grateful to the reviewer for the positive feedback and appreciate the careful and thoughtful comments.

Major concerns:

1. The analysis of ATAC-seq and RNA-seq of MEFs from WT and IRTKS knockout mice was weak. The analyses shown for ATAC, ChIP and for RNA need to include p-values rather than “enrichment levels”.

R: Many thanks for the valuable suggestion. As suggested by the reviewer, we re-analyzed statistically the ChIP-seq, RNA-seq and ATAC-seq data of livers and MEF cells from WT and *Irtks* KO mice.

In our ATAC/ChIP-seq analyses, there was no bias or preference in the selection of loci for the different groups of samples. The visualizations and p-value calculations are based on data generated by the computeMatrix command in the deeptools toolkit, by utilizing a matrix with scaled height values of the upper and lower 5kb (3kb for ChIP-seq) widths of peaks for all loci. These height data are presented in 10bp bins, and the overlay of these heights is the final line charts or heatmaps. We also performed a significance test of difference analysis on the peak heights within the central 200bp (100bp for ChIP-seq) by paired student's t-test to get the p-values with setting adjusted false discovery rate (FDR), and these results were shown as **Fig. 5a, b, f, Extended Data Fig. 6a, b, h-k and n-q** in the revised version, the conclusion still remains consistent with the previous version. And we have modified some sentences for clear description in the corresponding Methods section (**pages 46-48**).

In the RNA-seq analysis, we initially obtained the expression of each repetitive element (RE). These expression values were subsequently normalized using *Actb* as a negative control, and then depicted according to RE-class in the paired knock-out and wild-type samples. It is important to emphasize that, despite the y-axis in Fig. 5g being presented in a logarithmic format, the p-values were calculated by the Student's t-test of values prior to logarithm transformation. Accordingly, we have modified certain sentences for clear description in the corresponding Methods section (**page 45, lines 14-21**).

We found that the occupation by H3K9me3 and HP1 α on diverse repetitive sequences, particularly SINE, LINE and LTR, was significantly decreased in the livers

and MEFs from *Irtks* KO mice with p-values less than 0.05, as compared with those from WT mice. Consistently, IRTKS deficiency leads to the significant increase of chromatin accessibility on repetitive DNA sequence regions with p-values less than 0.05, as shown by ATAC-seq data, and accordingly, the transcript levels of repetitive DNA sequences were significantly upregulated with p-values less than 0.05. In the revised version, these new results were provided in **Fig. 5a, b, f, g and Extended Data Fig. 6a, b, n and o** with p-values. Additionally, we also added more methodological details in the Methods section (**pages 45-48**).

2. The authors show use ChIP-seq on H3K9me3 and HP1a, which show decreases in both signals (15-20% in H3K9me3 and > 50% in HP1a) at several types of repeat elements. While the correct normalization protocols seem to have been used for quantifying possible genome-wide changes (which run the risk of being indistinguishable under scalar multiplication), the authors also perform ChIP-qPCR, and find the same results with this orthogonal method. One comment is regarding the wording that the authors use to describe the ChIP-seq result: they claim that not only are the levels of H3K9me3 and HP1a decreasing, but that the co-localization of the two is reduced. This implies an extra step – not only does IRTKS decrease the levels of H3K9me3 and HP1a (which is a reader of H3K9me3 and should theoretically “follow” its signal), but it effectively decouples the two from each other. If this were the case we would see regions with high levels of H3K9me3 but low levels of HP1a, or vice versa. None of the ChIP-seq data presented by the authors seems to corroborate this claim – rather, the data suggests concurrent decreases in both signals across all measured regions upon IRTKS knockout. It is much more straightforward to conclude that by reducing the levels of H3K9me3, HP1a is reduced at the same peaks. I suggest the authors remove or at least reword any claims of “altered colocalization”.

R: We thank the reviewer for valuable comment and suggestion, and apologize for making it confusing. We agree the reviewer that heterochromatin endows the particular genomic domains with enrichment of H3K9me3 and its reader protein HP1 α , which are usually adjacent to repetitive sequences (PMIDs: 32504224, 32427330 and 30044650). Our ChIP-seq and ChIP-qPCR analysis showed that the enrichment of H3K9me3 and HP1 α on REs was obviously decreased in livers and MEFs from *Irtks* KO mice, as compared with those from WT mice (**Fig. 5a-d and Extended Data Fig. 6a-d**), which was similar to the observation of previous report (PMID: 32427330), suggesting that IRTKS is crucial to the regulation of heterochromatin domains.

However, we agree with the reviewer that our current data do not make sure that IRTKS could effectively decouples H3K9me3 and HP1 α from each other, even though our ChIP-seq analysis showed that IRTKS deficiency markedly decreased the occupation of H3K9me3 and HP1 α on the same chromatin binding sites across the genome in mouse livers and MEFs, especially on RE regions (**Fig. 5e and Extended Data Fig. 6e-g**). Accordingly, as the reviewer suggested, we revised the corresponding Results section and removed our claims of “altered colocalization”. In the revised version, we only state that, in the absence of IRTKS, the enrichment of H3K9me3 and HP1 α is reduced at the same chromatin binding sites across the genome in mouse livers

and MEFs, especially on RE regions (**page 18, lines 8-12**).

3. The ATAC-seq peak data at repeat elements corroborates the ChIP-seq results, however additional controls are needed alongside improved descriptions of genome-wide differential accessibility. Do the authors see changes at control loci, for example *Actb*, *Gapdh* or others? If control loci are changing accessibility, this likely suggests an issue with the experiment or with the analysis. When considering genome-wide changes in accessibility, do repeat regions harbor the largest changes in accessibility, or are there other regions that may be unexpectedly affected by knockout? Further, ATAC-seq should be performed in primary cells (i.e. the liver). In this analysis, the author should comment on whether there are different outcomes of accessibility in different cell types.

R: Many thanks for the professional suggestions. According to the advice of the reviewer, we do more analysis and experiments.

- 1) As the reviewer suggested, we perform the statistical analyses on the chromatin accessibility of *Actb* gene as control locus, showing that chromatin accessibility of *Actb* locus in livers and MEF cells from *Irtks* KO mice was not significantly changed, as compared to that from WT mice, indicating that the ATAC-seq data in our study is available. The new results were added in **Extended Data Fig. 6l and m** in the revised version.
- 2) In addition, as the reviewer suggested, we also further employed ATAC-seq data to assess the effect of IRTKS-mediated chromatin accessibility on non-RE regions, including transcription termination site (TTS), intergenic, intron, exon, 5'UTR and 3'UTR regions. ATAC-seq analysis demonstrated that chromatin accessibility was significantly increased on most of non-RE regions in livers and MEFs from *Irtks* KO mice, as compared with those from WT mice (**Extended Data Fig. 6p and q**), which is similar to the effect of other known heterochromatin stabilizers such as SIRT3 and ZKSCAN3 on non-RE regions as previously reported (PMIDs: 33706382, 32427330). Furthermore, we re-analyzed the ChIP-seq data, showing that the enrichment of H3K9me3 and HP1 α on most of non-RE regions was significantly decreased in livers and MEFs from *Irtks* KO mice, as compared with those from WT mice (**Extended Data Fig. 6h-k**). Consistently, IRTKS deficiency leads to a significant increase of transcript levels of non-RE, as shown by RNA-seq analysis (**Fig. 5g**). Collectively, these new results suggest that IRTKS deficiency almost affects all chromatin regions, including heterochromatic and euchromatic regions. In the revised version, we have modified some sentences for clear description related to Fig 5 on **pages 18-19** in the Results section. The underlying mechanisms by which IRTKS affects the euchromatic regions will be a direction of our future study. We briefly discuss it in Discussion section accordingly (**page 25, lines 21-25; page 26, lines 1-2**).
- 3) Moreover, as suggested by the reviewer, we also performed ATAC-seq analysis to evaluate the genome-wide changes in livers, as primary cells, from WT and *Irtks* KO mice (n=2 biological replicates). We observed that the chromatin accessibility on repetitive DNA sequence regions and non-RE regions was markedly increased in livers from *Irtks* KO mice, as compared with those from WT mice, which was

consistent with the genome-wide changes in MEF cells. These results were added as **Fig. 5f, Extended Data Fig. 6m, n and q**.

- 4) In the revised version, as the reviewer suggested, we describe and comment the outcomes of accessibility in different cell types in the Results section (**page 18, line 25; page 19, lines 1-4 and 8-11**).

4. For the RNA-seq data, if I understand correctly, there are 4 mice for each condition, so statistics could be calculated as described in the figure legend and histograms plotted for each of the 7 repeat types as in 5c and 5d. Although the data is presented as “mean +/- SD”, there is only one color on the heatmap for each square, and it is clear that the authors normalized to the WT controls but it is not clear which of the knockouts was normalized to which control. I think a simpler, histogram-based method with p-values would be more straightforward here since there are only 7 species of transcripts that the authors are analyzing.

R: We thank the reviewer for the valuable suggestion. As the reviewer pointed out, we performed RNA-seq analysis to dissect the role of IRTKS in regulating the transcript levels of 7 repetitive DNA sequences in the livers from WT and *Irtks* KO mice (n = 4 for each condition). We are sorry for not clearly describing the normalization and control as statistically calculating the transcript levels of these 7 repetitive DNA sequence types in previous manuscript. In the revised version, as suggested by the reviewer, we employed a simpler, histogram-based method to re-analyze our RNA-seq data related to 7 repetitive species of transcripts, and generated these histograms with p-values, as shown in **Fig. 5g**. We also added details about the analysis in Methods section (**page 45, lines 4-19**).

5. Two minor notes: 1. Fig. 5 title should be “IRTKS deficiency leads to genome-wide epigenetic alterations.” 2. In Materials and Methods for ATAC-seq, “Ice-clod PBS” should be “Ice-cold PBS.”

R: We sincerely appreciated the reviewer for careful inspection and professional suggestion. Accordingly, in the revised version, the title of Fig. 5 has been revised as suggested by the reviewer (**page 17, line 20**), and we also corrected the error in Materials and Methods (**page 47, line 12**). We have carefully checked the manuscript to rectify other inaccuracies.

Reviewer: 3

Remarks to the Author:

Xie et al. study how insulin receptor tyrosine kinase substrate (IRTKS) affects heterochromatin organization in different mouse cell types and human SK-Hep-1 and HEK293T cell lines. The main conclusions are: (i) IRTKS undergoes liquid-liquid phase separation (LLPS) to form droplets at heterochromatin together with HP1 α . (ii) It enlarges phase-separated HP1 α droplets by promoting its sumoylation. (iii) IRTKS is essential to maintain heterochromatin structure, silencing of repeat transcription and

genome-wide chromatin accessibility via its interactions HP1 α .

The paper contains a broad range of interesting findings on IRTKS as a novel factor involved in heterochromatin organization. However, I am not convinced that the data would justify the main conclusion that IRTKS, together with HP1 α , is involved in LLPS to establish and maintain heterochromatin structure. My main concerns include, but are not limited to, the following points:

R: We would like to thank the reviewer for the constructive and expert comments on improving our manuscript. We also appreciate the critics that a few key concerns need to be addressed to make these findings more convincing. We have performed additional experiments to address these excellent questions, and rephrased some statements of the manuscript for the clarity and preciseness.

1. Introduction, l. 83 “Significantly, recent studies have provided an intriguing model in which heterochromatin formation can be driven by HP1 α via liquid–liquid phase separation (LLPS) 27-32” and other parts of the text, e.g., l. 221, “it is known that HP1 α can form phase-separated droplets to drive heterochromatin formation...”

I do not think that this is an appropriate way to describe the state of current research on heterochromatin and HP1 α .

- Ref 31 cited by the authors says in the title: “Mouse heterochromatin adopts digital compaction states without showing hallmarks of HP1-driven liquid-liquid phase separation”.

- Ref. 27, 29 and 32 draw conclusions about LLPS for heterochromatin based on in vitro experiments with purified factors but do not show that LLPS would occur in the cell.

- Another study not cited by the authors concludes that heterochromatin does not have liquid but rather solid properties (PMID 33326747).

- Many previous studies not mentioned by the authors have addressed the mechanisms by which heterochromatin forms and rationalize the experimental data by a variety of quantitative models that do not invoke LLPS (e.g., PMIDs 22704655, 25134515, 29190361, 29365171, 32778583).

R: We very appreciate the reviewer for the insightful and professional comments. In general, we agree the reviewer that the issue about LLPS-driven heterochromatin formation is in controversial, however, our current data support the notion that liquid-liquid phase-separation is involved in our current data.

1) As correctly pointed by the reviewer, Ref. 31 indicated that mouse heterochromatin lacks the HP1-driven liquid-liquid phase separation. However, this paper also exhibited that HP1 α still has a capacity to form droplets in living cells, albeit on short timescales. In addition, we also agree with the reviewer for pointing out that heterochromatin has solid properties in living cells, as shown by another paper (PMID: 33326747). Nonetheless, this paper also indicated that heterochromatin-associated proteins, such as HP1 α , displayed a liquid-like behavior. Furthermore, the latest paper (PMID: 36710442) explained that the chromatin exhibits solid-like state *in vivo* due to its large size, where the solid chromatin also provides a scaffold

for liquid-liquid phase separation of various chromatin-binding proteins. Additionally, Ref. 28 and 30 also concluded that HP1 α possesses the liquid properties in both *Drosophila* and mammalian cells. Collectively, the conclusion that LLPS was required for heterochromatin formation based on *in vitro* experiments, as drawn by Ref. 27, 29 and 32, is reasonable. Although the issue is in controversial, the HP1 α -driven liquid-liquid phase-separation is also supported by our current data.

- 2) We do agree with the reviewer that the mechanisms by which heterochromatin formation could not be involved in LLPS, as supported by some papers provided by the reviewer. However, several latest studies indicated that heterochromatin foci could be mediated by HP1 α (PMIDs: 37549295, 28636597 and 31543422), methyl-CpG binding protein 2 (MeCP2) (PMIDs: 32698189, 32111972 and 35156529), linker histone H1 (PMID: 38309957), scaffold attachment factor B (SAFB) (PMID: 31677973) and 53BP1 (PMID: 35042897) through LLPS, which suggested that LLPS drives the heterochromatin formation.

We thank the reviewer for these important points. Considering the issue being in controversial, we rephrased some sentences about LLPS, heterochromatin and HP1 α in Introduction, Results and Discussion section, where we also cited different views on the issue (**page 5, lines 8-10; page 11, lines 8-9; page 23, lines 18-19; page 24, lines 6-11**).

2. Various types of screens across organisms have been conducted to dissect the composition of heterochromatin. The pericentric heterochromatin in mouse cells (the “chromocenters”) has been characterized by mass spectrometry (PMID 25457167). A recent study dissects silencing by H3K9me3 heterochromatin in mouse cells with a CRISPR screen (doi: 10.1101/2023.02.27.530221). These previous studies have not identified IRTKS as a central heterochromatin factor as proposed by the authors. Thus, it would warrant some considerations as to why IRTKS has not been seen in previous studies. When considering the quite broad range of activities that have been assigned to this factor (binding to insulin receptor on the cell membrane, suppresses phosphatase activity, regulation of innate antiviral immunity, p53 ubiquitination etc.) the reported IRTKS^{-/-} heterochromatin phenotypes might also result indirectly from other activities as opposed to perturbing sumoylation of HP1 α .

R: Many thanks for the professional comment. We also carefully read the two papers mentioned by the reviewer.

- 1) The paper (PMID: 25457167) performed PICCh experiment using the probes that specifically target major satellites to characterize the composition of mouse pericentromeric heterochromatin. As the reviewer mentioned, IRTKS was not identified by the work, which could be due to the PICCh probes specific binding to major satellites (a repetitive DNA sequence), not to HP1 α . In our study, our current data only indicate that IRTKS is required for heterochromatin architecture via interacting with HP1 α , whereas IRTKS could not directly bind to repetitive DNA sequences, as mentioned in Results sections, “These REs did not appear to be

immunoprecipitated by an anti-IRTKS antibody, as detected by CHIP-PCR (data not shown), implying that IRTKS does not directly bind to genomic DNA.” Therefore, IRTKS could be very difficult to be identified by the PICh method in the paper. Additionally, we noted that the other known heterochromatin regulators as previously reported, such as ZKSCAN3, CLOCK and SIRT3 (PMIDs: 32427330, 32737416 and 33706382), also were not identified by the PICh experiment in this paper (PMID: 25457167), implying there exists the limitations of this PICh method.

- 2) Another paper (doi: 10.1101/2023.02.27.530221) performed a forward CRISPR-Cas9 genetic screen by targeting 1160 genes, most of which encode known chromatin and epigenetic regulators, DNA replication factors, nuclear periphery factors, and RNA processing factors. Although a fraction of them was associated with heterochromatin, IRTKS was not included in this CRISPR screen because its role in heterochromatin regulation has not been previously recognized. We speculated that the CRISPR screen has some limitations due to the screening scope or false-negative results of a large-scale screen.
- 3) In addition, regarding the potential involvement of IRTKS in heterochromatin formation, a recent study published in Nature utilized mass spectrometry following NeutrAvidin pulldown to identify proteins bound to endogenous retroviruses (ERVs). In this study, IRTKS (also referred to as BAIAP2L1) and other established heterochromatin regulators like HP1 α (also known as CBX5) were notably recognized as ERV-bound proteins involved in the regulation of H3K9me3 levels, where the data was shown in the Supplementary Table 1 of the published report (PMID: 37938770).

We also thank the reviewer for this concern related to the IRTKS $^{-/-}$ heterochromatin phenotypes, which might also from other activities as opposed to perturbing sumoylation of HP1 α . In fact, we and our collaborators have reported that IRTKS overexpression promoted tumor suppressor p53 ubiquitination and degradation in gastric cancer cells by recruiting the p53-specific E3 ubiquitin ligase MDM2 (GUT, 2018; PMID: 28647685). During viral infection, IRTKS can recruit ubiquitin-conjugating enzyme 9 (Ubc9) to SUMOylate PCBP2 within the cell nucleus (Nature Communications, 2015; PMID: 26348439). Furthermore, our research indicates that IRTKS could increase the levels of H3K9me3 by facilitating the accumulation of SETDB1 (Cancer Lett. 2023, PMID: 37739210). Accordingly, the statement has been added in Introduction and Discussion section in the revised version (**page 4, lines 9-16; page 21, lines 22-25; page 22, lines 1-2**). These data indicate that IRTKS may function as a nuclear protein, which also raise the possibility that IRTKS could play an unexpected role in regulating the eukaryotic genome and chromatin. In fact, in this work, we performed some powerful experiments to exhibit the direct evidences, showing that IRTKS is required for heterochromatin formation by its condensates with HP1 α , a crucial factor for heterochromatin. Certainly, the deep mechanism by which IRTKS participates in heterochromatin formation and maintenance should be further investigated in future.

3. Fig. 1, Extended Data Fig. 1, l. 112: “These results suggested that IRTKS could be required for heterochromatin”.

It is noted that the DAPI dense chromocenters in Fig. 1g, i and k as well as in the Extended Data Fig. 1i persist in *IRTKS*^{-/-} cells. This is in contrast to the TEM images that show a loss of heterochromatin domains, e.g., Fig. 1a vs Fig. 1i. In addition, it is not apparent to me from the images in Fig. 1e that there is a 3-fold reduction in heterochromatin in MEFs similar to the liver and kidney images, where the differences appear to be much more pronounced.

R: We thank the reviewer for these insightful questions, and apologize for the point being unclear in previous version. These concerns raised by the reviewer are important, to address it, we perform more analysis and experiments as following:

- 1) It is generally believed that heterochromatin domains can be detected as electron-dense regions (EDRs), which are located at the nuclear periphery or around the nucleolus, by transmission electron microscopy (TEM) (PMIDs: 36533441, 33307247, 30140741 and 25479748). As correctly pointed by the reviewer, we only focused on the change of heterochromatin domains at the nuclear periphery, which was inconsistent with the location of *IRTKS* throughout the nucleus. To address this point, we re-analyzed the TEM images of livers, kidneys, stomachs and MEFs from WT and *Irtks* knockout mice, and human SK-Hep-1 cells with or without *IRTKS*, and then quantified the heterochromatin domains based on EDRs around the nucleolus. Interestingly, the quantificational analysis showed that, in the absence of *Irtks*, the heterochromatic regions around the nucleolus were also dramatically reduced in these mouse livers, kidneys and stomachs. Consistently, we observed a slight loss of heterochromatin around the nucleolus in the *IRTKS*-KO MEF and SK-Hep-1 cells. Conversely, ectopic *IRTKS* overexpression increased the EDRs of heterochromatin around the nucleolus in MEFs and SK-Hep-1 cells. These new results, combined with our previous data, suggest that *IRTKS* could globally regulate heterochromatin domains. Accordingly, we have modified some sentences for clear description in the corresponding Results section (**page 5, line 25; page 6, lines 1-14**), and added these new results as **Extended Data Fig. 1a, b, e, f, i, l and p** in the revised version.
- 2) In addition, as to the point that it is not apparent that *IRTKS* deficiency results in 3-fold reduction in heterochromatin in MEFs in the previous version, we also further analyze it. The issue could arise from the lesser electron-dense regions (EDRs) in MEFs, as compared to that in mouse livers and kidneys, likely leading to statistical errors during quantification analysis. To address it, we adopt more TEM images to find the apparent EDRs in MEFs from WT mice, and these EDRs of TEM images from MEFs were strictly re-quantified and statistically analyzed. In the revised version, we redraw the statistical graphs with p-values, showing a significant reduction of heterochromatin regions in MEFs from *Irtks* KO mice, as compared with that from WT mice, as shown in **Fig. 1e and f**. The conclusion derived from new data remains the same as in the previous version.

4. 1. 441 summarizes the major findings of the study as “More significantly, *IRTKS*, based on both in vivo and in vitro experimental evidence, has the capacity for phase separation and concentrates heterochromatin condensates, converging with HP1 α liquid

droplets and diverse nucleosomal arrays, which results in the further compaction of chromatin and repression of repetitive DNA elements“

It is unclear to me, which experiments in the manuscript would show a compaction function of HP1 α that is induced or enhanced by IRTKS. There is also no chromatin compaction phenotype for HP1 α in mouse cells reported in other studies. MEF with double knockout of the Suv39h1 and Suv39h2 genes lose both H3K9me3 and HP1 α enrichment at chromocenters but that compaction of chromatin persists (PMID 11701123). The knock-out of mouse HP1 α also has no effect on chromocenter compaction in MEFs at fluorescence microscopy resolution (PMID 29166597). Rather, the later study shows that HP1 α knock-out leads to a loss in accessibility as mapped by MNase, i.e., the very opposite phenotype to the genome-wide increase in chromatin accessibility reported for the IRTKS $^{-/-}$ cells.

R: We thank for the professional comments from the reviewer. As correctly pointed out by the reviewer, according to our current data, IRTKS results in a decrease in chromatin accessibility and repression of repetitive DNA elements, but not chromatin compaction. We agree with the reviewer, we have modified the statement in Discussion section in the revised version (**page 22, lines 12-17**).

5. In Fig. 1i the authors show essentially a complete loss of the H3K9me3 signal in IRTKS $^{-/-}$ cells that occurs not only at chromocenters but throughout the complete cell. It is noted that there is a significant amount of H3K9me3 in euchromatic regions that should give some signal by immunostaining (e.g., PMID 14690609). The image is also inconsistent with the differences in the H3K9me3 ChIP-seq profiles at repeats in Fig. 5a where the enrichment is quite moderate and decreases only from something like 1.6-1.4 to 1.4-1.2 in the IRTKS $^{-/-}$ cells.

R: We thank the reviewer for raising these important points about an almost complete loss of the H3K9me3 signal in IRTKS $^{-/-}$ cells. The paper (PMID: 14690609) indicated that the robust colocalization for H3K9me3 at heterochromatin in 100% of the nuclei by immunostaining and statistical evaluation of the staining, which is consistent with the view that H3K9me3, a classical marker of heterochromatin, is enriched on the transcriptionally silent regions of the genome, rather than euchromatin, which is a non-condensed chromatin state that is enriched in genes and permissive for transcription (PMIDs: 29235574 and 35338361). Although so, we agree the reviewer that there is a significant amount of H3K9me3 in euchromatic regions, and therefore, we re-analyzed the ChIP-seq data, showing that the enrichment of H3K9me3 on most of non-RE regions was significantly decreased in livers and MEFs from *Irtks* KO mice, as compared with those from WT mice (**Extended Data Fig. 6h and j**). Consistently, IRTKS deficiency leads to a significant increase of transcript levels of non-RE, as shown by RNA-seq analysis (**Fig. 5g**). These new data provide a potential clue that IRTKS could play a role in regulating euchromatin by modulating H3K9me3, which should be further investigated in future.

Furthermore, to further address it, we performed immunofluorescence (IF) assay to detect the changes of H3K9me3 signal in livers from WT and *Irtks* KO mice. IF

images and line scan analysis showed that the signal intensity of H3K9me3 foci was obviously decreased in the livers from *Irtks* KO mice, as compared with that from WT mice, as shown by new data in **Extended Data Fig. 1u**, which is similar to the observation as previously reported (PMID: 31677973), further confirming our conclusion.

As to the question about the inconsistency with the results of the IF images and H3K9me3 ChIP-seq profiles in the liver raised by the reviewer, we re-analyzed the ChIP-seq data of livers from WT and IRTKS-KO mice with statistical p-values, which were generated by paired-t test. It is noted that the enrichment level of H3K9me3 ChIP-seq was generated from the total number of peaks within the loci located on diverse repetitive element sequences, ranging from several hundred to several thousand. The result showed that the enrichment of H3K9me3 on repetitive element (RE) sequence regions was significantly decreased in livers (**Fig. 5a**), although the decreased enrichment level appears moderate, which was similar to the observation as previously reported (PMIDs: 35286396 and 32427330). Based on these new data, combined with the result of ChIP-qPCR (**Fig. 5c**), the conclusion still remains the same as in the previous version.

6. The author study a variety of different types of heterochromatin repeats in mouse cell types (liver, kidney, stomach, MEFs) as well as in the human SK-Hep-1 and HEK293T cell lines. However, the heterochromatin organization is quite different between mouse and human cells. The intensely DAPI stained chromocenters are very prominent in mouse cell but are absent in the human cell lines. Thus, it is unclear to me how the HP1 α foci that are studied by FRAP in SK-Hep-1 cells relate to the compacted heterochromatin regions that are seen in the mouse cells. This is evident from Extended Data Fig. 1m. As in other human cells, there is only a moderate chromatin density increase at the nuclear periphery as visualized by Hoechst 33342 (or DAPI) staining. However, there is no colocalization of the HP1 α foci with these bona fide heterochromatic regions on the images. It is also noted that HP1 sumoylation was not detected by mass spectrometry in human HEK cells (LeRoy PMID 19567367), which points to fundamental differences for the role of HP1 sumoylation in mouse versus human cells.

R: Many thanks for the insightful comments and valuable suggestions. To address these interesting concerns, we perform more analysis or experiments as following:

- 1) We agree with the reviewer that human cell lines typically do not have prominent chromocenters. However, the biophysical feature of HP1 α foci was investigated by FRAP not only in mouse cells (PMID: 31677973), but in the human cells (PMID: 31677973 and 31543422). To address the reviewer's concern, we also performed live-cell fluorescence imaging and fluorescence recovery after photobleaching (FRAP) experiments using enhanced green fluorescent protein (EGFP)-tagged HP1 α in MEF cells, showing that HP1 α puncta were rapidly exchanged in the nucleus of MEFs, and also co-localized with the intensely DAPI-stained regions, as indicated by the line scan analysis. In contrast, the recovery of HP1 α puncta in

IRTKS-KO MEF cells was significantly slower than that in the WT control cells. The new results were shown as **Extended Data Fig.1x and y** in the revised version.

Furthermore, we performed the same experiments on human SK-Hep-1 cells, as the reviewer's professional comments, there are only moderate chromatin density regions in this human cell type as visualized by Hoechst 33342 staining, which was similar to the observation in other human cells, such as MDA-MB-231 and U-2OS cells, as previously reported (PMID: 31543422). Also, we analyzed the location of HP1 α foci by the line scan analysis, showing that HP1 α foci co-localized with these mild chromatin density regions. The recovery of HP1 α puncta in these IRTKS-KO SK-Hep-1 cells was also significantly slower than that in the control cells. These new results were added as **Extended Data Fig.1v and w** in the revised version.

- 2) Additionally, as correctly pointed out by the reviewer, HP1 α sumoylation was usually detected in the embryonic mouse fibroblast cell line NIH3T3 (PMIDs: 21317888, 22388734 and 27426629). We understand the reviewer's concern that HP1 sumoylation was not detected by mass spectrometry in human HEK cells, even though many SUMOylated proteins could be identified in HEK 293T cells (PMIDs: 36050397, 29588524, 29506078 and 27637147). To further confirm our observation, we followed the previous publications to perform *in vivo* sumoylation experiment in NIH3T3 cells. Both HA-tagged IRTKS and Flag-tagged HP1 α together with GFP-tagged SUMO1, were transiently co-transfected into NIH3T3 cells, and the subsequent immunoblotting results showed that SUMOylation of HP1 α could be detected in NIH3T3 cells, which was in agreement with previous reports (PMIDs: 22388734 and 27426629). More importantly, SUMOylation of HP1 α was obviously enhanced in the presence of IRTKS, conforming our conclusion that IRTKS can enhance HP1 α SUMOylation. The new results were shown as **Extended Data Fig.3f** in the revised version. These collective data suggest that the effect of IRTKS on HP1 α SUMOylation could be similar and conserved in mouse and human cell lines.

7. The evidence provided by the authors for a LLPS of IRTKS in cells is insufficient.
 - It is now well established that 1,6-hexanediol treatment simply weakens hydrophobic interactions and has a lot of confounding effects, e.g., PMID: 35963432, 36526633, 33814344, 33836078, 33536240. Thus, it is not suited to distinguish whether assembly is driven by LLPS or by a different mechanism (Fig. 3m).
 - The FRAP experiments in SK-Hep-1 appear to be without any quantification although it says "data are presented as the mean \pm SD and the p value of two-tailed unpaired Student's t test" in the figure legend. The observation that after 60 sec the HP1 α signal has recovered cannot be used to demonstrate the presence of an LLPS mechanism as discussed previously (PMID 31594803, 36526633). Numerous reports have shown that protein chromatin interactions (including FRAP experiments of HP1 from two decades ago (PMID 12560555, 12560554)) can be surprisingly short-lived. This is independent of any "fluid-like" properties or LLPS mechanism. It just means that many factors have typical residence time of only a few seconds in the chromatin-bound state.
 - The description of IRTKS properties in the cell (Fig. 3h-l Extended Data Fig. 4o, r)

consists of images of a few selected cells, lacks quantification and I have difficulties to relate what is shown to the proposed model. The IRTKS assembly structures look quite different (Fig. 3h, j, l) and in some instances not at all droplet-like. Furthermore, the IRTKS structures seem to form more in the DNA devoid regions rather than in the DAPI dense heterochromatin. Finally, what is described as a fusion event in Fig. 3k and in the corresponding movie looks to me more like the result of some deformation and rotation of the cell.

R: Many thanks for the reviewer's thoughtful and important concerns. To address these queries, we perform more analysis or experiments as following:

- 1) We agree with the reviewer that it is not most suitable to study the physical properties of LLPS by 1,6-hexanediol (1,6-HD) treatment due to its confounding effects, although 1,6-HD turned out to be the most potent inhibitor of phase separation, which has been widely used, compared with other inhibitors, such as 2,5-hexanediol and 1,5-pentanediol (PMID: 36400991). To address the reviewer's concern, we performed more experiments to support the phase-separated properties of IRTKS in cells. These include: (1) Time-lapse e fluorescence images and fluorescence recovery after photobleaching (FRAP) experiments, showing that EGFP-IRTKS puncta underwent fusion over time and displayed the dynamic feature of liquid droplets (**Fig. 3n and Extended Data Fig. 4x**), in accordance with phase-separated behavior. (2) Engineering HEK293T cells by using the CRISPR–Cas9 system to tag endogenous IRTKS with monomeric enhanced green fluorescent protein (mEGFP), showing that the mEGFP-tagged endogenous IRTKS formed condensates in the cytoplasm or nucleus, and that the endogenous mEGFP-IRTKS puncta could rapidly fuse and fission (**Extended Data Fig. 4u and v**). (3) FRAP experiment to examine the dynamic characteristic of these endogenous mEGFP-IRTKS puncta, showing that endogenous mEGFP-IRTKS puncta could be recovered after photobleaching (**Fig. 3k and Extended Data Fig. 4w**). These new data further strengthen our previous conclusion that IRTKS possesses phase-separated properties in cells.
- 2) We thank the reviewer for raising this great point and apologize for forgetting to provide the quantification for the FRAP experiments in SK-Hep-1 cells in the previous version of our manuscript. As suggested by the reviewer, we have added the results of the quantification for FRAP experiments in **Extended Data Fig. 1w**. In addition, we agree with the reviewer that many factors have typical short-lived residence time in the chromatin-bound state, which is independent of LLPS mechanism. To reinforce our conclusion, we performed live-cell fluorescence imaging and FRAP experiments using enhanced green fluorescent protein (EGFP)-tagged HP1 α in MEF cells, showing that HP1 α puncta were rapidly exchanged in the nucleus of MEFs (**Extended Data Fig. 1y**), which was similar to the observation of these previous reports (PMID: 31677973 and 35042897). In contrast, the recovery of HP1 α in IRTKS-KO MEF cells was significantly slower than that in the WT control cells (**Extended Data Fig. 1y**), which was consistent with the results from SK-Hep-1 cells, implying that IRTKS could enhance the mobility of HP1 α . Nonetheless, we understand the reviewer's concern, and therefore, we

remove the statement that the rapid recovery of HP1 α was consistent with the recent idea that HP1 α has liquid-like properties (**page 7, lines 4-5**). These new results, combined with kinds of evidences in the previous version, strongly supported our conclusion.

- 3) We thank the reviewer and apologize for the lack of quantitative analysis. As suggested by the reviewer, we quantified IRTKS puncta number in the nucleus and cytoplasm per 10 cells of livers and kidneys from WT mice as well as livers from mice with adeno-associated virus serotype 8 (AAV8) vector containing ZsGreen-tagged IRTKS (AAV8-IRTKS). These new results related to the quantitative analysis were showed as **Extended Data Fig. 4o, q and r** in the revised version.

In addition, as to the point of the different structures of IRTKS puncta, as indicated by the reviewer, we assume that IRTKS puncta may undergo concentration-dependent changes. It is believed that the size and morphology of phase-separated are relevant to the concentration of components (PMIDs: 29930091, 30682370 and 30682370). Our results showed that the enforcedly overexpressed IRTKS in mouse livers exhibited more apparent liquid-like puncta than endogenous IRTKS (**Fig. 3h and i**). The similar results were also observed in the HEK293T cells expressing EGFP-IRTKS or mEGFP-tagged endogenous IRTKS (**Fig. 3j and l**), which similar to the observation of FXR1 puncta in the previous report (PMID: 32328638).

- 4) Although we do agree with the reviewer that the typical liquid droplets are considered to be spherical condensates, there are a number of studies showing aspherical morphology of some known molecules with liquid-like features in the complex cellular environment, where they include 53BP1 (PMID: 35042897), NLRP6 (PMID: 34678144), FXR1 (PMID: 32328638), SAFB (PMID: 31677973) and TIS granules (PMID: 33650968). Therefore, the aspherical structures of IRTKS could not be something that is totally unexpected, and we assume that they might be formed by the following possibilities: (1) anisotropic protein-protein (eg: IRTKS-HP1 α) or protein-nucleotide complexes that are probably driven by heterotypic electrostatic interactions among these different molecules within the condensate complex in complicated cellular environment; (2) IRTKS condensates may undergo time-dependent changes, some of them or other unknown molecules within condensates could have partial gel-like properties. In the revised version, we mentioned the aspherical morphology of IRTKS condensates and possible explanation (**page 24, lines 16-24**).
- 5) As correctly pointed by the reviewer, in fact, it is believed that IRTKS is mainly located in the cellular cytoplasm and nucleus under certain condition (PMIDs: 23896986, 31212584, 26348439 and 28647685). Our results showed that IRTKS formed puncta in the cytoplasm and nucleus (**Fig. 3j, k**). In addition, to address the reviewer's concern that the fusion events may result from the rotation of the cell, we repeated the time-lapse experiment, clearly displaying that IRTKS puncta underwent fusion and fission over time. The new result was provided as **Extended Data Fig. 4u, v**.

6) Additionally, to address the concerns raised by reviewer, we examined whether IRTKS puncta accumulated within the heterochromatin foci in the mouse fibroblast NIH3T3 cells and human liver endothelial SK-Hep-1 cells, showing that EGFP-IRTKS mainly accumulated in the nucleus of NIH3T3 cells, and overlapped well with the intensely Hoechst-stained regions. In contrast, the empty vector expressing EGFP failed to lead to formation of puncta, ruling out the possibility that the puncta were artificially formed by the EGFP tag. Additionally, line scan analysis further confirmed the conclusion, similar to the observation as previously reported (PMID: 38101414). In addition, live-cell images and line scan analysis showed that IRTKS puncta in the cytoplasm and nucleus, at least partially, co-localized with Hoechst-dense regions. These new results, combined with our previous data, indicate that IRTKS puncta are likely to be, at least partially, localized in heterochromatin domains. In the revised version, these new data were provided in **Extended Data Fig. 5k-n**.

8. The RNA-seq data on silencing of repeat transcription in Fig. 5g lack a quantification. Although it says in the legend “Data are presented as the mean \pm SD and the p value of one-way ANOVA followed by Tukey’s post hoc test” the figure shows only a heat map of the experimental data from 4 different experiments. For the IRTKS^{-/-} cells, the differences of the results between experiments appear to be large.

R: We thank the reviewer for raising this great point and apologize for not being clearer on these data. In fact, we performed RNA-seq analysis to dissect the role of IRTKS in regulating the transcript levels of diverse repetitive DNA sequences in the livers from WT and *Irtks* KO mice (n = 4 for each group), but we apologized for lacking the quantification to show the changes of the transcript levels of repetitive DNA sequences mediated by IRTKS in the previous manuscript. To address this issue and the similar concern from the reviewer #2 (Question 4), we re-analyzed RNA-seq data and generated more straightforward histograms with statistical p-values (**Fig. 5g**), as the reviewer #2 suggested.

Additionally, the issue about the differences of the results from four different *Irtks* KO mice could arise from the individual differences among mice, which were randomly selected from different littermates for this study. However, the heatmaps analysis employed to evaluate the relative expression of various repetitive DNA sequences in *Irtks* KO mice, as compared with that in these littermate WT mice (**Fig. 5g, in the previous version**), showed similar results, although there exist individual differences.

In the revised version, we provided these new results as **Fig. 5g**, and added details about the analysis in Methods section (**page 45, lines 4-21**).

Reviewer: 4

Remarks to the Author:

The editor summarized the main points that the authors need to address. In my reading of the revised manuscript and the response to reviewers, I feel the manuscript has improved (note that I have not read the prior submission).

The one thing I feel is not well addressed, and this is one of the main concerns of reviewer 3 is this comment from the editor:

“Appropriately cite the known literature on IRTKS and HP1alpha that could have effects independently of any direct causal relationship and independently of biomolecular condensation”

I do not think the authors have done a good job addressing this very important issue. This issue of whether the condensate forming capacity of the system is mechanistically linked to heterochromatin condensation and silencing is a major topic of debate in the field right now and the authors need to acknowledge this much more than they do. In fact, I believe most of the results in the paper can be understood as simply that IRTKS promotes sumoylation of HP1, which stabilizes the protein, increases its concentration and facilitates heterochromatin condensation and silencing. The fact that these proteins form condensates can then be presented a possibly mechanism of action, rather than claiming that the work proves condensate formation per se drives heterochromatin formation.

Here are my views of the author's responses to reviewers 2 and 3. I do not comment on the issues raised by reviewer 1 because I understand reviewer 1 is available for evaluating the revised manuscript.

We sincerely thank the reviewer for the positive feedback, constructive comments, and thorough review of the manuscript. In the revised version, we have tried our best to address concerns by incorporating new experiments, re-analyzing data and making textual changes accordingly.

1. This reviewer is concerned about the statistical basis the ATAC/ChIP/RNA-seq analyses and asks for p-values. The authors have added p-values, but for me it is still unclear what these mean. Are these p-values somehow corrected for multiple testing? Did the authors set some false discovery rate? In the methods a paired t-test with $P < 0.05$ is mentioned, but this is insufficient to understand the meaning of these p-values.

R: We thank the reviewer for these insightful questions, and regret any lack of clarity in previous version.

In our ATAC/ChIP-seq analyses, there was no bias or preference in the selection of loci for the different groups of samples. The visualizations and p-value calculations are based on data generated by the `computeMatrix` command in the `deeptools` toolkit, by utilizing a matrix with scaled height values of the upper and lower 5kb (3kb for ChIP-seq) widths of peaks for all loci. These height data are presented in 10bp bins, and the overlay of these heights is the final line charts or heatmaps. We also performed a significance test of difference analysis on the peak heights within the central 200bp (100bp for ChIP-seq) by paired student's t-test to get the p-values. Additionally, as the reviewer pointed out, we replaced p-values with adjusted false discovery rate (FDR), and these results were shown as **Fig. 5a, b, f, Extended Data Fig. 6a, b, h-k and n-q** in the revised version, the conclusion still remains consistent with the previous version. And we have modified some sentences for clear description in the corresponding Methods section (**pages 46-48**).

In the RNA-seq analysis, we initially obtained the expression of each repetitive element (RE). These expression values were subsequently normalized using *Actb* as a negative control, and then depicted according to RE-class in the paired knock-out and wild-type samples. It is important to emphasize that, despite the y-axis in Fig. 5g being presented in a logarithmic format, the p-values were calculated by the Student's t-test of values prior to logarithm transformation. Accordingly, we have modified certain sentences for clear description in the corresponding Methods section (**page 45, lines 14-21**).

2. The third point raised by the reviewer is that controls need to be analyzed to show that the increased ATAC seq signal seen on REs is not some experimental artefact. The authors then show that ATAC-seq signal is increased almost at all elements they look at. To me this seems to confirm the concern of the reviewer that there may be some experimental artefact that makes all signals at peaks stronger (e.g., a higher signal-to-noise ratio). Can the authors rule this out? What about replicates? Do independent biological replicates always show this phenomenon?

R: We thank the reviewer for these insightful questions and professional suggestions. These concerns raised by the reviewer are important, to address it, we perform more analysis as following:

1) As the reviewer pointed out, to rule out the experimental artefact that makes all signals at peaks stronger, we perform the statistical analyses on the chromatin accessibility of *Actb* gene as a control locus, showing that chromatin accessibility of *Actb* locus in livers (n=2 biological replicates) from *Irtks* KO mice was not significantly changed, as compared to that from WT mice (see **Fig 1a below**), indicating that the ATAC-seq data from our study is available.

Fig 1a

Fig 1a. Visualization of chromatin accessibility at the control locus (*Actb*) of livers from WT and *Irtks* KO mice. n=2 biological replicates.

2) In addition, as the reviewer suggested, we re-analyzed the ATAC-seq data to assess the effect of IRTKS-mediated chromatin accessibility on RE regions in livers from WT and *Irtks* KO mice (n=2 biological replicates), showing that chromatin accessibility was visibly increased on most RE regions in livers from *Irtks* KO mice,

as compared with those from WT mice (see Fig 1b and c below), which is similar to the previous conclusion. Consistently, IRTKS deficiency leads to a significant increase of transcript levels of almost RE regions, as shown by RNA-seq analysis (Fig. 5g), which is similar to the effect of other known heterochromatin stabilizers such as CLOCK, SIRT3 and ZKSCAN3 on RE regions as previously reported (PMIDs: 32737416, 33706382 and 32427330). Similarly, we also have analyzed the ATAC-seq data of MEF cells previously reported (PMID: 37739210), showing that chromatin accessibility had no visible changes at *Actb* locus as a negative control, and effectively ruling out any experimental artefacts. However, a significant increase in chromatin accessibility was observed on RE regions in *Irtks*-deficient MEFs as compared to those in WT controls (Extended Data Fig. 6l, o). The conclusion still remains the same as in the previous version. In the revised version, these results were shown as Fig. 5f, Extended Data Fig. 6m, n.

Fig 1b

Fig 1b. Heatmaps showing the read signals of ATAC-seq peaks ranging from 5 kb upstream to 5 kb downstream of repetitive sequence regions (SINE, LTR, LINE, DNA transposon, low complexity and simple repeat) in livers from WT and *Irtks* KO mice. n=2 biological replicates.

3. Other concerns raised by reviewer 2 are well addressed.

R: We appreciate the reviewer for the positive comments on this revision.

4. This reviewer brings up several technical and experimental concerns, and these are generally addressed appropriately by the authors. The more critical concern from the reviewer is that the authors do not acknowledge extensive literature on HP1 that shows that its role in heterochromatin formation does not require LLPS. I think the authors do not do a good job addressing this very valid concern. This issue of LLPS and its role in HP1-mediated heterochromatin formation is hotly debated in the field, and I feel this manuscript can make a valuable contribution, but only when the authors present their data more neutrally: show all the results that identify a role of IRTKS in HP1-mediated heterochromatin formation and silencing, and also show that these proteins can form condensates (by themselves, with each other, and including chromatin). Then propose that these phenomena (formation of EDR, gene silencing on the one hand, and condensate formation on the other) are mechanistically linked, while acknowledging that this is not directly proven yet.

R: We greatly appreciate the reviewer for the insightful and professional comments. Overall, we do agree with the reviewer that the role of HP1 α in heterochromatin formation can be independently of droplet formation under some certain circumstances. The topic of liquid-liquid phase separation (LLPS) and its involvement in HP1-mediated heterochromatin formation remains a subject of controversy and warrants further investigation.

Indeed, several latest studies indicated that HP1 α can mediate the formation of heterochromatin foci through LLPS (PMIDs: 37549295, 28636597 and 31543422). These findings strongly support the notion that LLPS of HP1 α plays a role in heterochromatin formation. Furthermore, mounting published reported demonstrated that LLPS has been shown to be involved in heterochromatin formation through various other proteins, including methyl-CpG binding protein 2 (MeCP2) (PMIDs: 32698189, 32111972 and 35156529), linker histone H1 (PMID: 38309957), scaffold attachment factor B (SAFB) (PMID: 31677973) and 53BP1 (PMID: 35042897), in addition to HP1 α . In light of the ongoing debate in the field, we have rephrased certain sentences about LLPS, heterochromatin and HP1 α in Introduction, Results and Discussion section, where we have also cited different views on the issue (**page 5, lines 8-10; page 11, lines 8-9; page 23, lines 23-24; page 24, lines 6-11**).

Regarding the potential involvement of IRTKS in heterochromatin formation, a recent study published in *Nature* utilized mass spectrometry following NeutrAvidin pulldown to identify proteins bound to endogenous retroviruses (ERVs). In this study, IRTKS (also referred to as BAIAP2L1) and other established heterochromatin regulators like HP1 α (also known as CBX5) were notably recognized as ERV-bound proteins involved in the regulation of H3K9me3 levels, where the data was shown in the Supplementary Table 1 of the published report (PMID: 37938770). Moreover, we and our collaborators have reported that IRTKS may function as a nuclear protein involved in regulating the eukaryotic genome and chromatin, and specifically, our research indicates that IRTKS could increase the levels of H3K9me3 by facilitating the accumulation of SETDB1 (Cancer Lett. 2023, PMID: 37739210; GUT, 2018; PMID:

28647685; Nature Communications, 2015; PMID: 26348439). Accordingly, the statement has been added in Introduction and Discussion section in the revised version (page 4, lines 15-16; page 21, lines 22-25; page 22, lines 1-2). We believe that our current results, alongside these pivotal published studies, have unveiled a previously unrecognized role of IRTKS within heterochromatin, where it undergoes LLPS with the heterochromatin protein HP1 α .

Additionally, we thank the reviewer for pointing out that the mechanistic links between IRTKS-mediated heterochromatin and biomolecular condensation. We appreciate that this is a critical issue and endeavor to address it with additional experiments as following:

- 1) To investigate the impact of IRTKS phase separation on heterochromatin regulation, we first examined if IRTKS puncta were co-localized in the heterochromatin foci. Live-cell fluorescence images showed that EGFP-IRTKS, but not EGFP alone, mainly localized in the nucleus of mouse fibroblast NIH3T3 cells, and overlapped well with Hoechst-dense regions as indicated by the line scan analysis, which is similar to the observation as previously reported (PMID: 38101414). In addition, IRTKS puncta were showed in the cytoplasm and nucleus of human liver endothelial SK-Hep-1 cells, and also co-localized with the intensely Hoechst-stained regions.
- 2) Additionally, we conducted EGFP-tagged IRTKS truncations containing I-BAR, IDR and IDR-mutant (P to A mutant of IRTKS-IDR), and then overexpressed in NIH3T3 cells. Compared with the full-length IRTKS, IDR of IRTKS also was capable of forming puncta and co-localized well with the Hoechst signals, as shown by the line scan analysis of NIH3T3 cells. In contrast, I-BAR and IDR-mutant of IRTKS were dispersed without puncta-formation ability, suggesting that IRTKS condensates involved in heterochromatin formation. These new results were added as **Extended Data Fig. 5k-n**.
- 3) Furthermore, to further confirm the role of IRTKS condensates in heterochromatin organization, we overexpressed the EGFP-tagged IRTKS with different truncations (full-length, I-BAR, IDR and IDR-mutant) or EGFP control in MEF cells lacking endogenous IRTKS, showing that re-introduction of IRTKS-full length and IRTKS-IDR markedly restored the distribution of heterochromatin at the nuclear periphery and around the nucleolus in MEF cells of *Irtks* KO mice, as shown by transmission electron microscopy, while IRTKS-I-BAR only slightly rescued the EDRs of heterochromatin. In contrast, LLPS-defective IRTKS-IDR-mutant as well as EGFP control failed to restore the heterochromatin distribution in *Irtks* KO MEFs. These new results were provided in **Extended Data Fig. 5o, p**.
- 4) Moreover, considering that heterochromatin is characterized by abundant presence of repetitive sequences that are typically depressed, we performed chromatin immunoprecipitation-quantitative polymerase chain reaction (ChIP-qPCR) to investigate how IRTKS condensates influence the regulation of repetitive sequences associated with heterochromatin. The results showed that the occupation of H3K9me3 and HP1 α (heterochromatin markers) on known subtypes of genomic

repetitive sequences, including SINE, LINE1 and intracisternal A particle (IAP) (an LTR retrotransposon), was significantly decreased in MEFs from *Irtks* KO mice compared with those from WT mice. More significantly, in IRTKS-KO MEF cells, ectopic expression of IRTKS-full length and IRTKS-IDR, but not IRTKS-IDR-mutant and EGFP control, markedly increased the enrichment of H3K9me3 and HP1 α on these repetitive sequences. These data firmly support the importance of LLPS-forming ability of IRTKS in the regulation of heterochromatin. The new data was shown in **Extended Data Fig. 5q**.

5. I realize that I should not be bringing up new comments on the manuscript, but I find it somewhat odd that many cells have no IRTKS droplets, while having condensed chromatin domains, and that some IRTKS droplets are in the cytoplasm. These observations certainly suggest that (maintenance of) heterochromatic domains and IRTKS condensates can be uncoupled in some instances.

R: We thank the reviewer for valuable comment and suggestion, and apologize for making it confusing. In fact, we and our collaborators have reported that IRTKS, a member of the IRSp53/MIM homology domain family that is well known to play crucial roles in the formation of plasma membrane protrusions, is located in the cell cytoplasm and nucleus (PMIDs: 10931946, 19913105, 23896986, 26348439 and 28647685). Therefore, IRTKS droplets indeed could be observed in the cell cytoplasm and nucleus (**Fig. 4h-l and Extended Data Fig. 5k-n**).

Additionally, to address the concerns raised by reviewer, we examined whether IRTKS puncta accumulated within the heterochromatin foci in the mouse fibroblast NIH3T3 cells and human liver endothelial SK-Hep-1 cells, showing that EGFP-IRTKS mainly accumulated in the nucleus of NIH3T3 cells, and overlapped well with the intensely Hoechst-stained regions. In contrast, the empty vector expressing EGFP failed to lead to formation of puncta, ruling out the possibility that the puncta were artificially formed by the EGFP tag. Additionally, line scan analysis further confirmed the conclusion, similar to the observation as previously reported (PMID: 38101414). In addition, live-cell images and line scan analysis showed that IRTKS puncta in the cytoplasm and nucleus, at least partially, co-localized with Hoechst-dense regions. These new results, combined with our previous data, indicate that IRTKS puncta are likely to be, at least partially, localized in heterochromatin domains. In the revised version, these new data were provided in **Extended Data Fig. 5k-n**.

As accurately pointed out by the reviewer, we also noticed that the location of IRTKS puncta were slightly variant among different types of cells, and we speculate that these differences might arise from the following possibilities: (1) Unlike IRSp53, IRTKS lacks a CRIB domain, which could result in diverse functions in distinct subcellular locations. (2) The location of IRTKS was influenced by numerous factors under specific conditions. For example, insulin stimulation induced the translocation of IRTKS to the cell membrane, as previously reported in our study (Cell Research, 2013, PMID: 23896986). We thank the reviewer for raising this question, which will be helpful to our future exploration.

6. I also still find the merging of droplets not very convincing.

R: Many thanks for acknowledging the reviewer's thoughtful and important concerns. It is indeed noteworthy that IRSP53, the founder of the inverse-BAR (I-BAR) domain family, shares similarities with IRTKS without a CRIB domain. Given that IRSP53 has been shown to undergo liquid-liquid phase separation, as reported previously (PMID: 35819332), we thus believe that IRTKS may also possess the capability to form phase-separated droplets.

To address the reviewer's concern, we have performed more experiments to support the phase-separated properties of IRTKS. These include:

- 1) Time-lapse e fluorescence images and fluorescence recovery after photobleaching (FRAP) experiments, showing that EGFP-IRTKS puncta underwent fusion over time and displayed the dynamic feature of liquid droplets (**Fig. 3n and Extended Data Fig. 4x**), in accordance with phase-separated behaviors.
- 2) Engineering HEK293T cells by using the CRISPR-Cas9 system to tag endogenous IRTKS with monomeric enhanced green fluorescent protein (mEGFP), showing that the mEGFP-tagged endogenous IRTKS formed condensates in the cytoplasm or nucleus, and that the endogenous mEGFP-IRTKS puncta could rapidly fuse and fission (**Extended Data Fig. 4u and v**).
- 3) Utilizing FRAP experiment to examine the dynamic characteristic of these endogenous mEGFP-IRTKS puncta, showing that endogenous mEGFP-IRTKS puncta could be recovered after photobleaching (**Fig. 3k and Extended Data Fig. 4w**).

We hope these additional experiments, along with other evidences in the previous version, will address the reviewer's concern and further substantiate our conclusion that IRTKS possesses phase-separated properties.

Dear Dr. Han,

Thank you for submitting your manuscript for consideration by the EMBO Journal. It has now been seen by referee #5 whose comments are enclosed. As you will see, this referee thinks that you have sufficiently addressed many of the concerns raised during the previous rounds of revisions but still remarks that some of the conclusions are overstated and do not properly reflect the conclusions that can be drawn from the data.

Given the referees' positive recommendations, I would like to invite you to submit a revised version of the manuscript, addressing the comments of referee #5 in the last and previous reports to describe and interpret the results in a balanced and critical manner. I should add that it is EMBO Journal policy to allow only a single round of revision, and acceptance of your manuscript will therefore depend on the completeness of your responses in this revised version.

We generally allow three months as standard revision time. As a matter of policy, competing manuscripts published during this period will not negatively impact on our assessment of the conceptual advance presented by your study. However, we request that you contact the editor as soon as possible upon publication of any related work, to discuss how to proceed.

Thank you for the opportunity to consider your work for publication. I look forward to your revision.

Yours sincerely,

Cornelius Schneider, PhD
Editor
The EMBO Journal
c.schneider@embojournal.org

We realize that it is difficult to revise to a specific deadline. In the interest of protecting the conceptual advance provided by the

work, we recommend a revision within 3 months (3rd Sep 2024). Please discuss the revision progress ahead of this time with the editor if you require more time to complete the revisions. Use the link below to submit your revision:

Referee #1:

I have reviewed this manuscript before, and was surprised NCB rejected it. The paper is overloaded with data, which makes it overwhelming. My main concern was the authors appeared to force-fit the results into popular and overhyped models related to aging, senescence, genome stability and all through the lens of phase separation. I felt there was no need to hype the story so much, and I had asked the authors to be more self-critical and just present the data more matter-of-factly. I still think they can do a better job describing the results in a more balanced way, and also acknowledge that phase separation of HP1 is in some more recent studies no longer believed to be dependent on LLPS.

I still think it is odd that some cells do not have IRTKS droplets, and the authors do not really respond to that comment.

Other than that I think the paper is acceptable for publication. The authors have done an enormous amount of work to address the comments, and I think it is not right to ask for even more.

The point-by-point response to the editor and referee comments

We sincerely thank the Editor and all the reviewers for their constructive comments on our manuscript, which are very helpful to improve our study. In the revised version, we have endeavored to address all concerns raised by the reviewer. We believe these changes have increased the accuracy and clarity of the revised version.

Prior to presenting a detailed point-by-point response to the reviewer, we would like to outline the major revisions as follows:

1. As suggested by Reviewer #1, we have revised the text in Introduction, Results and Discussion section, where we have cited different views on the issue about the relationship between HP1 α -mediated heterochromatin and liquid-liquid phase separation, acknowledged the limitation of our study, highlighted the direction of our future exploration, and presented our data in a more balanced and self-critical manner (**page 5, lines 18-19; page 11, lines 15-16; page 24, lines 3-25; page 25, line 1**).

2. To address the concerns raised by Reviewer #1, we had performed the live-cell imaging and line scan analysis to verify the accumulation of IRTKS condensates within the heterochromatin foci in both NIH3T3 and SK-Hep-1 cells (**Fig. EV5K-N**). In the Discussion section, we also mentioned the slightly different subcellular location of IRTKS puncta among various types of cells and provided potential explanations for this observation (**page 25, lines 10-18**).

3. According to the editor's professional recommendation and the editorial policy of *EMBO J*, we made some changes to the manuscript as below:

- We shortened the abstract to less than 175 words, and added 5 keyword terms on the abstract page of the manuscript.
- We moved Expanded View (EV) Figures 6 and 7 to the Appendix section and referenced them as Appendix Figures S1 and 2.
- The textual changes are highlighted in blue in the revision.

Point-by-point response to Reviewers

Reviewer: 1

I have reviewed this manuscript before, and was surprised NCB rejected it. The paper is overloaded with data, which makes it overwhelming.

R: We would like to thank the reviewer for the constructive and expert comments on improving our manuscript. We also appreciate the critiques pointing out that a few key concerns need to be addressed to enhance the persuasiveness of our findings. We have rephrased some statements of the manuscript for the clarity and preciseness.

My main concern was the authors appeared to force-fit the results into popular and overhyped models related to aging, senescence, genome stability and all through the

lens of phase separation. I felt there was no need to hype the story so much, and I had asked the authors to be more self-critical and just present the data more matter-of-factly. I still think they can do a better job describing the results in a more balanced way, and also acknowledge that phase separation of HP1 is in some more recent studies no longer believed to be dependent on LLPS.

R: We very appreciate the reviewer for the insightful and professional comments. In general, we do agree with the reviewer that the issue about LLPS-driven heterochromatin formation is in controversial, which warrants further investigation. We also acknowledge that the role of HP1 α in heterochromatin formation can be independently of droplet formation under certain circumstances.

Herein, as suggested by the reviewer, we have amended the text in Introduction, Results and Discussion sections, where we have cited different views on the issue, acknowledged the limitation of our study, highlighted the direction of our future exploration, and presented our data in a more balanced and self-critical manner (**page 5, lines 18-19; page 11, lines 15-16; page 24, lines 3-25; page 25, line 1**). We hope these modifications could address the reviewer's concerns.

I still think it is odd that some cells do not have IRTKS droplets, and the authors do not really respond to that comment.

R: We thank the reviewer for valuable comment, and apologize for making it confusing in the previous version. As indicated by the reviewer, we also noticed that the subcellular location of IRTKS puncta were slightly variant among different types of cells, some of them have no IRTKS puncta in condensed chromatin domains. We speculate that these differences might arise from the following possibilities:

(1) Unlike IRSp53, the founder of I-BAR protein family, IRTKS lacks a CRIB domain that binds specifically to Cdc42, a plasma membrane-associated small GTPase. Therefore, IRTKS could play diverse functions upon its distinct subcellular locations and actually, we and others have reported that IRTKS is located in cell cytoplasm and nucleus (PMIDs: 10931946, 19913105, 23896986, 26348439 and 28647685). Consistent with these previous reports, IRTKS puncta were indeed observable in the cell cytoplasm and nucleus in this study (**Fig. 4H-L and Fig. EV5K-N**).

(2) The subcellular location of IRTKS was influenced by numerous factors under specific conditions. For example, insulin stimulation may induce the translocation of IRTKS to the cell membrane, as previously reported in our study (Cell Research, 2013, PMID: 23896986).

Additionally, to address the concerns raised by the reviewer, we had conducted additional experiments to examine whether IRTKS puncta accumulated within the heterochromatin foci in the mouse fibroblast NIH3T3 cells and human liver endothelial SK-Hep-1 cells, showing that EGFP-IRTKS mainly accumulated in the nucleus of NIH3T3 cells, and overlapped well with the intensely Hoechst-stained

regions. In contrast, the empty vector expressing EGFP failed to lead to formation of puncta, ruling out the possibility that the puncta were artificially formed by the EGFP tag. Moreover, line scan analysis further confirmed the conclusion, similar to the observation as previously reported (PMID: 38101414). In addition, live-cell images and line scan analysis showed that IRTKS puncta in the nucleus, at least partially, co-localized with Hoechst-dense regions. These new results, combined with our previous data, strongly suggest that IRTKS puncta are likely to be, to some extent, localized within heterochromatin domains. In the revised version, these data were provided in **Fig. EV5K-N**.

We thank the reviewer for raising this question, which will be helpful to our future exploration. In the revision, we mentioned the slightly different subcellular location of IRTKS puncta among various types of cells, and provided the possible explanation in the Discussion section (**page 25, lines 10-18**). We hope these additional experiments, along with other evidences in the previous version, will address the reviewer's concern.

Other than that I think the paper is acceptable for publication. The authors have done an enormous amount of work to address the comments, and I think it is not right to ask for even more.

R: We sincerely appreciate the reviewer for the constructive comments during the review process, which significantly improved the manuscript.

Dear Dr Han,

Thank you for submitting a revised version of your manuscript. We find that you have addressed all the additional remarks raised by the referee. There remain only a few mainly editorial points that have to be addressed before I can extend formal acceptance of the manuscript:

1. FUNDING INFO: missing in eJP: National Science and Technology Major Project (2017ZX10203207); Shanghai Jiao Tong University Scientific and Technological Innovation Funds (2019TPA09)
2. REFERENCE FORMAT: should be 10 authors + et al. instead of 20 authors + et al.
3. CRediT has replaced the traditional author contributions section because it offers a systematic, machine-readable author contributions format that allows for more effective research assessment. Please remove the Authors Contributions from the manuscript and use the free text boxes beneath each contributing author's name in our online submission system to add specific details on the author's contribution. More information is available in our guide to authors.
4. FIGURE CALLOUTS: Figure 4 should be called out after Fig. 3, not after the callout for Fig. 3B; missing callouts for Dataset EV1-EV2; there is a callout for Supplementary Table 2/Supplementary Table 2, but no such table uploaded
5. Figures in separate files: yes, but there should be only 5 EV figures, not EV1-1, EV1-2, EV2-1, EV2-2, etc. Each figure should be uploaded as one figure file named Figure EV1, Figure EV2, etc.
6. DATASET EV LEGENDS: dataset legends should be removed from the "README" file and uploaded as a separate sheet in each Excel file
7. APPENDIX 1 FILE WITH ToC: Appendix figures should be only uploaded in Appendix PDF, no need to upload them individually; page numbers missing in ToC
8. Please fill out the SD checklist; Please make sure to correct the naming 'panel level folders' for all figures, e.g. in the folder 'figure 4' the subfolders are named "1A", where it would be expected '4A' etc. Please also double-check the READ.ME file uploaded with SD for Fig. 5 - it should say 5A, B,E,F,G instead of 1A, B,E,F,G.; Source data files need to be reorganized to one file/folder per figure and ZIPping for each main figure. For EV and/or appendix figures, ZIP together all source data.
9. The synopsis image should have a size of 550x300-600 pixels large (width x height, jpeg or png format).
10. Please note that the specific URLs for PRJNA888247 and PRJNA714184 datasets are not provided in the data availability statement.
11. Figure Legends (main + EV): 1. Please define the annotated p values **/**/*** as well as provide the exact p-values for the same in the legend of figure 1b, d, f; 2a, d-e; 3c-d; 4b, d; 5a-d, f-g; 6b-e, g-k; EV 1a-b, d-f, h-i, k-l, n-o, v, x; EV 2d; EV 3k-l; EV 4d, g, i, o, q, r; EV 5d, f, p-q; as appropriate.
12. Please note that information related to n is missing in the legends of figures EV 1v, x.
13. Although 'n' is provided, please describe the nature of entity for 'n' in the legends of figures 1b, d, f; 2a, d-e, g; 3c-d, f, m-n; 4a, d; EV 1h-i, k-l, n-o; EV 2d; EV 3k-l; EV 4d, g, i, w; EV 5d, f, p."
14. Please note that the white arrows are not defined in the legend of figure EV 1p, t-u, w; EV 5k, m.
15. Please note that the red arrows are not defined in the legend of figure 1e; 2a; EV 1g, j, m, v, x; EV 2d; EV 5o.
16. 14 movie files - playing. The legends should be removed from the "README" file and zipped with each movie file.

With best regards,

Cornelius

Cornelius Schneider, PhD
Editor | The EMBO Journal
c.schneider@embojournal.org

- a point-by-point response to the referees' comments, with a detailed description of the changes made (as a word file).

- a word file of the manuscript text.

- individual production quality figure files (one file per figure)

- a complete author checklist, which you can download from our author guidelines

(<https://www.embopress.org/page/journal/14602075/authorguide>).

- Expanded View files (replacing Supplementary Information)

We realize that it is difficult to revise to a specific deadline. In the interest of protecting the conceptual advance provided by the work, we recommend a revision within 3 months (23rd Sep 2024). Please discuss the revision progress ahead of this time with the editor if you require more time to complete the revisions. Use the link below to submit your revision:

The point-by-point response to the editor

In the revised version, we have carefully addressed all editorial points mentioned in your revision guidelines, and all changes have been highlighted in blue color within the manuscript.

Point-by-point response

Thank you for submitting a revised version of your manuscript. We find that you have addressed all the additional remarks raised by the referee. There remain only a few mainly editorial points that have to be addressed before I can extend formal acceptance of the manuscript:

1. FUNDING INFO: missing in eJP: National Science and Technology Major Project (2017ZX10203207); Shanghai Jiao Tong University Scientific and Technological Innovation Funds (2019TPA09).

R: We very appreciate the editor for careful inspection. We have added the information related to these fundings in eJP.

2. REFERENCE FORMAT: should be 10 authors + et al. instead of 20 authors + et al.

R: We thank the reviewer for valuable comment. We modified the reference format, as suggested by the editor (pages 41-44).

3. CRediT has replaced the traditional author contributions section because it offers a systematic, machine-readable author contributions format that allows for more effective research assessment. Please remove the Authors Contributions from the manuscript and use the free text boxes beneath each contributing author's name in our online submission system to add specific details on the author's contribution. More information is available in our guide to authors.

R: Many thanks for these valuable suggestions. We removed the Authors Contributions from the previous manuscript, and specified the contribution of each author directly in the Author Information page of the submission system.

4. FIGURE CALLOUTS: Figure 4 should be called out after Fig. 3, not after the callout for Fig. 3B; missing callouts for Dataset EV1-EV2; there is a callout for Supplementary Table 2/Supplementary Table 2, but no such table uploaded.

R: We sincerely appreciated the editor for careful inspection and professional suggestion. We have corrected the callout of Fig. 4 after Fig.3, replaced by Supplementary Table 2 with Dataset EV2, and also added the callout for Dataset EV1

in the revised version (page 12, lines 1-9; page 29, line 3; page 29, line 10; page 34, line 3; page 35, line 22; page 36, line 9).

5. Figures in separate files: yes, but there should be only 5 EV figures, not EV1-1, EV1-2, EV2-1, EV2-2, etc. Each figure should be uploaded as one figure file named Figure EV1, Figure EV2, etc.

R: We sincerely appreciate the editor for the professional suggestions. In the light of editor's advice, we corrected each figure as a single figure file named Figures EV1, EV2, EV3, EV4 and EV5, respectively.

6. DATASET EV LEGENDS: dataset legends should be removed from the "README" file and uploaded as a separate sheet in each Excel file.

R: Many thanks for the professional suggestions. According to the advice of the editor, we removed the dataset legends from "README" file, and uploaded them as a separate sheet in each Excel file.

7. APPENDIX 1 FILE WITH ToC: Appendix figures should be only uploaded in Appendix PDF, no need to upload them individually; page numbers missing in ToC.

R: We sincerely appreciate the editor for the constructive comments. In the revised version, we added page numbers in ToC of Appendix file, and consolidated all Appendix figures into the Appendix PDF file.

8. Please fill out the SD checklist; Please make sure to correct the naming 'panel level folders' for all figures, e.g. in the folder 'figure 4' the subfolders are named "1A", where it would be expected '4A' etc. Please also double-check the READ.ME file uploaded with SD for Fig. 5 - it should say 5A, B,E,F,G instead of 1A, B,E,F,G.; Source data files need to be reorganized to one file/folder per figure and ZIPing for each main figure. For EV and/or appendix figures, ZIP together all source data.

R: Many thanks for these valuable suggestions. As suggested by the editor, we have filled out the SD checklist, and carefully confirmed the naming 'panel level folders' for all figures and checked the READ.ME file uploaded with Source Data. Additionally, we also reorganize the Source data files of main figures, EV and appendix figures according to the editor's advice.

9. The synopsis image should have a size of 550x300-600 pixels large (width x height, jpeg or png format).

R: We thank the editor for raising this important point. We provided a synopsis image with a size of 550 × 546 pixels large.

10. Please note that the specific URLs for PRJNA888247 and PRJNA714184 datasets

are not provided in the data availability statement.

R: We sincerely appreciate the editor for the valuable comments. In the revised version, we provided the URLs for PRJNA888247 and PRJNA714184 in the data availability section (**page 39, lines 1-2**).

11. Figure Legends (main + EV): 1. Please define the annotated p values */**/** as well as provide the exact p-values for the same in the legend of figure 1b, d, f; 2a, d-e; 3c-d; 4b, d; 5a-d, f-g; 6b-e, g-k; EV 1a-b, d-f, h-i, k-l, n-o, v, x; EV 2d; EV 3k-l; EV 4d, g, i, o, q, r; EV 5d, f, p-q; as appropriate.

R: Many thanks for the valuable suggestion. As suggested by the editor, we have provided the exact *p*-values in the corresponding figure legends (**pages 45-50; pages 52-60**).

12. Please note that information related to n is missing in the legends of figures EV 1v, x.

R: We thank the editor for pointing this out, we have added the information related to 'n' in the legends of figures EV 1V, X (**page 53, lines 11-12; page 53, lines 18-19**).

13. Although 'n' is provided, please describe the nature of entity for 'n' in the legends of figures 1b, d, f; 2a, d-e, g; 3c-d, f, m-n; 4a, d; EV 1h-i, k-l, n-o; EV 2d; EV 3k-l; EV 4d, g, i, w; EV 5d, f, p."

R: We sincerely appreciate the editor for the constructive comments. In the revised version, we described the nature of entity for 'n' in the legends of figures mentioned above (**pages 45-60**).

14. Please note that the white arrows are not defined in the legend of figure EV 1p, t-u, w; EV 5k, m.

R: We applaud the editor for careful inspection of the data. As suggested by the editor, we have defined the white arrows in the corresponding figure legends in the revised legends (**page 52; pages 59-60**).

15. Please note that the red arrows are not defined in the legend of figure 1e; 2a; EV 1g, j, m, v, x; EV 2d; EV 5o.

R: Many thanks for the professional suggestion, and we apologize for not defining the red arrows in the legend of figures mentioned above. In the revised version, we added the definition of the red arrows in the corresponding figure legends (**pages 45-46; pages 52-54; page 60**).

16. 14 movie files - playing. The legends should be removed from the "README"

file and zipped with each movie file.

R: Thanks very much for the editor' comment. We removed the legends from the "README" file and zipped them with each movie file.

Dear Prof. Han,

I am pleased to inform you that your manuscript has been accepted for publication in the EMBO Journal.

Yours sincerely,

Cornelius Schneider, PhD
Editor
The EMBO Journal
c.schneider@embojournal.org
